# Comparison of Quebec's Project Delivery Methods: Relational Contract Law and Differences in Contractual Language

**Gabriel Jobidon** [1,*]**, Pierre Lemieux** [2] **and Robert Beauregard** [3]

[1] CIRCERB–CRMR, Faculty of Law, Université Laval, 2325 Rue de l'Université, Ville de Québec, QC G1V 0A6, Canada

[2] Faculty of Law, Université Laval, 2325 Rue de l'Université, Ville de Québec, QC G1V 0A6, Canada; pierre.lemieux@fd.ulaval.ca

[3] Academic and Student Affairs, Université Laval, 2320 Rue des Bibliothèques, Ville de Québec, QC G1V 0A6, Canada; vice-recteur@vre.ulaval.ca

[*] Correspondence: gabriel.jobidon.1@ulaval.ca; Tel.: +1-581-989-1032

**Abstract:** The province of Quebec, Canada, seeks to implement relational alternate project delivery methods to achieve sustainability and energy efficiency in public construction. However, the relational differences between the formal written parts of different delivery methods have yet to be analyzed and understood, as is the case with the relational aspects of contracts and the achievement of sustainable and energy-efficient infrastructure. Using a hermeneutic interpretation of Macneil's relational contract norms and grounded theory, 26 contracts involving Quebec's largest public client of vertical infrastructure and representing three different types of project delivery methods (design–bid–build (DBB), design–Build (DB), and construction manager–general contractor/integrated project delivery (CMGC/IPD)) were analyzed using NVivo. It was found that CMGC/IPD is the most relational project delivery method available to Quebec's public clients, namely because of the public client's active involvement in the realization process, the increasing complexity of roles, the multitude of common management structures, and the internalization of sustainability measures and conflict resolution. Furthermore, Quebec's CMGC/IPD was found to be an IPD-ish delivery method, lacking the early involvement of the construction manager and the risk/reward sharing mechanisms necessary to achieve pure IPD status. The findings and theoretical considerations discussed here will help policymakers, contract drafters, and public clients interested in implementing relational contracting practices in public construction projects.

**Keywords:** public procurement; project delivery methods; integrated project delivery; relational contracts; contractual language; sustainable infrastructure; energy efficiency

## 1. Introduction

Transactional contracts put emphasis on legal rules, formal documents, and self-liquidating transactions (Williamson 1979). These contracts, traditionally used in public construction procurement, have created problems for practitioners, in part, because of the impossibility of predicting every possible outcome that may, and will, arise during a long-term relationship (MacNeil 1974a, 1980a). In other words, contracts are inevitably and inherently incomplete (Hart 2003). Rather than focusing on project goals, transactional contracts are segmented, sequential, and create a silo effect, leading to a focus on individual goals (Ghassemi and Becerik-Gerber 2011). The construction industry has suffered from the repeated use of formal and transactional contracts, leading to adversarial relationships, low productivity rates, poor overall quality, and the use of inefficient methods, causing job recommencement

(Egan 2002; Latham 1994; Lichtig 2006). Furthermore, transactional contracts have been found to have a constraining effect on innovation and creativity (Walker and Hampson 2008). The public sector has been specifically identified as a fragmented, specialized, and litigation-prone industry (Alashwal and Fong 2015). Multiple contracts between different firms not always having a precedent knowledge of one another, the absence of risk/reward sharing mechanisms, and the existence of a procurement regulatory framework considered by researchers to be an exogenous uncertainty create ex-ante transactional and infrastructural complexity (Roehrich and Lewis 2014; Zheng et al. 2008).

On the other side of the contractual spectrum resides relational contracting. Rather than traditional adversarial relationships, the focus is on reciprocity, good faith, trust, and greater social capital to foster individual and collective prosperity (Latham 1994). The industry has turned towards more integrated and relational delivery methods, such as design–build (DB), construction manager at risk (CMAR), construction manager/general contractor (CMGC), public–private partnerships (PPPs), and integrated project delivery (IPD). These project delivery methods have in common an earlier involvement of key stakeholders, namely the general contractor, whether as part of an integrated team (joint venture, partnering, alliancing, DB, IPD, and PPPs) or as a consultant and manager (CMAR, CMGC), and exhibit superior performance. For instance, IPD was proven to have a superior performance compared with more traditional delivery methods in metrics related to quality, communication, and change performance (El Asmar et al. 2013). Quebec's Société québécoise des infrastructures (SQI), a public entity responsible for advising the government on public infrastructure projects and providing contractual services to ministries and other public entities, wishes to implement IPD and the building information modeling (BIM) process in every major project (Société Québécoise des Infrastructures 2016b), and its approach to contracts is the focal point of this paper.

Alternate delivery methods rely on more relational governance structures and contracts. Relational contract theory was introduced by Macauley and formalized by Macneil (Macaulay 2018; MacNeil 1980a). Macneil identifies 10 common contractual norms present in every exchange relationship, namely role integrity, reciprocity, planning, obtaining consent, flexibility, solidarity, linking or cohesive norms (restitution, reliance, and expectation), power, propriety of means, and harmonization with the social matrix. These norms—meaning how contracting parties must, or should, behave during the exchange—are interpreted according to a discrete/relational spectrum. Many researchers have validated and used these common contractual norms in marketing, management, construction, and legal research (Blois et al. 2006; Blois and Ivens 2007; Harper et al. 2016; Harper 2014; Harries and Vincent-Jones 2001; Heide and John 1992; Kaufmann and Stern 1988; Kaufmann and Dant 1992; Vincent-Jones and Harries 1998; Vincent-Jones 2012). While studies have considered transactional and relational contracting as general substitutes (Dyer and Singh 1998; Gulati 1995; Larson 1992; Macaulay 2018), others have found them to have a complementary role (Cao and Lumineau 2015; Lorenz 1999; Poppo and Zenger 2002; Sitkin 1992).

In public contract theory, there is an obligation to use formal written contracts to best serve the public interest. Therefore, relational contracting needs be understood as a complementary means to formal contractual governance. Relational mechanisms do, however, exist in the public procurement process for construction work and counterbalance the transactional features of the regulatory framework (Harries and Vincent-Jones 2001; Macneil and Campbell 2001; Vincent-Jones and Harries 1998). Most researchers have analyzed either the external level of relational contract theory, such as regulatory frameworks, or the internal level, which deals with the actual outcomes of the relationship, but little has been said about how contractual arrangements affect a team's ability to innovate (Forgues and Koskela 2009). Some researchers have conducted a comparison of project delivery methods using the frequency of occurrences of relational contracting language as a variable and found that the construction industry needs to introduce different contractual language into project delivery methods (Harper et al. 2016; Harper 2014). Still, no research has identified the material differences in contractual language and the mechanisms between less relational and more relational public contracts,

even though Macneil assumes relational contract theory is useful for contract planning, as pointed out by Diathesopoulos and illustrated in Macneil's work (Diathesopoulos 2010).

The aim of this research is, therefore, to build on pre-existing literature concerning relational and transactional features of different project delivery methods and to identify specific relational content or mechanisms that can complement the formal contractual public procurement process. IPD projects are complex and inherently uncertain because of the long-term nature of the arrangement. Because different contract types offer different characteristics, some delivery methods are more suited to the use of relational contracting principles (Cheung et al. 2006). In the present paper, we will demonstrate the relational gaps between different delivery methods. We will also identify the principal differences between the IPD-ish delivery method used in Quebec and pure IPD. Finally, because IPD seeks to achieve improved energy-efficient and environmental performance, we will illustrate the different path provided for by CMGC/IPD to achieve environmental and sustainable objectives through informal and relational governance mechanisms. The findings and discussion developed here will help policymakers, contract drafters, and public agencies interested in implementing relational contracting practices in public construction projects.

## 2. Conceptual Background

To understand the context of this research, it is necessary to describe the evolution of contract theory. Authors of diverse fields, such as law, economics, governance, construction management, and marketing, have analyzed the transactional and relational aspects of commercial relationships and have identified best practices. This section will also describe the different delivery methods used in Quebec and their respective strengths and weaknesses and present the general principles of sustainable construction.

### 2.1. From Classical to Relational Contract Law

This subsection describes the principal characteristics of classical, neoclassical, and relational contract law, the latter having recently gained traction in the construction field. C such as discreteness, presentation, contract incompleteness, flexibility, and solidarity are discussed, as well as their application to the field of construction.

#### 2.1.1. Classical Contract Law

Classical contract law was developed in the 19th century and epitomized by Samuel Williston's work, *The Law of Contracts* and *Restatement of Contracts* (Williston and Lewis 1920). Neoclassical contract law is founded on the same overall system but with substantial modifications regarding some aspects (MacNeil 1978). With contracts becoming more complex and of longer duration, neoclassical adjustment processes were slowly pushed aside in favor of more transaction-specific, ongoing administrative processes proposed by relational contracting (MacNeil 1978; Williamson 1979).

Classical contract law attempts to facilitate exchange—a purpose shared by any system of contract law (MacNeil 1978; Williamson 1979)—by enhancing discreteness and intensifying presentation. Discreteness is materialized through a series of considerations: (1) The identities of parties to a transaction are treated as irrelevant; (2) the subject matter of contracts is commodified; (3) formal sources trump informal sources as the substantive content of the transaction; (4) limited contract remedies are available to facilitate predictability; (5) clear lines are drawn between being in or out of a transaction; and (6) third-party participation in the transaction is discouraged, because multiple poles of interest could be harmful to discreteness (MacNeil 1978). Presentation of a transaction means restricting its expected future effects to those at the inception of the transaction (MacNeil 1978). Classical law emphasizes presentation through mechanisms such as (1) equating the promises creating a transaction to their legal effects, (2) providing for a predictable and precise body of law to deal with the gaps left out by the promises so that the parties know what the future holds, and (3) emphasis on expectation remedies—a way to calculate all theoretical risks at the moment the deal is struck

(MacNeil 1978). In summary, classical contract law stresses legal rules, formal documents, and self-liquidating transactions (Williamson 1979).

A theoretically fully discrete transaction would be separate from all other past, present, and future relations (MacNeil 1978). The classic example would be to buy a pack of gum from a convenience store in a city where you have never been and where you will never return. Theoretically, fully discrete transactions are scholars' playgrounds but have limited, if any, practical applications. Any bargaining regarding quantities or price, past dealings, reputation, or projection of the transaction in the future through promises reduces discreteness, which is nonetheless present in every transaction (MacNeil 1978). Indeed, quite discrete transactions do occur, and they are characterized by little or no social exchange (Blau 1964), little personal involvement of the key stakeholders, easily monetized commodities, and no significant consideration of reputation or future relations (Campbell and Harris 1993; MacNeil 1978). In our advanced economies, relying greatly on the phenomenon of the specialization of labor and organic solidarity (Durkheim 1893) and, especially, on the procurement of construction work, in which high levels of specialization and planning are necessary, discrete transactions seem to fall short as a means of production. Their inherent rigidity, as opposed to internal flexibility, and their heavy focus on promises, legal rules, and formal governance mechanisms have led to the emergence of neoclassical contract law, which partially frees itself of the difficulties of classical contract law (Williamson 1979).

2.1.2. Neoclassical Contract Law

The emergence of concepts such as good faith—an anti-presentation and anti-discrete notion (MacNeil 1978)—coupled with the desire for greater flexibility to adjust to unforeseeable problems that will inevitably arise over the course of a relationship, are characteristics of neoclassical contract law. Long-term arrangements in a complex industry facing conditions of uncertainty render the use of presentation techniques practically impossible because of the contract's inherently incomplete nature (Hart 2003; MacNeil 1974a; Williamson 1979) and the contract planner's quest for flexibility (MacNeil 1978). Because all future contingencies for which adaptation will be required cannot be fully anticipated at the beginning of the transaction, a different contracting framework that enables the preservation of the relationship and therefore of the exchange, while also providing an additional governance structure, had to emerge.

Neoclassical contract law mainly concerns flexibility—a way to structure the exchange and the evolution of the key stakeholders' relationship—which could lead to cost and time savings in future disputes (Nystén-Haarala et al. 2010). The use of standards, direct determination of performance by third parties, one-party control of terms, cost, and agreements to agree are considered flexible mechanisms of neoclassical contract law (MacNeil 1978). External standards are not usually controlled by either of the parties or by an unrelated entity or party. Balance needs to be struck between sufficient flexibility and too much flexibility to ensure that parties will not have to repeatedly negotiate values or performance. Examples of external standards include certification agencies, such as the International Organization for Standardization (ISO), Leadership in Energy and Environmental Design (LEED), or the National Building Code of Canada, and the norms they produce.

Direct third-party determination of performance, while not offering a guarantee of smooth performance in the context of construction projects (MacNeil 1978), refers to the use of an expert to determine contract content or performance. To this end, Macneil extensively cites the example of the architect's role under the American Institute of Architects standardized construction contracts because of their role in determining the general administration of the contract. To this we could add engineering firms' control of general contractors' performance in terms of certifying when a project reaches a certain stage, such as substantial completion. The same goes for the mediation and arbitration process for the harmonization of conflicts affecting a relationship, which offers a less formal procedural approach than litigation, with the arbitrator being able to frequently interrupt witnesses with questions in an informal education process, thereby enabling the parties to proceed more intelligently with the case (Fuller 1963). Continuity, or at least the completion of the contract, is therefore presumed in an arbitration process

(Williamson 1979), which is not always the case with litigation. In cases where external standards and direct third-party determination of performance mechanisms are not used, the contract may designate one party to define parts of the relationship, such as termination mechanisms (MacNeil 1978). Even if this is considered to be a more relational approach, this mechanism still has a heavy weight in the power balance, or imbalance, of the relationship.

Cost can also be looked at as being a flexible mechanism if it provides compensation to the supplier for the expenses incurred in the production of the good or service, to which additional fees can be added, such as an anticipated overhead, with or without a definition of what is a cost (MacNeil 1978). Cost-plus contracts, as opposed to fixed-price contracts, are a prime example of a flexibility mechanism, because the contractors' expenses are all covered plus additional payments to allow a level of profit. Finally, agreements to agree refer to parties expressing their willingness to participate in a future process of expression of will (MacNeil 1978). The inherent incomplete nature of long-term arrangements can lead parties to agree on details of the relationship's performance at the time it best suits them. However, because there are no guarantees that parties will agree to terms at said time, contract drafters should prepare an alternative to fill the gaps of the contract.

Redetermination and renegotiation mechanisms represent other flexibility mechanisms (Crocker and Reynolds 1993) and are considered to be viable options in uncertain or changing market conditions that could potentially affect the contractual relationship, as is the case in long-term arrangements (Crocker and Masten 1991; Nystén-Haarala et al. 2010). Redetermination implies the reassessment, or change, of a value or direct measurement of performance, as predetermined in the contract. On the other hand, renegotiation means changing the agreed-upon contract terms through a new negotiation and therefore concerns the decision-making process, which is an organic way of dealing with changes and uncertainties as opposed to more mechanical redetermination mechanisms. In a way, redetermination mechanisms could be linked to the use of external standards, cost measures, or the unilateral control of terms by a party, whereas renegotiation mechanisms relate to third-party direct determination of performance and agreements to agree. Contracts including renegotiation modalities should specify what is to happen in the event no agreement is reached between the parties (Alleman et al. 2014). Flexibility measures are essential because of the incomplete nature of long-term arrangements and are considered to be the most cost-effective form of governance for this type of contract (Campbell and Harris 1993).

Neoclassical contract law tries to move away from presentation and discreteness by acting upon flexibility and recognizing the conflict between planning for all future contingencies and the inevitable changes that will occur throughout a relationship and by being willing to take action when such changes arise instead of only awarding monetary judgments (MacNeil 1978). In other words, neoclassical contract law focuses more on the preservation of the relationship than its classical counterpart. However, this system has been proven to have shortcomings when the self-interest or other motivations of the parties impede the continuation of the relationship. Because the overall structure of neoclassical contract law resembles classical contract law, it can be unsuitable for dealing with relational issues (MacNeil 1978).

### 2.1.3. Relational Contract Law

A classic way to look at the financial records of a company is to consider the balance sheet as a snapshot of what the company owns and owes at a specific moment in time, whereas the income statement is more of a video showing the income and expenses generated during a given period (Levitt 1998). The same analogy applies to the main difference between neoclassical contract law and relational contract law. In neoclassical contracts, the reference point for raised issues concerning changes is the original agreement or the framed picture of the initial contract, whereas relational approaches use the entire relation and the way it has evolved up to the time of the change issue arising—a video of the relationship evolution (MacNeil 1978; Williamson 1979). Increasingly long and complex contracts and the necessity to sustain long-term relationships have led to the displacement of

neoclassical contract law by the more transaction-specific, ongoing administrative adjustment process of relational contract law in which discreteness and presentation become just another decision factor (MacNeil 1978; Williamson 1979).

Relational contract theory is derived from empirical works led by Macaulay (Macaulay 1963), as well as Beale and Dugdale (Beale and Dugdale 1975), before being theorized by Macneil (MacNeil 1980a). It was found that contract law is often not followed in business transactions and that it might be substituted with non-legal norms, such as bargaining power (Macaulay 1963). Informal social controls are therefore seen as complements to formal controls and even sometimes supplant them (Black 1976; Ellickson 1994; Granovetter 1985). Parties use incomplete contracts and undetailed planning and avoid legal remedies because of their flexible nature (Beale and Dugdale 1975). Macneil conceptualized the relational approach, which he describes as a "mini society with a vast array of norms beyond those centered on the exchange and its immediate processes" (MacNeil 1978)—a different paradigm of contracts and contract law (Belley 1998). In his theory, Macneil assumes the existence of 10 common contractual norms—meaning ways parties must or should behave during the exchange—interpreted according to a discrete/relational spectrum: role integrity, reciprocity, planning, effectuation of consent, flexibility, solidarity, cohesive norms (restitution, reliance, and expectation), power, propriety of means, and harmonization with the social matrix (MacNeil 1980a). Of these norms, two are considered to be more discrete: implementation of planning and effectuation of consent, which in turn he renames as discreteness and presentation. He also singles out five particularly relational norms: role integrity, preservation of the relationship (derived from solidarity), harmonization of relational conflict (derived from harmonization with the social matrix), propriety of means, and supra-contractual norms, which refers to broader norms, such as distributive and procedural justice, liberty, human dignity, and social equality (MacNeil 1978, 1980a, 1983). Relational norms can be internal, internalized, or external (Macneil and Campbell 2001). Internal norms concern the contract terms and the development and evolution of obligations during the actual relationship, whereas external norms are found in the context and environment surrounding the relationship, such as in the regulatory framework for the procurement of work and professional services. Mandatory obligations derived from the regulatory framework and included in the contract are of an internalized nature (Jobidon et al. 2018).

A fully relational contractual arrangement has defining characteristics. Communications between parties are not limited to the formal linguistic form but instead are informal, deep, and extensive. Parties find satisfaction not only in the economic exchange, but also in the complex and personal social exchange. The nature of the exchange is difficult to monetize, and the commencement and termination of the relationship is gradual. The planning focus is mainly on the structures and processes of the relationship with limited specific planning of substance. Mutual planning is important, as parties are not likely to adhere to contractual terms without bargaining and will try to jointly develop such planning. Tacit assumptions are an aspect of relational planning and a precondition to a relationship's survival. Part of the planning might be binding, but a degree of tentativeness will remain with respect to some or all of the terms. Future cooperation regarding planning and performance is anticipated and considered necessary to a successful long-term arrangement. The benefits and burdens deriving from the exchange shall be shared and undivided. Sources of obligations are both external and internal to the relationship, which may also develop obligations itself. Obligations are usually nonspecific, non-measurable, and of a restorative nature. The number of parties to the arrangement will likely be more than two but will often include more stakeholders or parties, which are expected to behave in altruistic fashion. Instead of the presentation expected in a transactional exchange, relational arrangements futurize the present and anticipate troubles and conflicts as a normal feature of a long-term contract, which should be dealt with using cooperative and restorative measures (MacNeil 1974b, 1978).

Contract law has seen a paradigm shift from freedom to morality, transaction to relationship, fragmentation to integration, and from the closing to the opening of the system (Rolland 1998). Individualism and antagonism now leave room for collaboration and solidarity in dynamic, complex,

and ever-changing economies—a phenomenon partly created by division and specialization of labor (Durkheim 1893). According to Durkheim, social harmony results from the division of labor, which is "characterized by a cooperation which is automatically produced through the pursuit in each individual of his own interests. It suffices that each individual consecrate himself to a special function in order, by the force of events, to make himself solidary with others" (Durkheim 1893). More specifically, Durkheim refers to the notion of organic solidarity, which is a form of necessary trust based on the interdependence of highly specialized roles in a complex system of division of labor requiring the cooperation of all the groups and individuals in society (Durkheim 1893). Construction contracts are a prime example of the necessity of organic solidarity, as they are characterized by an ever-growing number of key stakeholders involved in the production process. These include architects, engineers, general contractors, construction managers, representatives from public entities, end users, and others who need to cooperate and trust each other during the long-term arrangement binding them.

### 2.1.4. Relational Contract Law in Different Fields

Even if the roots of relational contract theory are deeply embedded in sociolegal and sociological theory, it has been utilized in many fields, such as law, economics, governance, marketing, management, and, specifically, construction management. Each will be treated in this order.

Belley (1991, 2000) distinguishes his work from that of Macneil by inscribing it in the legal pluralism current, which proceeds from the assumption of a plurality of legal orders whose interaction, whether complementary or conflicting, would be manifested in contractual practices and contract representation. According to Belley, relational contract theory presents a very rich, and perhaps too rich, representation of the sociological reality to the detriment of the coherence of the legal analysis. Although he criticizes some aspects of Macneil's theory, he still mostly abides by it throughout his analysis of the role of law and legal institutions in the regional supply department. His study reveals the rather marginal role of state contract law and court law as planning instruments or as methods of conflict resolution and states that trust, flexibility, and the desire to preserve the commercial relationship are the prime factors to which parties agree in a compromise (Belley 1991). He also claims in his later work that problems confronting public and private law cannot find a judicial or administrative solution without the express consideration of standards that are proven, presumed, or deemed to be constitutive of the relevant social context (Belley 2011). Researchers have also tried to operationalize Macneil's theory. Campbell and Harris attempted to create a testable model of the cooperative attitudes and behavior of parties to a long-term contract. Their model could offer a potentially more interesting explanation than the individual utility-maximization model, therefore rejecting classical law when explaining long-term arrangements and endorsing relational contract law as the appropriate framework for their studies (Campbell and Harris 1993).

Goetz and Scott analyzed the issue in greater depth by examining contractual language usually found in relational agreements: a "best efforts" standard of performance and a discretionary termination privilege (Goetz and Scott 1981). In complex and uncertain economies, parties will more frequently rely on interactive and dynamic contractual arrangements because of the difficulty of circumscribing, analyzing, or identifying risks associated with contingencies, therefore complexifying the ex-ante negotiation process. The capacity of humans to respond to external complexity and uncertainty can be extremely costly and sometimes render the quest for presentation impossible for parties that are inherently rationally bound (Simon 1957). Despite these premises, Goetz and Scott advance that parties to a long-term contract are still seeking cost minimization, because the "best efforts" concept can be translated into an expected level of effort needed by the parties in order to maximize the net joint product resulting from the relationship, especially in a context where performance is not precisely specified (Goetz and Scott 1981). To achieve maximum output, parties should use appropriate monitoring mechanisms, such as a discretionary termination provision, which act as risk allocation mechanisms. The authors suggest that even if courts seem to understand the relationship between "best efforts" and termination, the legal doctrine is still uncertain and ambiguous. The reconciliation of

the two could facilitate the drafting of these types of mechanisms in complex and uncertain contexts while also giving the courts the opportunity to understand the mechanisms' inherent tension and the policies meant to regulate them (Goetz and Scott 1981).

Other researchers have investigated the role of court intervention, whether passive or active, in areas of contract law. Schwartz found four factors making courts inclined to activism (Schwartz 1992): (1) The contract formation process or the course of performance affects process values, such as asymmetric information and parties not being commercially sophisticated or facing monopoly power; (2) third parties are affected by the enforcement of the contract; (3) a substantially unfair outcome results from the contract; and (4) the incomplete contract is completed by the court with terms that parties will accept and the court will apply. Relational contracts often do not satisfy the first and fourth factors, because parties are usually commercially sophisticated and because the incomplete nature of the contract derives from missing, incomplete, or asymmetric information, and parties looking for legal enforcement will usually reject terms relying on unverifiable or unobservable information. The author suggests that contract theorists now explain contract content as the echoing parties' foresight of the possibility of renegotiation. Moreover, because parties are not able to predict all future risks associated with contingencies, they will create "structures" that will enable them to answer these contingencies in a desirable fashion. Schwartz concludes that regulatory caution is probably justified by the suggestion that relational contracts are efficient structures for the governance of commercial exchange and that judicial passivity may facilitate the parties' behavior in this context. He also notes that imminent regulatory attention to relational contracts might need further study (Schwartz 1992). The notion of "structures" is essential for the analysis of public construction projects and the content of professional services contracts. The actual behavior of parties not being part of the present study, our focus is placed on relational governance structures and mechanisms present in the contract language and serving as foundations for the development of the long-term relationship.

Researchers have suggested where future research regarding relational contract theory should be focused. Feinman suggests that future research should be concentrated towards further fragmenting analysis, such as the possibility of commercial construction contracting, rather than extending Macneil's general theory of contracts. Standardized contracts repeatedly used by recurrent or occasional parties of different sizes and sophistication, between which interactions in a variety of settings occur and difficulties surely arise, are considered by Macneil to be a premium choice for the application of relational contract theory (Feinman 2000). On the other hand, Gudel came to the realization that the limitation of legal techniques has led to a growing interest in alternative dispute resolution, mediation, and counseling, which are interested not only in extending a relationship's norms, but also in giving structure and order to a clash of power. He therefore invited future researchers to investigate how the law could be made a more effective instrument for reinforcing contract norms (Gudel 1998). Only a few scholars in the legal field have addressed these pleas for scholarly attention (Harries and Vincent-Jones 2001; Stipanowich 1998; Sweet 1997; Vincent-Jones and Harries 1998; Vincent-Jones 2012).

Stipanowich analyzed the transactional features of the procurement system and concluded that genuine systemic reform, from the transactional framework to a more collective, cooperative, and relational framework, is essential in an era of evolving delivery systems that is witnessing a shift in roles and relationships (Stipanowich 1998). The United Kingdom's competitive tendering system has been analyzed by researchers looking to evaluate the transactional or relational nature of the system. They found that, while the procurement process is prone to transactional tendencies, relational techniques, such as (1) framework agreements implying the awarding of contracts to the same provider over a number of tendering rounds, (2) the negotiation of details after the decision to award the tender, (3) the use of approved lists, and (4) the importance of reputation and past dealings of tendering parties, could be used to counterbalance these tendencies (Macneil and Campbell 2001; Vincent-Jones 2012). The lack of trust affecting the compulsory procurement system was found, among others reasons, to lead to contractual problems (Vincent-Jones and Harries 1998). Studies have also focused on the application of relational contract law in the context of housing management and

legal services and found that voluntary tendering, such as greater choice for public authorities and more flexible timeframes and selection criteria procedures, which facilitate cooperative relationships, presents more relational features than its compulsory counterpart (Harries and Vincent-Jones 2001; Macneil and Campbell 2001; Vincent-Jones and Harries 1998; Vincent-Jones 2012). Jobidon et al. analyzed the procurement legislation, regulations, and context of three jurisdictions using a comparative law approach with respect to relational contract law and found that the relational features of the procurement systems analyzed could be implemented within a regulatory framework to counterbalance their transactional aspects in order to facilitate the implementation of more collaborative delivery methods (Jobidon et al. 2018). Studies have thus mainly focused on the validity and pertinence of relational contract law in the context of long-term contractual relationships, as well as on the direction of future work in the field and the application of the theory to the procurement process. The fact that the actual content of construction and professional services contracts and the language used to form the parties' obligations have not been specifically studied within the field of law contributes to the pertinence of this paper.

Economics scholars have also paid attention to the concept of relational contract law and especially to the incompleteness of long-term contracts (MacNeil 1974a). In practice, contracts cannot foresee every possible contingency (Grossman and Hart 1986; Hart and Moore 1990; Hart 1995). Hart defines an incomplete contract as one that "has gaps, missing provisions, and ambiguities and has to be completed (by renegotiation or by the courts)" (Hart 1995). Hart researched this concept in the context of PPPs, privatization, and integration (Hart 2003). Hart confirms that long-term contracts are not only effectively incomplete in practice but that they should be incomplete. He developed a model of incomplete contracting to analyze PPPs, suggesting that the choice between PPPs and conventional provisions depends on whether it is easier to write contracts on service provisions rather than on building provisions (Hart 2003). The incompleteness of contracts has been attributed to the transaction costs associated with contracting for a given contingency, which are greater than the benefits of such formalization (Ayres and Gertner 1989; MacNeil 1978; Williamson 1985). Transaction costs include the negotiation process and fees, drafting costs, legal fees, and researching of a contingency's effects and probability, as well as the judicial fees associated with determining whether the contingency occurred (Ayres and Gertner 1989). The notion of incomplete contracts is therefore similar to the theory of the firm because of the difficulties arising in foreseeing the uncertain future and the parties' desire to minimize transaction costs (Hart 2003; Hart and Moore 2008), defined by Arrow as the costs of running the economic system (Arrow 1969).

Moreover, incomplete contracts are considered to be the most cost-effective form of governance for long-term contracts (Campbell and Harris 1993). Filling the gaps left in long-term contracts requires the use of flexibility and soft terms, which structure the exchange and its evolution, therefore enabling parties to minimize costs and save time in certain future disputes (Nystén-Haarala et al. 2010; Woolthuis et al. 2005). Of course, there are risks to the use of incomplete contracts and soft terms, such as opportunistic and narrow interpretation by parties in the event of a dispute, as well as parties not shifting their attitudes to more cooperative stances (Campbell and Harris 1993). Transaction costs would not be of the greatest importance if parties had the opportunity to find easy contracting alternatives, but because a breakup is more often than not inefficient, parties are often faced with the hold-up problem (Klein et al. 1978; Williamson 1985). Specific investments of the ex-ante relationship increase the potential surplus generated by the long-term exchange, and most of these investments will be lost when the relationship is broken. When negotiating the sharing of the ex post surplus, ex-ante investments have already been made and therefore do not affect the bargaining outcome, which could lead parties to rely on the wrong investment incentives. Because the sharing of the ex post surplus cannot be fixed ex ante, renegotiation of the initial contract will therefore inevitably occur when the exchange surplus becomes clear (Bös and Lülfesmann 1996). However, the classic hold-up problem has been developed in a private procurement or private buyer–private seller context (Hart and Moore 1988; Williamson 1985). In a public procurement setting, where public bodies seek to maximize social welfare

and private firms wish to maximize profit, authors claim that the suppliers' overall profit should be zero but not necessarily at every step of the procurement process. Negative economic profit should be earned in the procurement's innovation phase, meaning the design phase of a building process, whereas positive profit should be earned in the realization phase (Bös and Lülfesmann 1996; Rogerson 1992). Soft budget constraints for public entities are considered necessary to enable renegotiation when the private contractor is not willing to complete the project because of an underestimated ex-ante trade price (Bös and Lülfesmann 1996). A need for innovation—so important in the case of public procurement for ecofriendly buildings—should lead public entities to pay a price to enhance innovative practices (Rogerson 1992).

Bridging the gap between the fields of economics and governance is Williamson's work regarding transaction cost economics and the governance of contractual relations (Williamson 1979). He identifies the critical factors that need to be accounted for while describing contractual relations or the type of transaction: uncertainty, frequency (recurrence of the transaction), and idiosyncrasy (the degree to which durable transaction-specific investments are incurred). The classification of the type of transaction is important because, depending on the type, transactions should be matched with the appropriate governance structure. Williamson describes idiosyncratic services as those in which investments of transaction-specific human and physical capital are made and gives the example of trades where delivery for a specialized design is extended over a long period of time, such as some construction contracts (Williamson 1979). He, therefore, gives the example of the construction of a plant, which is considered to be an occasional, idiosyncratic transaction. Williamson then identifies three broad types of governance structures: non-transaction-specific, semi-specific, and highly specific. The prime example of a non-transaction-specific governance structure is the market, an impersonal structure for the exchange of standardized goods at an equilibrium price (Ben-Porath 1980), whereas highly specific structures are bespoke to the transaction's special exigencies. Williamson then relates these governance structures with Macneil's distinction between classical, neoclassical, and relational contract law, adding that relational contracting is best-suited for recurrent and non-standardized transactions, whereas neoclassical contracting (or trilateral governance) is adapted for occasional, non-standardized transactions. This implies that public contracts for construction projects and professional services should, according to Williamson, be governed by a neoclassical structure. While recent approaches to governance applied to the broader field of global supply chains have nuanced Williamson's three main governance structures to include a four-way classification of governance structures (Gereffi 2014; Locke 2013; Salminen 2017), the more relevant field of procurement of complex performance has let go of Williamson's neoclassical governance structure by continuously trying to oppose classical formal and relational governance, leaving the neoclassical intermediary relegated to an afterthought.

Governance research has recently addressed the question of procurement of complex performance as it pertains to relational contract theory (Caldwell and Howard 2010; Lewis and Roehrich 2009). Complexity is considered to be a mesh of different factors, covering the number of project stakeholders, the length of the planning negotiations, and the construction phase, as well as the presence of bespoke infrastructural components. It can be applied to IPD projects, which deal with organizational complexity in terms of horizontal differentiation, personal specialization, and reciprocal interdependencies, as well as with technological complexity, such as task interdependency created, for instance, by the utilization of BIM (Baccarini 1996; Mohr 1971; Roehrich and Lewis 2014). Other forms of complexity specifically affect the procurement process for construction projects and professional services, mainly involving ex-ante transactional complexity, such as design and service specifications, as well as infrastructural complexity, such as financial and organizational structures (Zheng et al. 2008). As discussed above, contractual and relational structures are the two main governance structures that have been proposed in the literature in recent years. While it seems odd to oppose the two usually intricate notions, governance literature diverges from law inasmuch as contractual governance loosely relates to discrete transactions and the predominant role of formal contracts to safeguard parties from opportunistic behavior (Cao and Lumineau 2015). Relational governance more specifically relates to trust as an

alternative to formal contracts and to presentation to mitigate uncertain and transaction-specific investments (Cao and Lumineau 2015; Heide and John 1992; MacNeil 1980a).

As is the case with many concepts, striking a balance between seemingly opposite poles seems to be the preferred option. The literature states that contractual and relational governance should be considered complementary mechanisms (Cao and Lumineau 2015; Ferguson et al. 2005; Poppo and Zenger 2002). In complex procurement arrangements, an increase in relational exchange governance should help contractual exchange governance (Roehrich and Lewis 2014) mainly by improving relationship performance, especially in a high uncertainty contractual relationship (Cannon et al. 2000). Researchers have identified a correspondence principle between the complexity of projects and that of governance structures: the more complex the project, the more complex the governance structures, decision processes, and strategies should be (Boisot and Child 1999; Eisenhardt et al. 2000). The importance of trust in the procurement process is repeatedly emphasized in the governance field. Fostering interpersonal and inter-organizational trust through the iterative processes of bargaining, commitment, and joint activities between partners helps to establish feedback channels and develop team familiarity, which in turn can lead to increased project performance (Ring and Van de Ven 1994; Roehrich and Lewis 2014). Establishing trusting relationships during the more sensitive early stage of the procurement process was also found to help with the development of the relationship. This also helps parties resolve contractual issues arising in later project phases due to the incompleteness of long-term arrangements in a more flexible manner (Roehrich and Lewis 2014). IPD projects are often tinged with complexity, whether organizational or technological, and are inherently uncertain because of the long-term nature of the arrangement and the myriad possible design outcomes. There is, therefore, a need to identify specific relational content or mechanisms that can complement the formal contractual public procurement process—one of the aims of the present paper.

The field of marketing has also contributed to relational contract theory by analyzing the applicability of Macneil's norms in the context of a commercial exchange. The norms of solidarity, mutuality (reciprocity), flexibility, role integrity, restraint of power, conflict resolution (harmonization of conflict), and relationship focus (implementation of planning and effectuation of consent) were found to be operationalizable to analyze buyer–seller relationships (Harper 2014; Kaufmann and Dant 1992). Reciprocity, role integrity, and solidarity were also found to have an effect on the perception of unfair treatment by a party, which also affects the ex post retained level of hostility, thereby relating solidarity to the level of perceived fairness (Harper 2014; Kaufmann and Stern 1988). Other researchers have also confirmed the pertinence of using a transactional/relational continuum and have empirically observed that transactional and relational types of exchange can coexist at the supply chain level (Lefaix-Durand and Kozak 2009). Furthermore, it was found that healthy business relationships are considered to be a major source of value creation and that a link between value creation in relationships and competitiveness exists (Lefaix-Durand 2008).

Management scholars have also shown interest in Macneil's relational contract law and application, especially in the concept of trust, or solidarity. Sake and Helper carried out a comparative study regarding the determinants of inter-organizational trust in supplier relations in Japan and the United States (US). They found significant differences between each country's perception of written contracts. Contracts were considered to be irrelevant governance mechanisms in Japan, whereas the use of formal agreements and the length of past trading were correlated with greater opportunism in the US. The researchers also identified conditions facilitating the creation and sustenance of trust, such as long-term commitment, information exchange, technical assistance, and customer reputation (Sako and Helper 1998). In earlier work, Sako also found that trust reduces transaction costs in relation to bargaining and monitoring, thus enhancing performance (Sako 1991). Jeffries and Reed explored the effects of interaction between organizational and interpersonal trust on negotiators' motivation to problem solve in a relational contract environment. Trust has almost always been perceived as good and as having positive effects on performance, but the authors propose a counter-intuitive conclusion by stating that there is a downside to trust, mainly in the form of reduced motivation for negotiators. They, therefore,

demonstrate Granovetter's paradox of trust: while trust facilitates the emergence of norms, such as expectations of proper behavior, it also reduces risk perception and enables abuse through opportunism (Dyer and Singh 1998; Granovetter 1985). However, Jeffries and Reed suggest that when organizations enter relational contracts in a low trust environment, negotiators should not change over a sufficiently long period of time to help develop affect-based trust (Jeffries and Reed 2000).

Blomqvist et al. analyzed the roles of trust and contracts in the context of an asymmetric research and development (R/D) collaboration. First, they reiterate the quasi-impossibility and futility of trying to presentiate a long-term arrangement embedded in uncertainty by stating that a successful collaborative exchange cannot be guaranteed by a detailed contract. However, they do affirm that the contract and the contracting process have the potential to increase mutual understanding and learning, thereby building trust (Blomqvist et al. 2005). This relates to the previously addressed vision of trust by governance scholars, who state that trust is a cyclical process of recurrent bargaining commitment and execution of events between partners—a notion somewhat similar to Zucker's process-based or relational trust (Zucker 1985). While earlier scholars thought that trust was either present or absent (Hirshleifer 1983; Raiffa 1957), Zucker confirmed that there exist trust-producing mechanisms whether from individuals, firms, or industries (Zucker 1985). In the present paper, we adhere to this interpretation of trust-producing mechanisms and identify those mechanisms associated with different procurement delivery methods.

More specifically, the field of construction management is related to public construction contracts. Alsagoff and McDermott believe the industry's fascination with relational contracting, long-term partnering, and collaborative contracting stems from "Japan manufacturing"-style contracting, notably the works of Asanuma (Asanuma 1988), Ikeda (Ikeda 1987), and Morris and Imrie (Morris and Imrie 1993), which see relational contracting as a solution offering the benefits of vertical integration while still maintaining flexibility in competitive markets (Alsagoff and McDermott 1994). The authors also note that major institutional research wishes to encourage healthy long-term relationships in the United Kingdom (UK)'s construction industry through the use of joint ventures and partnering (Latham 1994). The famous Latham Report conveys relational attitudes, ideas, and principles, such as the fact that (1) good relationships based on mutual trust benefit clients, (2) the use of contractual documents should place the emphasis on teamwork and partnership to solve problems, and (3) there is a need for a general duty to trade fairly (Latham 1994). Following the Latham Report, scholars have paid further attention to the concept of relational contracting in the field of construction. Cheung et al. discussed the application of relational contracts in construction by examining the relational level of construction contracts using a relational index based on eight factors: (1) cooperation, (2) organizational culture, (3) risk, (4) trust, (5) good faith, (6) flexibility, (7) the use of alternative dispute resolution, and (8) contract duration. Although their analysis was restricted to the context of the traditional design–bid–build (DBB) method, it was found that the main contract and domestic subcontract forms are more relational than those of the nominated subcontracts and the direct labor contract. The authors also note that a major prerequisite for fostering cooperation relies on maintaining a good relationship between the client and the main contractor. Because the client is the main source of work and revenue, the maintenance of a healthy long-term relationship is key for the survival of a main contractor. More importantly, the authors found that the concepts of relational contracting may not be applicable to every delivery method type, such as DBB. According to them, different contract types foster different characteristics, whether transactional or relational, and some delivery methods might be ill-suited to the use of relational contracting principles (Cheung et al. 2006). This last point is of great importance to the theme of this paper, which will demonstrate the gap in relational content in the contractual language presented by different project delivery methods.

In a series of papers regarding relational contracting and the necessity of relationally integrated teams, Rahman and Kumaraswamy reiterate the need for appropriate contracting methods and documents coupled with the attitude of contracting parties and their cooperative relationship to ensure successful project delivery (Rahman and Kumaraswamy 2002). They underline the necessity of joint

and dynamic risk management and flexible contract conditions, allowing for quick adjustment by the parties to the inevitable problems that will arise and highlighting the advantages of joint risk management (Rahman and Kumaraswamy 2002). In a subsequent paper regarding the need for relational integration, these researchers found four factors facilitating the team-building process: (1) the client's competencies and overall learning/training policy; (2) previous interactions, performance, competencies, and specific input and outputs of various partners; (3) compatible organizational culture, longer-term focus, and emphasis on trust building; and (4) improved selection of project partners and better delegation of responsibilities (Kumaraswamy et al. 2005). The factors deterring the team-building process are divided into five major components: (1) lack of trust, open communication, and uneven commitment; (2) commercial pressure, absent or unfair risk/reward planning, and incompatible personalities or organizational cultures; (3) lack of general top management commitment and client's knowledge/initiative; (4) lack of good relationships among the team players; and (5) exclusion of some team players in risk/reward plans, errors, and cultural inertia (Kumaraswamy et al. 2005). Although these components go beyond the frame of the present study, many of them can be related to contractual language and content, such as the presence or absence of trust-producing mechanisms (Zucker 1985) and the risk sharing or risk allocation mechanisms laid out by the contract. In further research, Rahman and Kumaraswamy highlight the need for reviewing contractual conditions to accommodate relational contracting principles (Bayliss 2002; Motiar Rahman and Kumaraswamy 2005), such as using contractual incentives to persuade contractors to adopt relational contracting approaches (Ling et al. 2006). Kumaraswamy et al. also note that relational contracting principles are more easily applied to private sector projects than public ones (Kumaraswamy et al. 2010; Ling et al. 2013a). They also point out that it is still not well known if public projects can fully benefit from relational contracting principles (Dulaimi et al. 2007), because of the necessity of public clients to use the competitive bidding process, which does not put them in a position to offer incentives for future relationships. The mistrust of public clients developing business relationships with private providers—a fear embedded in the state's desire to reduce corruption in construction—has also affected institutional trust and relational development in Quebec (Jobidon et al. 2018). A recent study conducted by Ling et al investigated relational transactions and overall relationship quality by using common contractual norms of role integrity, flexibility, solidarity, propriety of means, and harmonization with the social matrix (Ling et al. 2013b). Relationship quality between project partners can be predicted according to which relational practices are implemented, such as the adoption of flexible strategies, trust among team members, and the sharing of project information, just to name a few. Significant correlations between relationship quality and its impact on time performance and client satisfaction have been found, as is the case with high propriety of means, better cost performance, and client satisfaction (Ling et al. 2013b).

One of the most recent studies focused on the construction of multiple-statement scales to measure relational norms—mainly the norms of role integrity, reciprocity, flexibility, propriety of means, reliance and expectations, restraint of power, contractual solidarity, and harmonization of conflict—in the context of construction projects (Harper et al. 2016; Harper 2014). The study essentially operationalized relational contracting norms to measure project integration in different construction contracts, although it mixed common contractual norms while leaving some behind, such as planning implementation with relational norms (harmonization of conflict). In his doctoral thesis, Harper found correlations between contractual norms and project success, while also concluding that IPD contracts present more relationally integrated teams than those of design–build, construction manager at risk, and design–bid–build delivery methods (Harper 2014), thus confirming the previously stated fact by Matthews and Howell that IPD is an example of relational contracting (Matthews and Howell 2005). Although Harper analyzed the contractual language of different delivery methods, he did so by using occurrences, without specifying the main differences in the relational mechanisms provided for by different contracts or looking into the nature of the obligational content—elements that will be at the core of the present research.

This literature review of relational contract law, relational contracting principles, and their applications in different fields identifies the main gaps present in the literature, particularly pertaining to the application of relational contracts in the field of public contracts for construction projects and professional services in the context of vertical construction. Table 1 below illustrates the key takeaways from this literature review.

**Table 1.** Key takeaways from the literature review.

| | |
|---|---|
| Classical contract law | • Classical contract law is characterized by the predominance of discreteness, presentation, legal rules, formal documents, and self-liquidating transactions (MacNeil 1978; Williamson 1979).<br>• Discrete transactions are characterized by little or no social exchange (Blau 1964), little personal involvement of stakeholders, and no consideration of reputation or future relations (Campbell and Harris 1993; MacNeil 1978). |
| Neoclassical contract law | • Neoclassical contract law is characterized by flexibility mechanisms and good faith (MacNeil 1978).<br>• External standards, third-parties, one-party control of terms, cost, agreements to agree, redetermination, and renegotiation are considered flexible mechanisms (Crocker and Masten 1991; Crocker and Reynolds 1993; MacNeil 1978; Nystén-Haarala et al. 2010).<br>• Its resemblance to classical contract law can make it inappropriate to deal with relational issues (MacNeil 1978). |
| Relational contract law | • Relational contract law arises from the increase in contract complexity and the necessity of sustaining long-term relationships (MacNeil 1978; Williamson 1979).<br>• It includes Macneil's 10 common contractual norms (MacNeil 1978, 1980b, 1983).<br>• Norms are present at three levels: internal, internalized, and external (Macneil and Campbell 2001).<br>• The defining characteristics of a relational contract are as follows: importance of mutual planning, degree of tentativeness in planning, anticipated and necessary future cooperation, futurizing of the present, and conflict resolution through cooperative and restorative measures (MacNeil 1974b, 1978, 1984). |
| Law | • Trust, flexibility, and preservation of the relationship are prime factors of a relational contract (Belley 1991).<br>• The endorsement of relational contract law is used as a framework for the study of long-term arrangements (Campbell and Harris 1993).<br>• The "best efforts" standard of performance and discretionary termination privilege are examples of contractual language found in relational contracts (Goetz and Scott 1981).<br>• Structures are necessary to prevent future risks (Schwartz 1992).<br>• Future research regarding relational contract theory could concern commercial construction contracting and standardized contracts (Feinman 2000).<br>• Relational techniques can counterbalance transactional tendencies in public procurement (Jobidon et al. 2018; Stipanowich 1998).<br>• Lack of trust leads to contractual problems in public procurement (Vincent-Jones and Harries 1998). |
| Economics | • It is characterized by the importance of incompleteness for the governance of long-term contracts (Hart 2003; MacNeil 1974a)<br>• Public clients should pay a price to enhance innovative practices (Rogerson 1992). |
| Governance | • Transactions should be matched with the appropriate governance structure (Williamson 1979).<br>• Relational contracting is best-suited to recurrent and non-standardized transactions (Williamson 1979).<br>• For the procurement of complex performance, contractual and relational governance are complementary mechanisms (Cao and Lumineau 2015; Ferguson et al. 2005; Poppo and Zenger 2002). |

**Table 1.** *Cont*.

| | |
|---|---|
| Governance | • An increase of relational exchange governance in a complex procurement context should help contractual exchange governance (Roehrich and Lewis 2014).<br>• The importance of trust in the procurement process is emphasized (Ring and Van de Ven 1994; Roehrich and Lewis 2014). |
| Marketing | • Macneil's 10 common contractual norms are operationalizable and applicable in a commercial exchange context (Harper 2014; Kaufmann and Dant 1992).<br>• The pertinence of using a transactional/relational continuum is highlighted (Lefaix-Durand and Kozak 2009). |
| Management | • Trust reduces transaction costs (Sako 1991).<br>• It is futile to presentiate long-term arrangements (Blomqvist et al. 2005).<br>• The existence of trust-producing mechanisms is emphasized (Zucker 1985). |
| Construction management | • Contracts should emphasize teamwork and partnership (Latham 1994).<br>• Concepts of relational contracting may not be applicable to every delivery method (Cheung et al. 2006).<br>• Appropriate contracting methods to ensure successful project delivery are necessary (Rahman and Kumaraswamy 2002).<br>• It is still unclear if public projects can benefit from relational contracting (Dulaimi et al. 2007).<br>• Relationship quality between project stakeholders can be predicted according to the relational practices implemented (Ling et al. 2013b).<br>• A correlation exists between the presence of Macneil's common contractual norms and project success (Harper 2014). |
| Gaps in the literature | • The actual content of construction and professional services contracts has not been studied.<br>• There is a need to identify specific relational content or mechanisms to complement the formal contractual public procurement process.<br>• The gap in relational content between different project delivery methods is not well understood. |

Most of the studies cited focus on the validity and pertinence of relational contract law in the context of long-term relationships, the need for general relational contracting principles and their actual application during a project, the parties' behavior in collaborative environments, and the need for both formal and informal processes to achieve project success. MacNeil emphasizes the element of planning in his work, and he assumes that relational contract law is useful for contract planning, which relates to the content of the contract and to the processes followed in the resulting relationship (Diathesopoulos 2010). The present study will therefore shed light on the material differences between different public project delivery methods by analyzing the contractual language presented and the mechanisms provided for by the different contractual structures available to public clients in Quebec. In this paper, we will also identify what makes the content of a contract more or less relational and how different specific contractual mechanisms make a contract more or less relational, thus illustrating how a contract can serve as the foundation for the development of a healthy long-term relationship. The implementation of a relatively new contractual structure, such as IPD, which is tinged with organizational and technological complexity, creates the need for a better understanding of the different processes at hand for public entities wishing to enter long-term contractual relationships for the construction of vertical infrastructure.

### 2.2. Project Delivery Methods: From DBB to IPD

Project delivery methods refer to the different roles of key stakeholders of a construction project, the formal contracts between those stakeholders prescribing the sharing or allocation of risk and

rewards, and the management practices used to complete the project (Appelbaum 2012). Project delivery methods have undergone a significant paradigm shift since the Latham Report, with the industry choosing to slowly forsake the traditional DBB method for alternate delivery methods. Although there is a wide range of delivery methods, the present paper focuses on those being used by Quebec's major public clients: DBB, DB (called "turnkey" by the public client), and CMGC in combination with IPD, although some precision about the status of the latter will be addressed in Section 2.2.4.

### 2.2.1. Traditional (DBB)

The traditional, and still the most used, project delivery method for major projects in Quebec is the sequential design–bid–build method, in which the project is fully designed by procured professionals and then transferred to a contractor who builds the infrastructure. In this method, three parties are tasked to complete the project, with formal arrangements existing between the owner and design professionals, engineers, and architects, as well as between the owner and the contractor. Even if the parties subcontract part of the work, they remain contractually responsible (Pierce et al. 2003). The risks associated with the use of DBB include the owner assuming liability for insufficient detail and defects in the plans and specifications and the possibility of amendments arising from said design inadequacies leading to cost increases and schedule delays. The contractor is responsible for defects relating to the construction of the infrastructure, whereas the design professionals are responsible for the design but not any subsequent defective construction. As with every other project delivery method, DBB has benefits and drawbacks. As to its benefits, DBB is considered to be a well-known and commonly implemented method with established legal precedent and full completeness of design before construction. Another benefit is its corresponding cost estimation, with the owner retaining a high level of control (Harper 2014) over the design, selected materials, and documentation (Pierce et al. 2003). Relying on single-point responsibility simplifies risk allocation—a notion that is complemented by the use of standardized contracts, with which the parties are usually familiar. To these benefits can be added the intensive participation of the public client in the development of the project, coordination between the design and maintenance of long-term assets, and a hands-on approach by the client regarding the management of changes (Infrastructure Québec 2013).

The drawbacks associated with DBB are the following: (1) there is no formal arrangement between the designer and the contractor; (2) a competitive atmosphere exists, resulting in low trust and collaboration levels between the participating entities; (3) risk is allocated instead of shared; and (4) the decision-making processes and project goals are opportunistic and individualistic, which often lead to litigation regarding the source of responsibility (Harper 2014). The cost also remains uncertain early in the process, with probable project overruns occurring. Fast-track building is also impossible, because the design must be fully complete before the contractor tendering process is launched. Public clients must ensure coordination between the design professionals and the general contractor and are responsible for the project schedule and quality (Infrastructure Québec 2013). Scholars and practitioners have demonstrated the limitations of DBB, which lead to increases in costs, schedule delays, and poor quality. These limitations include adversarial relationships, low productivity rates, and a lack of innovation (Egan 2002; Latham 1994; Lichtig 2006).

### 2.2.2. Construction Manager/General Contractor

The drawbacks of DBB have led the industry to opt for alternative delivery methods, such as the use of a construction manager with or without risk, which has gained popularity over the last four decades. The construction manager is responsible for managing the entire construction process, specifically to aid in providing constructive input during the design process. The construction manager delivery method has two possible versions: (1) the construction manager (CM) acts only as an agent (Pierce et al. 2003), meaning without being at risk and with a status somewhat similar to that of

a consultant or a service provider (CM-as-agent) (Jobidon et al. 2018), or (2) the CM is the actual constructor of the project, which is based on a design to which the CM contributed advice (CMAR).

In the CM-as-agent method, the construction manager is an independent advisor to the public client and helps with the decision-making process regarding the selection of contractors and professional services providers but is not in a direct contractual arrangement with those parties—a role kept by the public client. The CM does not perform the construction work but administers the contracts and the paperwork. Thus, the CM-as-agent method is not a true delivery method and is more similar to a management process (Pierce et al. 2003) in which the CM oversees the schedule and project costs and has a reasonable care duty, but the public client still carries the risks of the project. The recognized advantage of this management process is the public clients' access to construction expertise, thereby fostering the selection of adequate and high-performance materials, products, and systems; more knowledgeable budget control; the precision and reduction of design errors; more efficient schedule control; value enhancement; the ability to use fast-track building; and the creation of a collaborative atmosphere between the key parties in the project (Harper 2014; Pierce et al. 2003).

In the CMAR variant, the CM is responsible for the previously stated management processes while maintaining direct responsibility for carrying out the work. The CM is part of the design team and offers advice to design professionals regarding efficient means and methods, materials, and overall constructability to reduce construction costs. Scholars have found that this can lead to fewer requests for information and change orders in the highway procurement context (Shane and Gransberg 2010). The CM acts as the general contractor and holds the trade contracts, with the risks associated with deficient performance, delays, and cost overruns. The use of a stipulated sum or a guaranteed maximum price (GMP) represents the remuneration form of the construction obligation of the CM. While the CMAR method maintains the previously stated benefits of the CM-as-agent method, it also enables single-source responsibility, a ceiling on costs using a GMP, and traditional risk protection coverage (Pierce et al. 2003). The public client still maintains intensive participation in the development of the project and control over the change order process (Infrastructure Québec 2013).

The main drawbacks associated with the use of construction management are the multi-source responsibility in the CM-as-agent scenario, which can lead to adversarial relationships and litigation claims; lack of early price confirmation, even if the risk is reduced using a GMP; multiplication of contracts and documentation due to the sequential and lot bidding process; limited applicable law and precedent; and less familiarity with the process (Harper 2014; Infrastructure Québec 2013; Pierce et al. 2003). In Quebec, CMs are not chosen through qualification-based selection but rather through a mixed price/quality mechanism (Jobidon et al. 2018). The relevant contracts analyzed in this paper use the CMAR method.

### 2.2.3. Design–Build (Turnkey)

The other alternative delivery method used in Quebec is design–build (DB). The official institutional appellation for this delivery method is "turnkey", because public clients have the option to choose between classical DB or DB including financing (DBF). To avoid confusion, and because the contracts analyzed here relate to the former option, the term DB will be used throughout the paper. DB strays away from the classic three-party arrangements, because there is only one formal arrangement between the public client and the design–builder, who is responsible for the design and construction of the infrastructure. Almost every risk is passed from the public client to the design–builder, who is therefore usually responsible for land assembly, design, construction, supervision of construction, and the commissioning of the infrastructure (Pierce et al. 2003). Risks, such as possible defects, deficiencies, and additional costs, are transferred to the design–builder, while the public client bears the uncertainty regarding the design and construction, thus driving the need for precautions and for the public client to maintain an active role in control and supervision to ensure that it receives true value. The main benefits of using DB include (1) the reduction of the project duration; (2) single-source responsibility, which reduces claims and litigation between the parties regarding design or defective work; (3) the

reduction of constructability issues and the creation of a collaborative and innovative atmosphere through the involvement of the builder during design and development; (4) early cost confirmation and schedule determination; (5) reduced administration and coordination burdens; and (6) cost savings in terms of professional fees (Harper 2014; Infrastructure Québec 2013; Pierce et al. 2003).

Having a single entity responsible for the design and construction of the infrastructure generates different problems, potential or actual. The planning complexity, which needs to be extremely detailed using performance-based specifications because of the unavailability of plans and specifications early in the procurement process, creates a burden for public clients who need to mitigate risk through considerable care and diligence. This is achieved by addressing every component or system critical to the infrastructure success or by reserving approval rights for design and construction (Pierce et al. 2003). There is also potential for the maximization of individual outcomes instead of project outcomes, because the designer is accountable to the contractor and not the public client and the constructor may seek to reduce construction costs and quality, in terms of life cycle and maintainability, to increase profitability (Harper 2014; Infrastructure Québec 2013; Pierce et al. 2003). The public client also has less control over the design phase, with the effect of leaving greater flexibility and innovation potential for the design–builder. Finally, there is relative inexperience in the administrative process with respect to the public client and the design–builders, limited applicable law and precedent, fewer such contractors available, and limited experiences with DB in general (Harper 2014; Pierce et al. 2003).

### 2.2.4. Integrated Project Delivery

To mitigate the drawbacks associated with traditional delivery approaches and further the collaborative ambiance of alternative delivery methods, the industry has recently turned to IPD. Typically, IPD involves a multi-party agreement between the client, the design professional, the contractors, and the other key stakeholders, such as a facilitator or end users. The advantage of having a multi-party contract resides in jointly created goals, as well as the sharing of benefits and burdens. Even though having a multi-party agreement does not guarantee a cooperative and collaborative process, as is the case with other contract types or delivery methods, it should help develop and sustain a healthy relationship, which in turn can lead to better overall project outcomes. Recent research found that owner expectations regarding budgets, schedules, design quality, and sustainability were met or exceeded when using the IPD method (Cohen 2010). IPD projects were found to attain a higher level of quality, a shorter completion time, and a lower number of change orders in comparison with non-IPD projects, without significant increases in project costs (El Asmar et al. 2013). Other researchers have also found that the IPD method enables higher trust and collaboration levels and a streamlined process (Sive 2009). The main drivers for choosing the IPD method include market advantage, cost and schedule predictability, risk management, and technical complexity (Cheng et al. 2011).

To say that IPD automatically creates a highly trustworthy environment might be a stretch, but it definitely has the potential to assemble a non-adversarial project team from different organizations (Jobidon et al. 2018; Lahdenperä 2012). The jointly developed goals seek to more efficiently achieve innovative functional, environmental, and economic objectives (Jobidon et al. 2018). To do so, IPD requires the early involvement of key stakeholders and ongoing collaboration between them, along with multiple iterations leading to innovation; collaborative decision-making driven by performance objectives; continuous value management; the sharing of risks and rewards among team members in conjunction with reduced liability for the participants; open communication; quality-assurance processes throughout the realization of the project; and post-occupation assessments of the infrastructure (CERACQ and GRIDD 2015). The relationships between key stakeholders—the informal part of the contractual process—are of prime importance for IPD (El Asmar et al. 2013; Rahman and Kumaraswamy 2004). These relationships and the processes underlying them do not appear out of thin air, especially not in the formalistic world of public contracts, which reinforces the need to analyze the formal processes laid out by the contract to help establish the basis for the development of such relationships.

IPD, in its essence, aims to generate integrated, optimal, innovative, and sustainable solutions, with particular regard to location impacts, energy efficiency, minimal consumption of non-renewable resources, water, and the quality of the indoor environment (Société Québécoise des Infrastructures 2016a). These publicly stated objectives are in harmony with a common theme proposed by scholars to achieve sustainable construction, which notably consists of the reduction, minimization, or prudent consumption of resources such as energy (Akadiri et al. 2012; Cole and Larsson 1999; DETR 2000; Halliday 2008; Hill and Bowen 1997; Kibert 2016; Miyatake 1996). To be effective, these objectives should be supported by process-oriented principles, such as (1) decision-making processes involving key stakeholders in a timely fashion (Brundtland 1987; Lam et al. 2010); (2) interdisciplinary and multi-stakeholder relations (McMichael et al. 2003; Robinson 2004; Sebastian 2011); (3) the use of a system's approach, such as an integrated design process; and (4) the development of energy-efficient technological processes (Akadiri et al. 2012). The relationships between the different stakeholders and organizations should rely on open coordination and communication to be truly efficient (Carter and Carter 1998; Powell et al. 2006; Van der Grijp and Russell 1998; Varnäs et al. 2009) while also being based on mutual commitment and understanding (Diabat and Govindan 2011; Salam 2008; Wong et al. 2016). Technological processes, such as advanced construction information technology (i.e., BIM), stimulate the construction process to consider and incorporate more environmentally friendly approaches and lead to waste reduction (Lam et al. 2010; Wong et al. 2016). The interplay between the parties to a construction contract, and especially the role of designers and their interactions with the other parties in the supply chain, is considered to be a huge facilitator of green infrastructure construction (Annunziata et al. 2016; Humphreys et al. 2003; Love et al. 2004; Wong et al. 2016). This is also the case with the early selection and involvement of experienced team members, which enables the alignment of individual goals with team goals (Lapinski et al. 2006).

Many actors in the construction industry believe that the solution to environmental and sustainable construction lies in legislation, a view that could lead to increased bureaucracy. This, coupled with insufficient collaboration and cooperation between parties in the building process, is the main reason for the inertia of the industry (Ruparathna and Hewage 2015; Varnäs et al. 2009), (Gluch et al. 2007). This inertia has been shown to exist in the Canadian industry, where only a few sustainable initiatives have been proposed with respect to project procurement, but many more have been proposed with respect to environmental sustainability criteria (Ruparathna and Hewage 2015). To better satisfy these environmental criteria, scholars have suggested emphasizing their presence during the preliminary design competition in the tender for construction contracts and building services (Clement and Erdmenger 2003; Varnäs et al. 2009). However, the main solution, according to industry practitioners and scholars, seems to lie in the use of LEED (Ruparathna and Hewage 2015; Xia et al. 2014). There is a desire for clearly stipulated and measurable requirements (Sterner 2002)—a somewhat formalistic approach to sustainable construction. Green specifications in contracts are considered to have greater business sense than other measures, because contracts are more inherently flexible and therefore represent a predominant management tool for sustainability in the construction process (Lam et al. 2010). Formal and informal governance through contracts, whether transactional or relational, have been found to influence the value-creation initiatives of parties through the reduction of transaction costs or the creation of incentives for such initiatives (Dyer and Singh 1998; Ring and Van de Ven 1994).

As this paper will further underline, Quebec's use of IPD is "IPD-ish" in nature, meaning that it does not have some of the key characteristics of IPD (Azhar et al. 2014), such as not using a multi-party agreement (El Asmar et al. 2013). IPD is therefore not an independent or autonomous delivery method but is used in conjunction with CMGC. It is implemented as an approach rather than a genuine delivery method. Although sustainability initiatives are usually analyzed at the macro level, there seems to be a lack of strategies at the project level (Akadiri et al. 2012). Solutions, such as incorporating formal stipulations in the contract, the need to utilize reusable material, or adhering to specifications arising from an external institution, such as LEED, seem to bring consensus within the industry. The present

paper will illustrate the different path provided for by CMGC/IPD to achieve environmental and sustainable objectives. Rather than conforming to external standards, CMGC/IPD seeks to achieve those objectives through informal and relational governance mechanisms.

IPD is considered to be the most relational delivery method of the previously presented methods, mainly because of the alignment of project objectives with the interests of key participants (Harper 2014; Matthews and Howell 2005). However, as seen with the other delivery methods, drawbacks to IPD do exist. These include the difficulty of finding adequate and appropriate insurance for a multi-party agreement (Post 2010) and the necessity of including waiver claims, which might be illegal in some jurisdictions (Harper 2014). The IPD's novel aspect can also create problems, because management and participating entities might not have enough pertinent prior experiences with the delivery method, which requires a radical change of culture in comparison with the traditional methods.

This paper will therefore identify, according to relational contract law, the qualitative differences between the main project delivery methods and approaches used by Quebec's public clients and the principal differences between CMGC/IPD and theoretically pure IPD.

## 3. Methodology

In the present paper, we seek to analyze the formal mechanisms of construction and professional services contracts according to relational contract theory in order to compare the relational aspects present in the contractual language of Quebec's three project delivery methods for public infrastructure: DBB, DB, and CMGC/IPD. Macneil looked at the written parts of contractual relations as constitutions establishing legislative and administrative processes for the relationship (Cox 1958; Fuller 1963; MacNeil 1978; Shulman 1955)—a view adopted by the authors of the present paper. Table 2 presents concise definitions of Macneil's common contractual norms and the transactional/relational poles of each norm, as adapted from Harper (Harper 2014). The formal written parts analyzed consist of 26 contracts (architecture, civil and structural engineering, construction, electrical/mechanical engineering, construction management, commissioning, soil and materials, and geotechnical studies) and 139 related contractual documents (addenda, mandates, annexes, preliminary studies, technical requirements, functional and technical requirements, specifications, directives to tenderers, and construction programs), as well as the SQI's eight standardized contracts and tender documents. Those documents stem from seven different projects (two DBB, two DB, and three CMGC/IPD). It is worth noting that the SQI is one of the 10 largest public clients in Canada (Jobidon et al. 2018).

Law, which previously claimed methodological purity in the form of mono-disciplinarity, is now leaning towards interdisciplinarity, and results are derived from contextualized experiences rather than a priori axioms (Ost and Van de Kerchove 2002). The formal written parts of the documents were thus analyzed using grounded theory—a social science method aimed at creating and making plausible categories, hypotheses, or properties—without trying to test them (Strauss and Corbin 1990).

**Table 2.** Macneil's common contractual norms: succinct definitions and poles.

| Norm | Definition | Transactional Pole | Relational Pole |
|------|-----------|-------------------|-----------------|
| Solidarity | Solidarity relates to the mindset of the stakeholders, as they believe not only in future peace among stakeholders, but also in positive future harmonious cooperation. It is a concept of trust. | Low level of trust; lack of inter-organizational support; a competitive and conflict-ridden atmosphere. | High level of trust and inter-organizational support; a cooperative and collaborative atmosphere. |
| Cohesive norms | Norms that aim to maintain the exchange and cohesion between stakeholders. | Broken promises; unclear expectations; no option to depend on other organizations; no rebalancing of gains. | Promises kept and completed; ability to depend on each other; rebalancing of gains. |

**Table 2.** *Cont.*

| Norm | Definition | Transactional Pole | Relational Pole |
|---|---|---|---|
| Power | Power is divided between the parties. | Organizations seeking as much control as possible; opportunistic behavior; taking advantage of opportunities to leverage power. | Organizations not controlling each other; refraining from using their powers; the promoting of mutual benefits from projects. |
| Propriety of means | Propriety of means requires the contracting organizations to have adequate means to fulfill their obligations. Different means and methods can be mobilized to achieve the desired results. | Use of advantageous means to maximize individual interests; unsuitable skills and experiences; imposed means and methods by one party. | Use adequate means to achieve the project objectives; necessary skills and experiences for the successful completion of the project; freedom in the choice of means and methods. |
| Reciprocity | Reciprocity ensures that there is equivalence of the exchange between the parties. This is a form of fairness and not necessarily equality. There is joint responsibility and mutuality. | Allocation of benefits and burdens; maximization of individual interests. | Sharing of benefits and burdens; maximization of collective interests. |
| Role integrity | Role integrity is characterized by a set of prescriptions defining the behavior and role of a party. | Individual interests taking precedence over collective interests; simple and defined roles; silo; use of formal rationality to achieve goals. | Collective interests overriding individual interests; complex and multidimensional roles; overcoming formal rationality to achieve goals. |
| Flexibility | Flexibility is the ability to change during the relationship and to adapt to external and internal circumstances. | Rigid agreement; difficulty adjusting the agreement and adapting to changes; revisions cannot happen or are complicated. | Supple agreement; ability to modify the agreement and adapt to changes; revisions are expected. |
| Harmonization | Harmonization requires conformity of the internal norms of the contract to the values shared by society, such as freedom, human rights, environmental matters, and mutual accommodation towards cooperative ends. | Contractual norms diverging from societal norms; dispute resolution in courts; strict and rigorous application of the contract; frequent litigation | Contractual norms in accordance with societal norms; internalization of dispute resolution; contract is not applied or referenced in the relationship; waiver of claims; no dispute. |
| Implementation of planning | Implementation of planning involves anticipating the performance and risks associated with executing the relationship. | Control of terms by a party; propensity for preliminary planning rather than future planning; use of standardized contracts; source of content is expressively communicated; specificity of content and obligations are high; external sources of obligations. | Use of external standards to plan performance; third-party determination; adjustments along the way; agree to agree; source content is partially determined by the relation; specificity is low, external and internal sources of obligations. |
| Effectuation of consent | Effectuation of consent is the trigger mechanism that causes the choice of an exchange or the perception of having a choice (not necessarily free of any external pressure). | Unique and absolute choice; restriction of future opportunities; formal rationality. | Agree to agree; indetermination of choice; substantive rationality. |

Grounded theory consists essentially of four steps, namely (1) coding the occurrences in the data, (2) integrating categories and their properties, (3) delimiting the contours of the theory, and (4) writing the theory (Strauss and Corbin 1990). Figure 1 below illustrates the process followed, in agreement with the four steps proposed by Strauss and Corbin, to identify the relational differences in the contractual language of the three project delivery methods analyzed here.

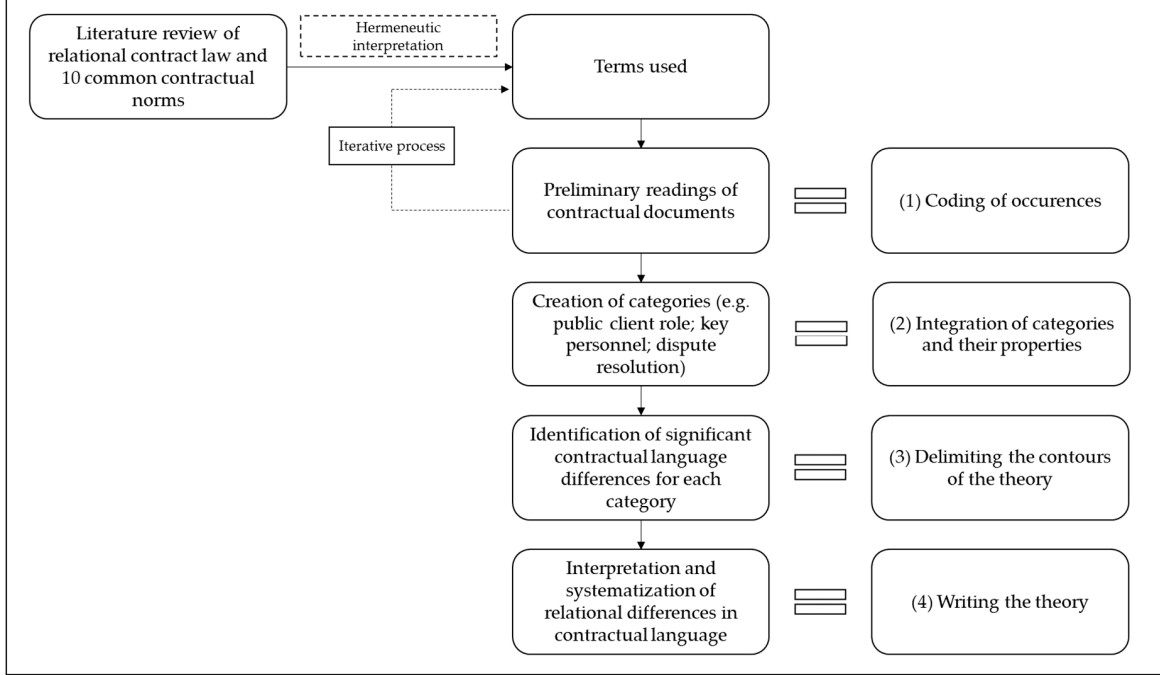

**Figure 1.** Grounded theory process.

Before proceeding to the coding of occurrences, a literature review of relational contract law and Macneil's 10 common contractual norms was effectuated. As illustrated in Section 2.1, Macneil and other researchers have conceived general principles and guidelines to qualify mechanisms according to the transactional and/or relational pole but have not developed a precise lexicon for identifying transactional and/or relational content in contracts pertaining to a specific industry such as public construction and professional services contracts. Because no blueprint is available, other than Harper's occurrence analysis of contractual language according to different delivery methods, which served as a foundation for this research (Harper 2014), a hermeneutic interpretation of Macneil's common contractual norms was necessary to identify the terms to be used in the first step of our application of grounded theory.

Hermeneutics is a methodology of interpretation aiming at identifying the meaning of a text, sentence, or term. Macneil's common contractual norms are not always precise and are often easily misunderstood. For example, the norm of solidarity has been misinterpreted as referring to contractual solidarity because of the linguistic similarity (Mandard 2012). However, contractual solidarity relates to the sharing of risks between parties, which is part of the reciprocity norm. When analyzing Macneil's theory and its influences, the term solidarity rather refers to Durkheim's organic and mechanical solidarity, which relates to trust (Durkheim 1893). In turn, trust refers to a bevy of concepts and approaches from which authors have elaborated related terms, such as collaboration and cooperation. Furthermore, because Macneil's common contractual norms are interdependent, terms, such as sustainability, are assigned to a fixed norm—in this case, flexibility—even though such terms could relate to other fixed norms, such as solidarity, depending on the type of mechanisms the different project delivery contracts provide for its achievement. The DBB and DB contracts we analyzed used external standards, such as LEED, to achieve sustainability, whereas the CMGC/IPD contracts we

analyzed used relational trust mechanisms. The use of external standards is a flexibility indicator under neoclassical contract law. Thus, in this study, choices had to be made to assign some terms to a fixed common contractual norm, which served as a primary node for coding the contractual documents using the NVivo software.

Once this first set of terms was established, we performed preliminary readings of the contractual documents to find precise language that could be related to the common contractual norms we used as primary nodes. Although the documents we analyzed are in French, as well as the terms used for the occurrence requests, they were translated for use in this paper and are presented in Table 3. Those terms and their synonyms were coded as child nodes. Afterwards, all the contractual documents were coded using the primary and child nodes, which enabled us to make a first attempt to create categories or core themes through axial coding, including themes such as the public client's role, the key personnel involved, and dispute resolution mechanisms. To delimit the contours of the theory, selective coding was used to more precisely relate the contractual language to the transactional and relational pole of the exchange spectrum, while analytical memos helped refine the nature of the categories. Following this third step, 10 core themes emerged from the contractual analysis: (1) from client spectator to client actor; (2) upstream involvement and multiplicity of key stakeholders; (3) BIM and the increasing complexity of roles; (4) different forms of work organization; (5) common management structures and communication tools; (6) immutability, mutability, competency, and expertise of personnel; (7) third-party involvement of independent certifiers and external professionals; (8) mutual planning and incompleteness; (9) sustainability measures from external standards to relational trust; and (10) hybrid forms of dispute resolution from amicable negotiation to reliance on courts. These core themes are discussed in Section 4. Finally, these core themes and the associated precise contractual language were systematized and interpreted to identify the significant relational differences between the project delivery methods analyzed and find which project delivery method is more relational and why. These differences are summarized in the following tables presented in the discussion of each subsection below.

**Table 3.** Terms used for occurrence requests according to Macneil's common contractual norms.

| *Common Contractual Norms* | *Terms Used* |
| --- | --- |
| Solidarity | collaboration, independent, coordination, committee, confidentiality, workshop, common, performance evaluation, common premises, meetings, help, assist, avoid conflicts of interest, proximity, cooperation, work together, multilateral, association |
| Cohesive norms | quality assurance, quality plan, quality management, quality control, control, oversight, authorization, approval, evaluation, internal/external audit, validation, inspection, follow-up, oversight, revision, recovery, guarantees, bail, safety, substitute, certification, examine, insurance, deductions |
| Power | consent, decision, recovery, must, can, authorization, allow, limit, force, final, power, authority, control under reserve of, approval, require, binding |
| Propriety of means | license, allowed, certificate, accreditation, skill, experience, training, qualification, equipment, means, methods, effort, care, judgment, knowledge, diligence, state of the art, expert, expertise |
| Reciprocity | at its expense and risk, responsibility, direct/indirect costs, property of digital data, fair, equity, reasonable, joint, mutual, sharing, remuneration, compensation, payment, charge, risks (management, control), assume, attributable, incentive, bonus, intellectual property, claim, damages (liquidated) |
| Role integrity | role, goal, interests, integration, common, key staff/personnel/employees, specialty, task, joint |

**Table 3.** *Cont.*

| Common Contractual Norms | Terms Used |
|---|---|
| Flexibility | change, standards, modifications, amendment, reset, negotiate/renegotiate, third party, arbitrator, independent body, independent auditor, obviously different condition, redetermine, expert committee, serious causes, delay, compelling reasons, force majeure, delay, grace period, energy, sustainability, environment, efficient, Leadership in Energy and Environmental Design (LEED) |
| Harmonization | good faith, arbitration, conflict, dispute, dispute resolution, relief notice, mediation, court, justice, jurisdiction, litigation, claim, decision, straighten, correct, protest, complaint, appeal, recourse, compensate |
| Implementation of planning | schedule, critical date, stage, critical stage, deadlines, critical deadlines, phase |
| Effectuation of consent | choice, freedom, agree to agree, binding, control, decision, opportunity, consent |

The use of grounded theory allowed us to create categories and hypotheses related to the common and relational norms proposed by Macneil. Our aim is to operationalize these norms according to the type of delivery method in order to add to a theoretical model which has been challenged because of its difficult application and the small number of theoretical key indicators. It should be noted that some norms are not individually addressed in the following sections because of the complex interaction between Macneil's norms. Macneil believed that the relationships between such norms could be illustrated by a spider web, and different tensions and pressures applied to a given norm would have the effect of deforming the web (Macneil and Campbell 2001). Role integrity, cohesive norms, solidarity, propriety of means, flexibility, and harmonization of conflict are the norms that reflect the greatest differences between the project delivery methods. Of the four remaining norms, namely power, implementation of planning, choice, and reciprocity, two are discussed in Section 4. For example, power is included in the discussion regarding the validation and control of other parties' work and the creation of common management structures. Implementation of planning is addressed in Section 4.2.1, Section 4.2.2, and Section 4.2.3 regarding forms of work organization, common management structures, and mutual planning processes. Reciprocity is treated in more general terms in Section 5 because of the absence of related mechanisms in the analyzed contracts. As pointed out by Jobidon et al., the reciprocity norm is predominantly influenced by the external regulatory framework and the procurement process, similar to the effectuation of the consent norm (Jobidon et al. 2018).

## 4. Results and Discussion

This section presents the different results of the contractual analysis based on the project delivery method. At the end of each subsection, a discussion and summary of the norm(s) analyzed is presented to highlight the relational aspects of the written and formal parts of the different project delivery methods, which amounts to the theory emerging from the analysis.

### 4.1. Role Integrity and Cohesive Norms: Participation, Multiplication, and Increasing Complexity

The first two prominent norms identified in the comparison of the project delivery methods are the role integrity and cohesive norms, which concern the behavior and roles of parties in a relationship, as well as the ways in which the exchange is maintained. It was found that CMGC/IPD contracts are more relational because of (1) the active role of the public client in the design and realization process, (2) the multiplicity and timely involvement of key stakeholders, and (3) the increasing complexity of roles using BIM.

4.1.1. From Client Spectator to Client Actor

One of the main differences between the project delivery methods is the role played by the public client in the design and construction process. The status of the administration as a moral and impersonal entity prevents the public client from directly participating in the realization of such infrastructure; however, a person or a team is typically invested with the necessary powers to ensure that the public clients' needs, which are usually addressed in the functional and technical construction program, are met. This is also true for architectural firms or more general professional services firms, who also designate a project manager to assume the coordination and supervision of the effectuation of the mandate.

In the traditional DBB process, the architect must work in close collaboration with the public client, which is represented by a project director, project manager, or representative, to ensure that the project meets the budget, schedule, and construction program. When the public client appoints the architectural firm as a designated professional for the project, this firm assumes the coordination of the other professionals retained by the public client without being responsible for the services provided. If the contract provides for site supervision, the architect also acts as the client's legal representative (Boulanger c. Commission Scolaire Régionale de l'Estrie 1993). However, there are limits to the supervision obligation of the architect: the work must be related to his specialty and must not require him to do the work of other professionals. Therefore, the public client has, via the architect's project manager, indirect control over the supervision of the mandate and the work of the different professionals. However, there is also residual direct control over the work of the architect and the general contractor. During the pre-conceptual study of the project, the architect must present the public client with its understanding of the construction program by identifying the main issues to be addressed, especially those related to sustainable development and the effective commissioning of the infrastructure. The architect must also present the concept of the design and demonstrate how it meets the performance objectives of the public client's program. When comments are made regarding the concept during the value analysis phase, the architect must modify the concept to obtain written approval of the final concept. This approval is not given directly by the public client or its representative but rather by an independent expert committee—a governance structure that will be more thoroughly analyzed in Section 4.3.1. The same process generally applies for the validation and control of the preliminary plans and specifications. This requires the submission of the plans, specifications, and study documents for comments to demonstrate the architectural firm's ability to achieve the expected performance goals and, if necessary, adjust the documents for written approval. While the independent expert committee retains control over the decisions, the public client is still passive participant in the process by virtue of the fact that (1) the public client is able to attend the presentation meetings and (2) the final plans and specifications must respect the planning established by the architect in collaboration with the public client's representative. The public client's representative has authority over any type of payment due to the professional services providers if they take any form of payment other than a lump sum. The representative plays a more active role in the services provided during construction by analyzing jointly with the architect the prices submitted by contractors for change orders, authorizing the date the contractor will commence work and issuing a payment recommendation to settle the contractor's payment claims.

A hybrid form of the public client's role emerges in DB contracts with the creation of a project management control committee (PMCC)—a structure composed of 12 different representatives. The PMCC includes three representatives of the public client: an official public representative, the project manager, and a representative who is at the discretion of the public client. The PMCC also includes two representatives of the executive director of infrastructure, one of whom is appointed by government decree, while the other is elected by the SQI, which can act either as a client or as a supporting organization for public clients in the management of such infrastructure projects. However, the public side of the committee is not always distributed this way; at times, the six public representatives are chosen in a discretionary fashion. The other six members are appointed

by the contractor: two are the official representatives of the contractor, one is a representative of the builder, and the other three are appointed at the discretion of the contractor and can, for example, be representatives of the architect. The PMCC has the power to establish subcommittees to address any issues that it considers relevant. The public client's official representative is obligated to act as the public client, exercising the functions and powers identified in the contract. The official representative can be substituted during the relationship, and if the official representative is not able to perform his functions for any reason, then the public client performs the functions that would otherwise be performed by the official representative. The public client's representative is not entitled to modify or waive any provision of the contract or to authorize any change to the contract.

The public client's representative often acts as communication intermediary between the public client and the contractor. For example, when contamination for which the public client is responsible is discovered by the contractor, the contractor must notify the public client's representative. The representative receives a lot of documentation from the contractor for information or notification purposes. This documentation includes the following: the design development program, a monthly report on design development, and a detailed schedule of work, among others. The main roles played by the representative in the realization process are those of reviewer and investigator.

As a reviewer, in accordance with a delivery schedule, the representative receives from the contractor each design data or grouping of design data developed in accordance with the design development program for review purposes. Design data mean all drawings, reports, documents, plans, software, formulas, calculations, and all other data related to the design of the infrastructure, and such data can be modified following the representative's review in accordance with the review procedure stipulated in the contract. The contract provides for workshops to allow the public client and the end users to comment on the design development of the infrastructure during the preparation of the design data. Workshops are organized by the contractor's representatives in collaboration with the public client's representative. After the workshops, the contractor prepares the minutes of the discussions, possible design solutions, and suggested design changes that will be sent to the public client's representative, as well as a copy of the notes, comments, sketches, drawings, drafts, plans, or diagrams prepared during the workshops.

The monthly design development report includes a summary presenting the general progress of the design and a description of the planning, design, energy efficiency, equipment selection, constructability, equipment integration, and building services issues that have been coordinated and incorporated into the documentation, and the public client and its representative have the option to provide the contractor with comments in accordance with the review procedure. Where appropriate, the contractor shall take into consideration the comments of the public client and review the report based on these comments. To accomplish this task, the contractor must give the public client's representative a reasonable opportunity to review all elements of the design data at any time and as soon as possible upon the receipt of a written request. A similar process is in place for the project management plan, which may be submitted to the public client's representative by the contractor in the case in which the public client requires revisions to the plan, if these revisions are considered reasonably necessary to the end users' health and safety or the quality of the services that will be provided in the infrastructure.

The project manager of the public client also has powers defined in the contract. If, in the project manager's opinion, the progress of the work is considerably behind schedule, then the project manager must deliver a notice to the contractor, who must produce and deliver, either to the representative or the project manager, a report identifying the reasons for this delay and a plan showing all the necessary measures that will be used to eliminate or reduce the delay. On the other hand, if the contractor is significantly ahead of schedule, the contractor shall notify the representative and the project manager. In this case, the contractor must also notify the representative if the contractor requests a provisional acceptance of work at least 180 days prior to the pre-established provisional acceptance date. According to the review procedure, the public client's representative has the right to demand that the contractor

present and submit a revised detailed work schedule explaining how and when the work will be performed to enable the public client to evaluate, at its sole discretion, whether it consents to the earlier proposed acceptance date.

The question of delays is omnipresent in DB contracts. In the case in which a contractor needs an extension to finish the work, the contractor must give a written notice to the public client's representative, who has the right to require the contractor to provide additional details in a written notice. The representative also has the right to access the facilities to investigate the validity of the contractor's claim. If the contractor does not comply with the extension process, the public client's representative then has the discretionary choice to accept the contractor's excuses or not. At the end of the construction process, during the commissioning phase, the contractor invites the public client's representative and the project manager to attend every aspect of the commissioning process and to comment on each aspect. Finally, it should be noted that the contractor must submit his proposals to the public client's representative for the safeguarding and storage of the project data and intellectual property rights. Therefore, the public client's representative, in his role as a reviewer, has a say in the design development program and its data, the project management plan, and the delays and extensions necessary to complete the project.

The second role played by the public client's representative in DB contracts is that of an investigator. Hence, all records, reports, and documents of the contractor may be inspected and verified by the public client at any time, at its request and with reasonable notice. The public client also has the right to review and verify the performance of the contractor, which means that the contractor must cooperate and require subcontractors to cooperate with the public client's representative responsible for the review and verification of the work. The public client's representative also has the right to access the design database of the contractor for inspection purposes.

The public client and its representative also have the authority to review the work on site by giving reasonable prior notice. If defects in design or construction are discovered following an examination by the public client's representative, the public client may, by giving notice to the contractor, increase its level of monitoring until the contractor demonstrates to the satisfaction of the public client that the contractor will be able to comply with the contractual requirements. There is, therefore, the possibility of increased control, review, and monitoring of the contractor's actions. If the public client's representative exercises this right and it is determined under the dispute resolution mechanisms that there were no actual design or construction defects, the public client's representative's action will be considered an event giving rise to a right to compensation. A similar process applies to the public client's representative's right to open the work for inspection if the public client's representative reasonably believes that there could be defects in design or construction or that the contractor has not complied with the requirements of the contract. In the public client's representative's request to the contractor, the public client's representative must justify and include the reasonable grounds on which the public client's representative bases the opinion. Once again, if there are no defects or if the contractor complies with the requirements of the contract, the public client's representative's action will be considered an event giving rise to compensation. The exercise of the rights of accrued monitoring and the right to open the work for inspection do not affect the obligations of the contractor with respect to the execution of the contract. The public client's representative may also instruct the contractor to stop and suspend work if the levels of vibration, noise, or dust could jeopardize the activities of the public client during such work. The compliance of the contractor to the policies of the public client is not intended to create or establish between the parties an employer–employee relationship or any other type of relationship.

CMGC/IPD contracts offer a clear contrast with those of the other two project delivery methods in terms of the public client's involvement. In DBB projects, the public client has a spectator role and a passive voice, and in DB projects, it acts as a reviewer and investigator. In CMGC/IPD contracts, the steering committee is the main governance structure, and it is formed as soon as possible to ensure the logistics and smooth running of the design and construction process. The committee is composed of a project manager who also acts as the committee leader, an IPD expert or facilitator, a representative

from the public client's technical expertise department, and a representative of the design team, who is usually an architect. The steering committee has multiple responsibilities, but not every member of the committee gets to participate in every aspect of the design and construction process. As such, the public client's representative has a say in different aspects of the process, including the planning of the IPD process as a whole, determining the agenda of each IPD workshop and the objectives of these meetings, identifying the participants whose presence is required and the inputs necessary to the smooth running of the workshops.

More specifically, the public client's representative is required to clearly articulate concerns, requirements, and expectations to guide the design team early in the process. The public client's representative should also be prepared to be challenged on some elements of the program to maximize the benefits of the IPD process while demonstrating openness and encouraging exchanges and the search for synergistic solutions. The public client must also ensure during workshops and throughout the process that its representative has the authority to make decisions and mobilize key internal resources to participate in the process when required. The contract also stipulates that the IPD process should begin at the start-up phase with the client and the program implementation team and then continue throughout the project with the extended project team (design, experts, and other consultants), until the building is occupied by its end users. It is suggested that one of the IPD workshops address the performance targets related to the public client's requirements. These include (1) a vision statement; (2) an analysis of the economic, environmental, and social objectives of the project; (3) the sustainability and energy-efficiency goals; (4) the strategies to achieve each goal and measure it; (5) an analysis of the project complexity (budget, schedule, approvals, site, policy, transportation, economy, etc.); (6) an analysis of strengths, weaknesses, opportunities, and threats; (7) a way to reach consensus decision-making earlier in the process based on common project objectives clearly defined from the outset; and (8) the mobilization of key decisionmakers, such as the client and the operator.

### 4.1.2. Upstream Involvement and Multiplicity of Key Stakeholders

In DBB contracts, there is simply no mention of the participation of other stakeholders in the process, whether design or construction, which illustrates and confirms the notion of the "silo" effect associated with this type of contract. Only classic stakeholders, such as the architect, engineers, and commissioning experts, take part in the realization process. The process is linear, with participants having to complete their tasks one after the other, thereby reinforcing the notion that there is a lack of collaboration between the parties, mostly due to the lack of structures through which they can interact.

The different committees provided for in DB contracts facilitate the participation of multiple and different stakeholders. The previously described PMCC may invite any other person to a meeting at the discretion of its members. Although not present in every DB contract we analyzed, there can be a provision allowing an independent certifier to attend PMCC meetings as a non-voting member. The role of the independent certifier will be more thoroughly discussed in Section 4.3.1.

The key personnel of the DB team consist of five persons, including the project director, the design manager, the construction manager, the quality manager, and the project finance officer. The team is flexible in its numbers and variety depending on the given stage of the design and construction process. During the risk management phase of the process, workshops bring together external members, representatives of the public client, and/or other stakeholders, as needed. This is not a formal obligation but reflects the voluntary approach of the DB team. Once the commencement of the work is authorized, arrangements are made to immediately identify the main parties involved in the technical aspects of the project, namely the representatives of the public client and its master teams; other governmental agencies and public services, including the city and the neighborhood where the infrastructure is being built; energy providers; the internal organization of the DB firm; and suppliers and subcontractors. For the final project to be completed successfully and according to the proposed timeline, all the stakeholders of the different disciplines are fully integrated. To this end, key stakeholders in the design, construction, and management of information technology, as well as key strategic entrepreneurs, are

quickly mobilized to define common objectives, establish strategies to achieve these objectives, and share the ideas that bring the most value to the project. The continuity of the implementation of these strategies with all the members of the design–build team from the beginning of the project's implementation makes it possible to link the conceptual engineering and architectural phases to the reality of the construction.

The CMGC/IPD process, which mainly takes the form of multidisciplinary workshops, brings together all project teams and their members, namely project managers, public client experts from the technical expertise department (architects, engineers, and estimators), planners, BIM and lean specialists, project architects, design engineers, contractors, construction managers, institutional representatives, and any other collaborator whose presence is deemed necessary at some point in the process. As in DB projects, this last point is of the utmost importance, because it brings flexibility to the CMGC/IPD team, allowing the inclusion of key stakeholders in the process. The first workshop in the programming phase also has the specific objective of validating and specifying the needs and issues of the various stakeholders, such as the end users, the operators, and the municipality. The contract also specifies that construction specialists should be involved as early as possible in the process to comment upon and influence the proposed solutions. They should actively participate in problem solving to guide the team towards the best solutions in terms of cost, constructability, and implementation. They also have the responsibility to (1) formulate recommendations to optimize operation and maintenance, (2) validate the choices affecting the costs of implementation, (3) ensure that the solutions put forward consider the transition from construction to commissioning, and (4) provide all necessary information to the design team for overall cost analysis. During the design phase, the construction manager must also provide advice to the design professionals concerning construction methods and choice of materials, thereby allowing for a faster and more economical realization of the project, and the construction manager will also determine the implementation strategy in collaboration with the construction professionals. The IPD workshops include a bevy of different stakeholders and the steering committee. The presence of an IPD expert, also called a facilitator, exemplifies the increasing complexity of the process under CMGC/IPD contracts, which requires external intervention to coordinate the professionals and stakeholders in the process. The design team, which collaborates in the planning of the workshops and actively participates in them, is composed of external professionals (architects and engineers) and an estimation specialist from the estimation department of the SQI. The experts, who also participate actively in all the workshops, are the construction manager, the professionals from the technical expertise department of the public client, an external cost specialist, and the operator. Consultants participate as needed in the workshops, depending on the issue, and include the public client's representative, a construction specialist, occupants and end users, a commissioning agent, an acoustician, a regulatory specialist, and other consultants, depending on the challenges at hand.

### 4.1.3. BIM and the Increasing Complexity of Roles

This subsection does not compare different delivery methods, because only some CMGC/IPD contracts and not all the contracts analyzed use the BIM process, although CMGC/IPD can be used in combination with all the project delivery methods. BIM will reach different maturity levels depending on the project delivery method chosen: for example, BIM used in conjunction with DBB would have a maturity level of 0 or 1, because achieving level 2 requires collaborative work, as is the case in Quebec's CMGC/IPD contracts. The impact of BIM on the different roles of the parties to the contract is reviewed here, while the actual process is discussed in Section 4.2.2. BIM is defined in the contract as a digital representation of the physical and functional characteristics of a building. It serves as a platform for sharing knowledge and data and is a decision support tool during the life cycle of a project.

The use of BIM in CMGC/IPD contracts impacts the construction manager by complexifying his role and responsibilities. The construction manager must collaborate with the public client and every other stakeholder involved in the BIM approach as soon as possible after the awarding of the contract to allow the public client and his team of BIM experts to prepare and draft the rules and procedures

envisaged to complete the BIM management plan. While this does not seem overly complicated, the BIM process mobilizes a large number of stakeholders. The team of BIM experts (senior manager, BIM integrator, and BIM coordinator), the architecture team (project manager, senior designer, and BIM manager), the mechanical engineering team (project manager, senior designer, and BIM manager), the structure team (project manager, senior designer, and BIM manager), the construction team (project manager, foreman, and BIM manager), and the other specialized resources (constructability analyst, IPD expert, sustainability specialist, and commissioning specialist) are to be consulted by the construction manager to prepare the rules and modalities for the BIM management plan, which is a complex and tedious operation. In addition to this process, the construction manager must work with the principal BIM manager and design professionals from the outset of the design activities. He needs to collaborate with all stakeholders involved in the BIM approach. This allows for optimal planning and coordination of the work methods of the various stakeholders and the required technologies, so that BIM supports the achievement of the guiding principles of the project in alignment with the public client's orientations regarding the deployment of integrated practices. The construction manager will also have to dedicate, for the duration of the mandate, a competent resource for the deployment of the BIM approach, namely a BIM construction manager. The BIM construction manager is responsible for implementing the BIM approach of the team, ensuring that the team is provided with all the information relevant to BIM management, and is also responsible for construction models.

The other role created using BIM in CMGC/IPD contracts is that of a BIM discipline manager, which is a BIM specialist from each of the professional services firms taking part in the project, such as the architecture and mechanical engineering firms. The BIM discipline manager is responsible for implementing the BIM approach. The BIM discipline manager must act as the main contact with the public client's team of BIM experts for the planning and deployment of the BIM approach. To fulfill this role, a BIM discipline manager must possess the credibility and power to influence the means and methods of the team to achieve the project's objectives. The construction manager shall therefore assign sufficiently qualified and experienced BIM resources to the project team to ensure the completion of the expected services. The BIM discipline manager is responsible for multiple tasks depending on the stage of the construction process. During the planning phase, the BIM discipline manager must identify the software that will be used by the team and provide it to the BIM integrator, who is part of the team of BIM experts, and validate the information technology requirements to ensure the interoperability of this software with the discipline. The BIM discipline manager must also document and update the integrated models for the discipline and submit it to the BIM integrator. Finally, the BIM discipline manager coordinates the training required by the team.

During the monitoring and quality control phase, which occurs throughout the development of plans, specifications, and work, the BIM discipline manager ensures the respect of the BIM management plan within the team and ensures the internal quality control of models and information before sharing them with other stakeholders. The BIM discipline manager also provides technical support to the team while supervising and coordinating its work to ensure adherence to the project schedule. The BIM discipline manager must also implement interference detection procedures and other types of review mechanisms and participate in the resolution of detected interference. Afterwards, during the construction phase, the BIM discipline manager must participate in the update of the models according to the change orders. As the project progresses, the BIM discipline manager coordinates the production of surveys effectuated by contractors and third parties and the modeling of erected systems and detects interferences between the planned work and the updated federated model. The BIM discipline manager is also responsible for communicating to construction professionals the detected variations and commenting on the impact of these variations, as well as proposing recommendations and actions related to the variations. Finally, the BIM discipline manager must establish the sequencing of critical work by using the BIM modeling tool.

### 4.1.4. Discussion and Summary Regarding Role Integrity and Cohesive Norms

Role integrity refers to a set of prescriptions defining the behavior and role of a party. The transactional pole is characterized by individual interests overriding collective interests, simple roles, and the use of formal rationality to achieve goals. Formal rationality refers to the calculability of means and procedures, such as contractual formalism (Kalberg 1980; Weber 1978). The relational pole of the role integrity norm states that collective interests override individual ones, roles become more complex and multidimensional, and parties overcome formal rationality to achieve goals (Macaulay 1985). Parties should thus strive for substantive rationality, which exists as a manifestation of one's ability to act in accordance with a set of values and is considered by Weber to be the only way to overcome the practical, rational way of life based on individual interests (Kalberg 1980; Weber 1978). The transactional pole of cohesive norms, which aim to maintain the exchange and the cohesion between stakeholders, is characterized by broken promises, unclear expectations, the impossibility of depending on other organizations, and the absence of gain rebalancing mechanisms. The relational pole means promises are kept, organizations can depend on each other, and gains are rebalanced. The validation, inspection, and revision processes of projects, which ensure that the parties' expectations are met, were the main themes used in the present contractual analysis. Table 4 summarizes the major relational differences regarding the role integrity and cohesive norms between the project delivery methods, which are discussed below.

**Table 4.** Key takeaways for the role integrity and cohesive norms.

| | *Public client role* | |
|---|---|---|
| Design–bid–build (DBB) | • Indirect control, through the architect's project manager, over supervision of the mandate and the work;<br>• Ability to comment on the concept;<br>• Passive participation through the attendance of presentation meetings;<br>• Authority over payments, other than lump sums, to service providers. | |
| Design–bid (DB) | • At least three representatives of the public client in the project management control committee (PMCC);<br>• Public client's representative's role as a reviewer and investigator;<br>• As a reviewer: control of design development program and data, control of project management plan, and control over delays and extensions;<br>• As investigator: inspection and verification of records/reports/documents of contractor, review and verification of performance of work, access to design database, on-site review of work, possibility of increased control/review/monitoring, and right to open the work for inspection in case of possible defect. | |
| Construction manager–general contractor/integrated project delivery (CMGC/IPD) | • Public client in the steering committee: planning of the IPD process; input in the agenda of the IPD workshops and objectives; and identification of the required participants and inputs necessary for the workshops;<br>• Can articulate concerns, requirements, and expectations to the design team early in the process;<br>• Participation as co-designer/equal in the workshops;<br>• Public client's representative's authority to make decisions and mobilize key internal resources. | |
| | *Upstream involvement and multiplicity of key stakeholders* | |
| DBB | • No participation of the stakeholders other than the traditional ones (the architect, engineers, the general contractor, and the commissioning agent);<br>• No upstream involvement of the stakeholders. | |

**Table 4.** *Cont.*

| | |
|---|---|
| DB | • PMCC representatives: 6 for the design–builder, 3 for the public client, 2 for the executive director of infrastructure, and 1 for Quebec's Société québécoise des infrastructures (SQI). The composition of PMCC may vary, but always 6 for the design–builder and 6 for the public side;<br>• Possibility of mobilizing an independent certifier;<br>• Possibility of including external members and other stakeholders during the risk management phase;<br>• Authorizing beginning of work: inclusion of a multiplicity of stakeholders according to needs (i.e., energy providers, city and neighborhood representatives);<br>• Quick mobilization of key stakeholders to define common objectives, strategies, and value-creating ideas. |
| CMGC/IPD | • Multidisciplinary workshops mobilizing a bevy of stakeholders: architecture, engineering and construction professionals, lean/BIM/IPD specialists and experts, planners, estimators, and any other stakeholder deemed necessary;<br>• Early validation and specification of the end users', operators', and municipality's needs and issues;<br>• Early involvement of the construction specialists. |
| | *Complexity of roles* |
| BIM | • BIM can be used in all the project delivery methods, but was only found in the CMGC/IPD contracts analyzed;<br>• Construction manager's collaboration with the public client and every stakeholder involved in the BIM approach;<br>• New roles under BIM: team of BIM experts, BIM manager (principal and disciplines), and BIM construction manager. |

In DBB contracts, the public client essentially has a spectator role in the DBB design and construction process, with the ability to preserve a passive voice and issue comments on the concepts, preliminary and final plans, and specifications. The public client's influence in the project essentially resides in the delegation of tasks to professional service providers. There is also no mention of the participation of other stakeholders in the design and construction process. The linearity of the process is confirmed, because the contracts do not provide structures to facilitate collaboration between different stakeholders. There is no way to create collective interests and contractual formalism forces tasks to be mostly performed in silos, thus placing DBB near the transactional pole.

DB contracts offer a different approach, with the public client's representative playing an active role as a reviewer and an investigator. DB contracts, in comparison with traditional DBB contracts, have a more internalized way of validating and controlling the work of the other parties. However, DB contracts still represent a power-centric way for the public client to participate in the design and construction process, based on reviews, investigations, and control. The public client is not a co-designer or an equal in the process. Regarding stakeholder involvement, PMCC members may invite any other person to a meeting, thus broadening the scope and opportunity for the involvement of key stakeholders in the design and construction process, which reinforces the relational aspects of the relationship. However, the involvement of end users is of the utmost importance in the design of complex infrastructure, and the fact that they are not de facto included in the PMCC is problematic. This is important regarding the role integrity norm, in which the relational pole is characterized by collective interests, as well as complex and multidimensional roles. Overall, there are a multiplicity of actors and key stakeholders involved in the design and construction phase of a DB project. While the upstream involvement of these stakeholders is not always clear and formalized, there is an option for a voluntary means of involving key stakeholders in DB projects, which reinforces the need for a strong ex-ante qualitative assessment of a firm's strategies for the realization of the project during the procurement phase.

In CMGC/IPD projects, the public client plays a more active role in the design and construction process, especially through its involvement in the steering committee. This point is key for allowing the administration to fulfill its role as defender of the public interest during public construction projects. Hence, collective interests override individual interests, especially because all the project stakeholders, including the public client, must establish clear and measurable objectives to ensure their commitment and active participation in the development of strategies to achieve these targets. The public client's representative has a more active voice and role in the design and construction process, as the public client's representative takes part in the team as a co-designer, and not merely as a reviewer, investigator, or a spectator. The upstream involvement of key stakeholders and their considerations is also one of the main differences between CMGC/IPD and the other project delivery methods. While DB teams have the option to involve as many stakeholders as necessary, it is the temporal aspect of their involvement and their multiplicity that distinguishes the CMGC/IPD process. Furthermore, the implementation of BIM in CMGC/IPD contracts creates multiple different roles and teams and complexifies the tasks and responsibilities associated with these roles. The different roles are in constant interaction during the whole process, thus approaching the relational pole of the role integrity norm, which requires the overriding of individual interests with collective ones. The interplay of these complex and multidimensional roles is a feat achieved in CMGC/IPD through mutual planning, coordination, and review.

### 4.2. Solidarity and Propriety of Means: Governance, Workshop Scope, and Key Stakeholders

The next two norms displaying major differences between the project delivery methods are solidarity, which understood as trust, and propriety of means—a norm relating to the competency of key stakeholders. It was found that CMGC/IPD contracts are more relational than DBB contracts because of their multilateral work organization, the existence of common governance and communication structures, and the existence of penalties associated with changes of key stakeholders. DB contracts also display important relational aspects, but it was difficult to compare these relational aspects to those of CMGC/IPD contracts because of the internalized work processes of DB firms, which are not all formalized in the contractual documents.

### 4.2.1. Different Forms of Work Organization

The organization of the work is of capital importance for the realization of the project, because it has the potential to be a source of cohesion and trust creation and to level the expectations of the parties and allow information exchange and follow-up, which in turn favors effective coordination.

Although interactions between stakeholders in a classic DBB setting are infrequent, workshops still occur to create value for the project. During the planning phase, the public client has the option to ask an architectural firm to provide input for a construction program. To carry out this service, one of the firms opted for an integrated design process to identify innovative strategies, techniques, and materials that could be incorporated into the project. The potential strategies identified matters of concern, such as energy, water management, heat islands, and materials selection. The architectural firm explored options concerning building orientation, air preheating mechanisms, natural ventilation, rainwater reuse, green roofs, and space and vegetative optimization, just to name a few. However, these strategies are voluntary approaches and not formalized contractual mechanisms, which were qualitatively evaluated during the procurement process. The only true contractual structures found in DBB contracts are value analysis workshops for design professionals held during the conceptual stage. These workshops are led by an independent expert coordinator and may involve other external professionals. The design professionals must actively participate in these workshops but do so according to their own autonomous contractual relationships with the client and not based on relationships with each other. The professionals must then verify the feasibility of the workshop recommendations and, if possible, develop them for integration into the project. However, they are not obligated to do so. This, once again, reinforces the silo notion associated with the use of the DBB method.

As in DBB contracts, DB contracts provide for pre-contractual processes in place to help validate information and create synergy between the stakeholders. As such, workshops concerning technical clarifications, seismic structure and design, architecture and construction, energy targets, commissioning, and financial and legal matters are organized to facilitate the comprehension of the projects by all the stakeholders. However, the subject matter of this paper concerns formalized contractual mechanisms and not ex-ante mechanisms, which have been previously explored and analyzed (Jobidon et al. 2018). DB contracts stipulate that the workshops should allow the public client and end users to comment on the development of the design. The objective of these workshops is to facilitate the incorporation of the public client's comments into the design data. These workshops are organized collaboratively between the public client's representative and the contractor's representative, who mutually develop an acceptable schedule. The parties to the contract must also agree in advance on who may participate in the workshops. The contractor is responsible for providing the minutes of workshop discussions, possible design solutions, and suggested design changes to the public client, as well as the agenda for the next workshop. Moreover, the public client is not bound by the comments made during these workshops.

In one of the projects analyzed, the DB firm promoted the integration of all the key stakeholders through an integrated design and construction process to ensure the optimization of the overall value of the project. Therefore, professionals specializing in project management, cost estimation and control, architecture, engineering, material procurement and control, planning, quality control, and the environment were mobilized by the DB team. This interdisciplinary team was mandated to continuously refine the definition of the project during the workshops. It could review the construction concepts and methodology, create procedures to obtain the various required permits, coordinate and collaborate with the PMCC, establish a supply and procurement group, plan and organize temporary facilities, ensure the availability of construction labor, update requirements and implementation practices, and negotiate with the trade unions for a harmonious execution of the construction. The firm also had a voluntary approach in terms of risk management and identification. This approach took place during monthly workshops that brought together external members, the public client's representatives, and other stakeholders. As to internal work organization, the design–builder had the freedom to voluntarily set up an integrated approach through multidisciplinary workshops. The only formalized DB workshop was the comment and review process between the public client and end users with the design–builder regarding design data.

The CMGC/IPD delivery method is on a whole other level in terms of sheer quantity and scope of workshops. It not only provides for the expected IPD workshops, but also workshops addressing (1) value analysis, (2) risk and opportunity analysis, (3) commissioning and quality management, and (4) visual coordination and interference detection when the BIM process is used. The IPD process objective is well stated in the contracts: to promote synergy among all members of the planning and design team so that all collectively developed solutions optimally meet the needs of the public client's functional and technical program. The efforts of each stakeholder should be harmonized and collectivized to reduce the long-term operating and maintenance costs by improving the environmental aspects of the infrastructure and the site.

During the start-up phase, which includes an IPD introduction, the creation of the program, and the analysis of real estate options, the IPD process is presented to the public client and to the team that will carry out the construction program by the steering committee. The steering committee has all the necessary latitude to plan the agenda and subject matter of the different IPD workshops. The first workshop of the start-up phase focuses on defining the vision and objectives related to the public client's requirements, examining the functional and technical needs, searching for optimization measures, and validating and specifying the needs and issues of the various stakeholders, such as the end users and operators. The second workshop identifies real estate options, analyzing them and obtaining a consensus on a preferred option.

During the planning stage, which includes the execution of concepts and preliminary plans, the roles and responsibilities of each stakeholder are established, and the roadmap of the IPD process is validated by external professionals. The first workshop during this stage establishes the vision of the project and the performance objectives. It presents the shared vision of the entire project and aligns the performance objectives, analyzes the complexity of the project to identify the main issues, and clarifies and optimizes the objectives of the construction program. The second workshop has a more exploratory focus, with the stakeholders creating different sketches based on the established performance objectives, seeking synergies between components, and improving the concept while also determining the costs of the proposed strategies and synergies. The third workshop focuses on optimizing the design solutions integrated into the preliminary plans and specifications with all the professionals. The stakeholders will then evaluate these solutions with respect to the performance objectives of the public client and confirm the energy targets and energy-efficiency strategies. Finally, the last workshop aims to develop the detailed design, document it, optimize the value and performance according to their costs, evaluate the synergy of systems, and make choices regarding systems, equipment, and materials. Afterwards, the team will focus on defining the commissioning options for better performance, validating the constructability of the different systems, and optimizing the schedule and commissioning. There is also an optional workshop for the realization stage, which concerns the final plans and work. BIM serves as a support throughout the IPD process, mainly by using federated models during the design follow-up workshops. The models are also tools for communication, three-dimensional (3D) coordination, and work management planning.

The objective of the value optimization process is to identify, develop, and implement any functional, operational, and technical option to respect or optimize the content, quality, cost, and schedule of the project. This process takes place in a value analysis workshop involving a committee consisting of the public client's project team and external project professionals whose goal is to optimize the project to increase value while reducing project costs. The workshop is held at the end of the concept stage or, at the latest, when preliminary plans are 50% completed. During the workshop, BIM serves as a communication and visualization tool. The construction manager must actively take part in the value analysis workshop and provide for preparation and work time outside of the workshop. The construction manager's role is to advise and support the professionals in reviewing documents and data to implement the value analysis recommendations retained by the public client. More specifically, the construction manager must assess the feasibility and risks associated with each analyzed option. The construction manager must also consider the relevance of the option; the risks and opportunities associated with its potential implementation; the needs of the project; the technical feasibility; the impacts on cost, schedule, and operation; the financial and economic feasibility; the strategy for implementation; and the organizational feasibility.

The management of project risks and opportunities relies on the identification of the elements that are potentially favorable or unfavorable to the achievement of the project objectives. The process mainly consists of identifying and evaluating risks and opportunities and their likelihood of occurrence and impact on the project and determining a response plan and a budget to build a reserve for the control of these risks and opportunities. The responsibility of this analysis falls on the construction manager, through a risk manager, who will participate in the risk analysis workshops, and on the public client's project team and the SQI estimation team. The construction manager will ensure the coordination of the risk and opportunity management efforts between the service providers, the construction professionals, and the project team.

Commissioning consists of a continuous quality assurance process applied throughout the life cycle of the project to ensure the performance of the systems in accordance with the requirements of the public client, the contractual documents, and the intentions of the designers. This process is based on recognized practices, including the American Society of Heating, Refrigerating and Air-Conditioning Engineers (ASHRAE) standards, and takes place throughout the life cycle of the project, until its closure. Commissioning focuses on the major architectural and technical components, including networks;

mechanical, electrical, functional, and operational systems; and mobile equipment requiring quality and performance controls. Commissioning is a process that requires the collaboration of the entire commissioning team. During the design phase, this team consists of an independent commissioning agent; the public client's team, including the project manager, internal expert professionals, technical specialists, facility operators, and end users' representatives; the design professionals; and the construction manager. During, and from, the construction phase, other members will join the team, including the general contractor, some subcontractors, and the site supervisor. Commissioning is continuously addressed in IPD workshops and coordination committee meetings.

3D coordination includes visual coordination and interference detection between different building systems. Just like commissioning, it serves a quality control purpose, is developed by professionals collaboratively, and is an important communication tool that supports decision-making during IPD workshops and follow-up meetings that include all the project stakeholders. Visual coordination is a collaborative process in which professionals use a 3D model to provide comments and validate certain aspects of the design. The contract stipulates that visual validation should be continuously used from the beginning of the design phase to the construction phase. Participation is supported using a digital collaborative platform. There are two types of workshops associated with visual coordination. The first is an interdisciplinary coordination workshop, which takes place during the development of conceptual and preliminary documents. Visual coordination brings together project stakeholders around 3D models developed by each discipline and is based on the use of an integrated model, which allows designers to develop concepts by considering information provided by other disciplines. The main advantages of these workshops are the effective communication of constraints, the optimization of the design, and the possibility of avoiding backtracking. The second type of workshop concerns the coordination and monitoring of the concept. Project stakeholders, other than designers, can visualize and understand the level of advancement of a concept through the dissemination of federated models on a digital platform. The models therefore serve as a common information structure for the follow-up of problems, which will be attributed to targeted stakeholders responsible for their resolution. This collaborative platform becomes the central point for the monitoring of the design and is always accessible by the stakeholders, who do not directly manipulate the models.

The interference detection process uses software to determine the potential conflicts, which means potential collisions or clashes between various structural and mechanical systems (Azhar 2011), between different building systems from the 3D models of each discipline, previously grouped together within a federated model. It is applied during the design phase, because the concept and the modeling are advanced at this point, and it may be difficult to visually determine clashes between different systems. Potential clashes are therefore detected prior to the call for construction tenders, which takes place at the beginning of the construction phase, just after the completion of the final plans and specifications developed in the planning phase. This allows the virtual coordination of the systems before they are manufactured and assembled on the construction site. For the interference detection workshops to be a value-creating process, it is necessary to start with the analysis and resolution of conflicts related to larger system components and then progressively analyze the smallest elements that may have an impact on the execution of the work. These workshops are initially conducted within each discipline and then carried out between different disciplines. There are three types of interference detection workshops, namely clash detection between architecture and/or structural models and mechanical, electrical, and plumbing models; clash detection between architectural and structural models; and clash detection between electromechanical elements. The organization and execution of these workshops are the responsibility of the project teams. BIM designers and the managers of the disciplines or sub-disciplines that are the subject of the workshop are required to participate, while representatives from other disciplines may attend the workshops if they are deemed useful. These workshops can be held in two stages, with the initial workshop consisting of performing the automated analyses, the categorization of interferences, and the preparation of the files for the collaboration

platform. The second stage is an interference resolution workshop to review interference that requires special coordination among the affected stakeholders and to determine solutions or adjustments.

4.2.2. Common Management Structures and Communication Tools

Other critical aspects pertaining to contracts are the common management structures and communication tools put in place to facilitate the realization of the project's objectives. Common management structures spread the power from a single point to multiple actors, whereas communication tools help foster trust and level the parties' expectations.

In DBB contracts, there are no common management structures, but some stipulations formalize technical aspects of the communication process. For instance, there is an information and identification table for each plan sheet that contains the name of the file, the title of the drawing, the public client project number, and so on. The format for the electronic copies of the plans and quotes is also formalized, and the parties use transmission slips to confirm the sending of these copies. Service providers in DBB projects must also prepare the specifications based on the presentation and formulation requirements of Construction Specifications Canada. DBB contracts also stipulate the creation of a project office located on site to promote complete and timely coordination between the key stakeholders. This office facilitates the monitoring of the work by making it possible to carry out the necessary verifications, whether quantitative or qualitative, and keeps a log of observations regarding the work. These common management and communication measures are quite shallow in comparison with those of the other project delivery methods.

Just as in DBB contracts, one of the critical components of the integrated process voluntarily chosen by the DB firm is the creation of a project office, which reports to the project director, is set up to mobilize the design–build team, and ensures that the construction and operation of the infrastructure correspond to every aspect of the design. However, DB contracts implement more common management structures than their DBB counterparts. The previously discussed PMCC can create subcommittees to address issues it considers to be relevant. The PMCC is obligated to make all reasonable efforts to promote cooperative and effective communication between the parties. The role played by the PMCC is to receive and review matters relating to tasks and corrective tasks, such as design, construction, and commissioning issues and the detailed work schedule. Moreover, the PMCC is responsible for matters arising from reports or documents submitted by the design–builder for significant changes affecting the delivery schedule, public interest issues, and quality assurance and security issues. The public client and the design–builder can also refer specific issues to the PMCC. Community and media relations are under the authority of the PMCC, as are the receipt and review of progress reports, commissioning plans, and final, accepted plans. Equipment and interface issues are also part of the PMCC's scope of work. The PMCC, therefore, has wide authority, although its role is limited to making non-binding recommendations to the parties. PMCC members can also adopt their own procedures and practices to carry out their activities and invite anyone to their meetings. Members meet at least once a month until the final acceptance certificate is issued, thus fostering relational trust through repetitive interactions. The contractor is responsible for keeping the minutes of the recommendations and meetings of the PMCC. These minutes need to be sent to the other parties, and the contractor provides the public client with a complete set of minutes of recommendations and meetings for review, upon which the public client may comment. In turn, the contractor must incorporate these comments and share the amended minutes with the other parties.

As for communications, the contractor in DB contracts must provide the public client with a copy of any notice, order, requirement, or important communication that the contractor receives from a government authority with respect to the project's activities. There are also numerous formalities for the transmission of notices under the DB agreement. When a notice is delivered by facsimile or email, an original copy of the notice must be delivered by hand or by registered mail without delay, and an acknowledgement of receipt must be obtained. Measures also exist to record the moment of reception of notices by mail and by hand and whether they took the form of a facsimile or email.

To take effect, notices and other official communications must be in writing, delivered according to the pre-established communication protocol, and signed by an authorized representative of the parties. Verbal communications do not constitute official communications, and no party is obligated to act on any verbal communication, instruction, or assurance unless confirmed in writing. If a party decides to act upon a verbal communication, it does so at its own risk. With respect to tools, a work breakdown structure is used as a basis for the process of encoding contract-specified work, data, documents, drawings and sketches, and project correspondence. Live and interactive information management systems are also made available to members of the project team. The project system modules are integrated into a relational database so that each stakeholder benefits from the consultation and sharing of the data.

CMGC/IPD contracts provide for a more complex governance structure than the other project delivery methods. First, CMGC/IPD requires the creation of three different governance committees, namely a management committee, a design coordination committee, and a site coordination committee. The construction manager is required to participate in all meetings necessary for the complete fulfillment of the mandate. If necessary, and at the request of the public client, the designated representatives of the construction manager may be convened to participate in meetings of the various governance committees.

The management committee holds meetings at the project office on a regular but undefined basis. As is the case with DBB and DB contracts, CMGC/IPD contracts also set up a collaborative work environment between the construction manager, the professional services providers, the public client's project manager, and the SQI estimation professionals. The management committee meetings held at the project office focus on planning, monitoring, control of content, cost, and schedule parameters, as well as change orders, risk, communications, and procurement management. During each management meeting, the stakeholders review the events and choices affecting the cost plan to anticipate potential issues with respect to budget compliance. The construction professionals and the construction manager, with the help of the estimators, must continuously update and identify project cost data pertaining to their area of expertise. When a potential problem is identified, they must recommend means by which to ensure compliance with the budget. The meetings are coordinated by the project manager, who is responsible for producing and distributing the minutes. The design coordination committee meetings are coordinated by the architect, who ensures the production and dissemination of the reports. Weekly or bi-weekly meetings can be scheduled in parallel to streamline the ordinary meetings when points need to be discussed in depth and when they do not concern all stakeholders, such as commissioning and sustainability issues. The site coordination meetings include all the project managers of the construction professionals and all the site supervisors, as well as the construction manager. The architect still plays the role of coordinator. The site coordination committee offers technical expertise by verifying shop drawings, material specification sheets, and work samples. The committee members must provide the list of shop drawings, material specifications, samples of work, lists of new directives, and all other studies, analyses, or follow-up documents no later than 24 h prior to meetings. Site meetings are held weekly or every two weeks, but the frequency of these meetings can be modified depending on the stage of progress, the evolution of the site, the nature and complexity of the intervention, and the need for prompt technical advice.

A steering committee, as previously stated, also exists in addition to these three governance committees. Throughout the IPD process, the steering committee must collaborate and share information, be determined to achieve the objectives of the project, act with confidence and transparency, be open-minded and creative, and continue collaborative work between the planned IPD workshops. This multidisciplinary IPD team must establish the general and specific objectives of the project in accordance with the parameters authorized by the public client. It must also establish quantifiable performance objectives for energy, indoor air quality, emissions, water consumption, site use, choice of materials, waste management, commissioning, and maintenance. The multidisciplinary IPD team also identifies the potential impacts of construction choices, initiates and integrates a design process that

will reduce project costs and schedule within the approved project parameters and the established environmental and energy performance targets. It is also responsible for identifying project risks and mitigation measures, as well as exploring the associated costs, timeframes, and expertise required to eliminate undesirable outcomes in the later stages of the project. Finally, it must validate the project schedule at each stage and the budget on an ongoing basis.

CMGC/IPD contracts implementing BIM have a statement of intent addressing their objectives, one of which is to improve communication by organizing information so it can be shared. BIM supports the goal of creating a centralized and easily accessible source of information by using open BIM standards for the exchange of data between different tools. BIM software permits the extraction of two-dimensional (2D) views of design templates that are used for issuing tender plans and visualization features of the federated model directly from a digital collaboration platform. Digital data, which include any information, communication, drawing, model, database, analysis, specification, or other document as created or hosted for the project, are shared between all the relevant stakeholders. This enables easy and early access to data, useful and up-to-date information in real time, better interdisciplinary coordination, fast and efficient communication, access to a centralized source of information, and the elimination of the duplication of information. Therefore, digital data must be shared and made available to all other project stakeholders involved in the BIM approach. The construction manager is responsible for disseminating and making available the information to all the project stakeholders. When and if possible, the digital data should be hosted on a centralized data server with continuous live links. Otherwise, they must be transmitted between the different project teams on a regular basis, via a pre-established tool for file transfer, so that the relevant models necessary for coordination are linked to the respective production models through dead links. Concept development and ongoing coordination between professionals is supported by the sharing, on a regular basis, of the design models from each discipline. These periodic exchanges allow the creation of integrated models and a federated model through which data analysis and optimization of communication are made possible. The integrated model allows designers to develop their concept while considering information provided by other disciplines. The federated model is a centralized coordination model under the responsibility of the principal BIM manager and integrates the models from all the disciplines for communication and analysis purposes. The federated model is particularly useful for interference detection workshops, constructability analysis, and design monitoring.

CMGC/IPD contracts using BIM also adopt a specific tool for the interference detection workshops. The parties use a common software that creates the federated model, performs interference detection analysis, and produces detailed reports of identified clashes. However, the contract specifies that the interference detection tools have two main limitations: They do not provide a platform for sharing and monitoring interference reports, and they can also detect many clashes that are not important or real. To counter these limitations, the parties must focus specifically on process planning to ensure that project teams do not waste unnecessary time or effort on coordination. The contract thus stipulates that it is necessary to set up a parallel process for the sharing and monitoring of conflicts, which will be attributed to a targeted manager. This last aspect is facilitated by the dissemination of conflicts on the collaborative platform using the BIM collaboration format, which allows visualization and rapid localization. The BIM collaboration format is an open file format that supports communication flow by allowing different software to exchange information seamlessly without having to manipulate the models in their native format. It also allows discussions to be created and followed based on a georeferenced graphic extracted from a BIM model. Once published on the platform, clashes can be shared and followed by the creation of a discussion topic, as well as by visual coordination.

A specific coordination strategy is also required during the period right before the submission of BIM deliverables. This ensures coordination and consistency between the deliverables of each of the disciplines. Prior to a submission, the shared models are frozen, and weekly submissions are suspended, after which each discipline will retrieve the shared models for final validation and coordination. Afterwards, the models are resubmitted for final sharing before the official submission.

Each discipline then recovers the shared models and finalizes the graphic aspects of the models. This last step is crucial, because special emphasis is placed on 3D modeling and coordination, but the valid contractual documents are the 2D deliverables, which must be of high quality to allow the effective communication of the developed concept. Finally, the quality control process carried out by the project teams under the supervision of the BIM discipline managers and the main BIM manager is an ongoing process that ensures that project teams use best practices for modeling project information. This process also ensures that each of the stakeholders involved in the BIM approach can build mutual trust regarding shared information. During this process, the public client will carry out summary quality checks at different project milestones to ensure that the models meet their requirements.

### 4.2.3. Immutability, Mutability, Competency, and Expertise of Personnel

Another important aspect regarding relational contracts is the influence of the propriety of means norm, which implies having the adequate means to fulfill the obligations and the necessary skills and experience for the successful completion of the project.

At the procurement stage of a DBB project, the project manager of a professional services provider must, at the time of the submission of the bid, be a permanent resource. In addition, all personnel allocated to the mandate must be domiciled in Quebec or, when an intergovernmental agreement applies, in a territory or province covered by said agreement. Neither the project manager nor any of the professionals proposed in the tender may be changed unless expressly authorized by the public client. If, and when, the service provider requires such authorization, the service provider must send the public client the curriculum vitae of the replacement personnel, which must indicate a level of competence equal to or higher than that of the replaced personnel. The public client has no obligation to authorize such substitutions. The architect in DBB projects must also, in the deliverables of the final plans and specifications stage, create a list of equipment and systems that require training for their operation. As for deliverables during construction, the architect must also plan and monitor training sessions for operating personnel according to the previously stated list.

As is the case in DBB contracts, DB contracts identify key personnel, usually the project director, the design manager, the construction manager, the quality manager, and the project finance officer. The contractor must make all reasonable efforts to ensure that these personnel continue to participate full-time in the project. The contractor does not have the authority to select or authorize a replacement for one of these key persons without the prior written consent of the public client, who cannot refuse when the proposed replacement has qualifications and experience equivalent to those of the replaced person, based on the public client's judgment.

The only mutable personnel in DB contracts are the representatives sitting on the PMCC, who can be appointed and removed by a party who gives written notice to the other parties at any time. However, a substitution procedure exists in specific cases. Under the DB agreement, the contractor does not have the right to terminate or agree to terminate related documents, unless it is to correct or prevent an event causing the contractor to default and where no other reasonable measure is available to correct or prevent this event. The other two cases where this exception arises is when the public client considers it necessary to rectify such circumstances by terminating a subcontract or when a party or the contractor has committed a prohibited act under the agreement. In these cases, the contractor will appoint a replacement, subject to the prior consent of the public client. No replacement will be agreed upon if (1) the proposed replacement's activities are incompatible with the role of the public client with respect to its activities; (2) the public client considers the financial situation of such replacement to be unsatisfactory; (3) the public client reasonably judges the experience, skills, or ability of such replacement and his team to be unsatisfactory; and (4) the subsequent agreement with the replacement is not substantially similar to that of the person replaced.

The contractor is also obligated to keep the appointed builder and cannot replace him without the prior written consent of the public client, and the contractor must require the builder to maintain and not replace the appointed principal subcontractor unless the prior written consent of the public client

is obtained. The consent of the public client cannot be withheld without a valid reason, such as an incompatibility with the role of the public client regarding its activities.

There is also a substitution procedure present in the direct lenders' agreement—an agreement entered into between the public client, the lenders' representative, and the contractor. There are three situations where it is possible to replace the contractor: (1) an event or default under the financing agreements or the surety's documents, (2) any situation that authorizes the rights related to the lenders' sureties, or (3) in the pre-established substitution period. If one of these criteria is met, the lenders' representative may, subject to the approval of the public client, appoint a suitable replacement contractor by submitting a notice to the public client and designated representatives. The notice shall be accompanied by all information reasonably necessary for the public client to decide whether the proposed contractor is suitable. The public client must promptly inform the lenders' representative of any additional information that is reasonably needed to assess the suitability of the replacement contractor. The public client will then inform the lenders' representative of its decision. It is considered reasonable for the public client to refuse the approval if the previous contractor still has uncorrected violations and there is no rectification plan, or if the proposed securities of the replacement contractor substantially differ from those of the replaced contractor or have the effect of increasing the public client's responsibilities. An appropriate substitute must thus have the legal capacity and authority to be a party to the contract; must employ persons with the appropriate qualifications, experience, and technical competence; and must have enough financial resources and subcontractors to fulfill the obligations of the agreement.

Furthermore, the contractor warrants in the contract that the contractor, the contractor's shareholders, and the primary subcontractors are experienced in the design, construction, and financing of large infrastructure similar to the project at hand and that they collectively possess the ability, experience, skills, and knowledge to conduct the project's activities. The DB firm also affirms, in voluntary fashion, that, because the project will occur over a period of several years, it is possible that the staff and personnel effecting the realization of the infrastructure will change throughout the implementation of the project. If this is the case, the DB firm will provide for a transition period during which the new resources and the resources leaving the project will exchange relevant information and allow for progressive integration. The project contractor is also obligated by the contract to always provide enough personnel, resources, and training for the purposes of the project, and such personnel, resources, and training are determined by the contractor himself to comply with the provisions of the public client's requirements. The public client is not liable for the personnel and resources hired by the contractor.

In CMGC/IPD contracts, the personnel assigned by the construction manager to execute the contract must be those designated in the tender. The construction manager may not replace such personnel without having obtained prior authorization from the public client and must seek to do so by forwarding the curriculum vitae of the replacement personnel, who must be at least as equally competent as the replaced personnel. Unlike DBB and DB contracts, CMGC/IPD contracts stipulate that, if for any reason other than serious reasons, such as the health status of the personnel, the substitution of the construction manager's project manager will lead to a pre-established fixed lump sum penalty deducted from the amount of the contract, as is the case if the superintendent is replaced. The same type of penalty exists for the professional services providers in the event they replace their project managers, main designers, or site supervisors, but in this case, the penalty takes the form of a pre-established fixed percentage penalty. The same penalty applies if the BIM discipline manager is replaced before the signing of the BIM agreement. These penalties exist to compensate the public client for problems, inconveniences, extra efforts, and additional delays caused by the replacement of key personnel.

Other stakeholders are more easily replaceable in CMGC/IPD contracts. The construction manager must provide qualified, competent, and experienced workers in sufficient numbers to perform the work promptly and in an appropriate manner. However, the public client may require the replacement of any employee whom it considers to be incompetent, negligent, or otherwise undesirable. A peculiar feature

of this process is that an oral communication from the public client's project manager is sufficient to trigger the exercise of this right. The construction manager must also provide a qualified superintendent and qualified foremen. The public client may require the replacement of a superintendent or foreman whom it considers incompetent or negligent or for any other good reason. Therefore, personnel, such as the construction manager's and the services providers' project managers, the main designers, and the BIM discipline managers, are immutable in CMGC/IPD contracts but not the superintendent, foremen, or workers.

The construction manager is also responsible for training. The construction manager must organize, in collaboration with specialized contractors, suppliers and operating personnel, training sessions for equipment, and systems where required. The construction manager must also satisfy the public client's air quality requirements through the training of his personnel and subcontractors. Because the construction manager is also part of the commissioning team, he must coordinate tests, start-ups, and training sessions for the team. The training given by the construction manager must be completed and the commissioning must be carried out before the work's reception procedure can begin. Once again, the use of BIM in CMGC/IPD contracts has an impact on the parties' obligations with respect to training. The construction manager's personnel involved in the BIM approach and IPD workshops must have the required knowledge and skills, including tools and processes, because they will have to manage the deployment of the BIM approach. BIM training of the construction manager's team is entirely the responsibility of, and at the expense of, the construction manager so that the team benefits from BIM experience. A major difference between CMGC/IPD and DBB and DB teams is that, instead of establishing their experience level through curricula vitae, the construction manager's team must demonstrate their competence to the public client's team of BIM experts, which will comment upon their skills. The construction manager must therefore coordinate the training required by the team according to the training offered by the team of BIM experts. The construction manager is not the only stakeholder who has new responsibilities with respect to training. Each BIM discipline manager, in collaboration with the senior BIM manager, must complete a BIM usage analysis grid to optimize the implementation of the proposed BIM uses for the project. This grid aims to validate the resources, skills, and experience of the different project teams. The senior BIM manager can include supplemental requirements and training plans, which will be the responsibility of the BIM discipline managers to ensure that their teams have the required BIM capacities.

### 4.2.4. Discussion and Summary Regarding Solidarity and Propriety of Means

The common contractual norm of solidarity, used as a synonym for trust, is an essential part of Macneil's relational contract theory (Macneil and Campbell 2001). Trust is defined by Zucker as a set of expectations shared by all those involved in an exchange, and it has two major components: background and constitutive expectations (Zucker 1985). Background expectations refer to the reciprocity of perspectives and attitudes, whereas constitutive expectations are derived from independence from self-interest and intersubjective meaning (Zucker 1985). Luhmann first differentiated system trust from personal trust (Luhmann 2000)—a distinction furthered by Zucker, who created three categories of trust: intuitu personae, relational, and institutional. Intuitu personae trust draws its sources from the peculiar characteristics of people, for example belonging to the same family, ethnicity, or attachment to a professional body. Relational trust is based on past or expected exchanges and is thus a process-based trust. Institutional trust, on the other hand, is attached to a formal structure that guarantees the specific attributes of an individual or organization (Zucker 1985). Relational trust relates to the vision of trust adopted by governance scholars, who view it as a cyclical process of recurrent bargaining commitment and execution of events between partners, which can lead to increased project performance (Ring and Van de Ven 1994; Roehrich and Lewis 2014). This means trust, even if its pre-existence is necessary for collaboration to arise (Neu 1991), can be created and be a result of collaboration (Gambetta 1988). Trust can increase predictability, facilitate flexibility, and reduce transaction costs (Blomqvist et al. 2005; Luhmann 2000; Williamson 1975).

The norm of propriety of means recognizes that there are often several ways or paths to achieve the same goal, but Macneil believes that societal constraints determine which means are appropriate in a given context (MacNeil 1980a). As the exchange becomes more relational, the relationship becomes more complex, and the principles and practices of behavior within the relationship also become more complex. Habits, rules of thumb, and standard operating procedures evolve in a context of more relational exchange and add to (or even replace) external societal behaviors. The result is a more complex network of relations between exchange parties, and this network of relationships is an important facet of more relational exchange relationships (Nevin 1995). Propriety of means is a requirement of contracting parties to have adequate means to fulfill their obligations. Multiple pathways may be available to achieve appropriate outcomes, which means that there may be different options and methods available for a contracting agency to fulfill a commitment, but only a few can provide positive results for the project and all the contracting agencies. Therefore, the means employed should not affect the quality of the work or adversely affect any of the other contracting parties (van der Veen 2009). Table 5 summarizes the major relational differences regarding solidarity and propriety of means norms between project delivery methods, which are discussed below.

**Table 5.** Key takeaways for solidarity and propriety of means.

| | *Forms of work organization* |
|---|---|
| DBB | <ul><li>There is minimal interaction between the stakeholders.</li><li>There is an absence of formalized contractual mechanisms for stakeholder interaction.</li><li>Firms may take voluntary approaches with respect to use of the workshops.</li><li>There are autonomous contractual relationships between the public client and the design firms but not between the design firms.</li></ul> |
| DB | <ul><li>Workshops are held for the public client and end users to comment on the design development. Workshops are organized collaboratively between public client and contractor.</li><li>There are options with regard to voluntary approaches to work organization: (1) integrated design and construction process; and (2) risk management and identification through monthly workshops.</li><li>The design–builder has freedom regarding the internal organization of work through workshops and/or integrated processes.</li></ul> |
| CMGC/IPD | <ul><li>IPD workshops consist of a multidisciplinary team of stakeholders responsible for (1) defining vision, objectives, functional and technical needs, optimization measures, and validation of stakeholders' needs and issues; (2) choice of real estate options; (3) defining roles and responsibilities of stakeholders and validation of the IPD roadmap by external professionals; (4) the establishment of vision and performance objectives, analysis of project complexity, and optimization of construction program objectives; (5) strategies/synergies to improve concept and determining cost of strategies/synergies; (6) the optimization of design solutions to achieve the public client's requirements and confirmation of energy targets and energy-efficiency strategies; and (7) the development of detailed design, optimization of value and performance according to costs, choice of systems/equipment/material, definition of commissioning options, validation of constructability, and optimization of schedule and commissioning.</li><li>Value analysis: the public client, construction manager, and external professionals optimize the project to increase value while reducing costs.</li><li>Risk and opportunity analysis: the construction manager, risk manager, public client, and SQI estimators identify and evaluate risks and opportunities and their likelihood and impact on the project.</li><li>Commissioning and quality management: the public client, design professionals, construction manager, commissioning agent, end users, and others address these issues continuously in IPD workshops.</li><li>Visual coordination and interference detection: quality control through comments and validation of the design. There are two types of workshops: (1) interdisciplinary coordination workshop and (2) coordination and monitoring of the concept.</li></ul> |

**Table 5.** *Cont.*

| | |
|---|---|
| | *Common management structures and communication tools* |
| DBB | • There is an absence of common management structure.<br>• Formalization of the technical aspects of the communication process, including format, slips, and standardization of specifications, takes place.<br>• A project office, which is used for complete and timely coordination of key stakeholders, is created. |
| DB | • Voluntary creation of the project office takes place.<br>• The PMCC creates subcommittees to address any issues; promotes cooperative and effective communication between parties; and reviews matters relating to work, corrective work, delivery schedule, public interest, quality assurance, security, progress reports, commissioning, and final acceptance plan; The PMCC makes non-binding recommendations.<br>• Formalization of the communication process via transmission of written notices occurs.<br>• A live and interactive information management system and a relational database are formed to consult and share data. |
| CMGC/IPD | • The management committee holds meetings at the project office in a collaborative work environment (the construction manager, construction professionals, public client, and estimation professionals) with a focus on planning, monitoring, control of content, cost, schedule, change orders, risk, communications, and procurement.<br>• The design coordination committee consists of the construction manager and design professionals, is coordinated by the architect, and holds in-depth discussions regarding design issues that do not concern all stakeholders.<br>• The site coordination committee consists of the construction manager, project manager of construction professionals, and site supervisors, is coordinated by the architect, and offers technical expertise by verifying shop drawings, material specification sheets, and work samples.<br>• The steering committee consists of the project manager, IPD expert/facilitator, public client, agreement manager, and design team representative, and its responsibilities include (1) establishing general and specific objectives of the project, (2) establishing quantifiable performance objectives (energy, air quality, emissions, materials, etc.), (3) identifying impacts of construction choices, (4) initiating and integrating design process to reduce costs and timeframe, (5) identifying risks and mitigation measures (cost, timeframe, and expertise required), and (6) validating the project schedule at each stage and the budget on an ongoing basis.<br>• BIM includes (1) the construction manager disseminating information to all the stakeholders; (2) a digital collaboration platform; (3) use of discipline and federated models; (4) use of common software for interference detection workshops; (5) sharing and monitoring of conflicts through BIM collaboration format; and (6) an ongoing quality control process for best information modeling practices. |
| | *Immutability, mutability, competency, and expertise of personnel* |
| DBB | • The project manager of professional services provider must be a permanent resource.<br>• A change of the project manager or professional proposed on the tender is prohibited unless expressly authorized by the public client. If allowed, the replacement personnel must have equal or higher competence than the replaced personnel. |
| DB | • Key personnel include the project director, design manager, construction manager, quality manager, and finance officer.<br>• The public client's consent is necessary for changes in key personnel. The public client cannot refuse the change if the replacement has qualifications and experience at least equivalent to that of the replaced key personnel.<br>• Substitution procedure: The replacement of the subcontractor is subject to the consent of the public client. No replacement will be agreed to if (1) the replacement's activities are incompatible with the role of the public client, (2) the financial situation of replacement is unsatisfactory, (3) the skills and ability of replacement are unsatisfactory, or (4) the replacement agreement is not substantially similar to that of the replaced party. |

**Table 5.** *Cont.*

| | |
|---|---|
| DB | • Lenders' agreement substitution procedure: The replacement is subject to the consent of the public client. No replacement will be agreed to if (1) the replacement does not have the legal capacity or authority to contract with the public client; (2) the replacement does not have the necessary qualifications, experience, and technical competence; and (3) the replacement does not have adequate financial resources and subcontractors. |
| CMGC/IPD | • Personnel assigned by the construction manager in the tender may not be replaced without the public client's consent. If allowed, the replacement personnel must have equal or higher competence than the replaced personnel.<br>• Substitution of the construction manager's project manager or superintendent leads to a pre-established lump sum penalty.<br>• Substitution of a professional services provider's project manager, main designer, site supervisor, and BIM discipline manager leads to a pre-established fixed percentage penalty.<br>• The public client may request replacement of the construction manager's workers, superintendent, and foreman if considered incompetent, negligent, or otherwise undesirable.<br>• The construction manager is responsible for training.<br>• BIM training of the construction manager's team is the responsibility of, and at the expense of, the construction manager.<br>• The construction manager's team must demonstrate their BIM competence to the public client's team of BIM experts, which will comment upon their skills.<br>• BIM discipline managers, collaboratively with the senior BIM manager, must complete a BIM usage analysis grid to ensure their teams have the necessary resources, skills, and experience. |

Before discussing the results from the analyzed contracts, an important point must be emphasized. The Civil Code of Quebec governs the conduct of parties who bind themselves contractually (Civil Code of Quebec, RLRQ c CCQ 1991). Regulation 1375 states that parties shall conduct themselves in good faith—an essential concept of solidarity—both at the time the obligation arises and at the time it is performed. Regulation 2805 also states that good faith is always presumed. Even if it is formalized in the Civil Code, good faith is based upon considerations of morality and equity (Lefebvre 1996). Acting in good faith refers to adopting a general behavior that can be described as ethical, reasonable, or acceptable (Deslauriers 2012). Derived from this good faith obligation, the obligation to cooperate, which requires more proactive behavior, can be separated into two distinct obligations: the obligation of information and the obligation to facilitate the execution of the contract (Baudouin 2013). Therefore, any party who knows information of decisive importance must provide it to its co-contractor as soon as possible. Contracting parties must also favor the achievement of the project's objective and collaborate in the event of difficulties. The goal here is not to make a complete analysis of these obligations in Quebec's jurisdiction but to understand that the Civil Code contains obligations providing a strong relational foundation for the execution of construction projects—formalized obligations that apply to every project delivery method.

DBB contracts have no formalized processes of interaction other than the value analysis workshops held during the conceptual stage for the design professionals, which are led by an independent expert coordinator and may involve other external professionals. This contributes to the silo notion associated with the use of the DBB method. Even if an integrated design process was used at the planning stage of one of the contracts analyzed, there remained the problem that the firms conducting the process were usually not the ones concretely designing and constructing the infrastructure. The common management and communication measures of DBB contracts, such as the standardization of communication and the creation of a project office, are quite shallow in comparison with those of the other project delivery methods. However, DBB still provides for the immutability of key personnel assembled for the realization of the contractual obligations of the project, namely the project managers of the professional services providers, as well as the professionals themselves, who were identified in the procurement

process. The incorporation of an obligation of collaboration, present in every other contract analyzed, may appear to show good intent on behalf of the contract drafters, but this formalized obligation, which is non-imperative and has an evocative, indicative, or moral value, cannot work unless it is in the form of real mechanisms of collaboration outlined in the contract and, more specifically, relational trust mechanisms, thus making the insertion of such a clause practically superfluous. Normativity would be created internally in the relationship according to the repetition of interactions between the actors via relational trust mechanisms and not according to a formal obligation outlined in the contract to which no sanction or penalty is attached. We believe that an obligation of collaboration would amount to an obligation of love inserted into a prenuptial arrangement: whether the contract says love or collaboration, the effectuation of the contractual language can only arise through the actions of the parties, not by a blind and automatic compliance with a formalized obligation.

DB contracts stipulate that workshops occur to allow the public client and end users the opportunity to comment on the development of the design and to facilitate the incorporation of the public client's comments into the design data. As similar as it might seem to the IPD workshops, in terms of quantity and variety of stakeholders involved, the scope of work in DB workshops is lesser than that in IPD workshops, and the DB workshops are more focused on the construction aspects of the infrastructure realization. The DB team has a lot of flexibility and freedom in the choice of the means, methods, and processes to achieve the project objectives, and there is the possibility of a voluntary approach to work organization. With respect to the internal organization of work, the design–builder has the freedom to voluntarily set up an integrated approach through multidisciplinary workshops. Nonetheless, the only formalized DB workshops consist of an interaction between the public client and end users, with the design–builder commenting and reviewing the design data. Although the PMCC's authority is wide and represents a shift from single-point control to joint control for significant issues and matters of the construction process, the PMCC's role is to make non-binding recommendations to the parties. Therefore, the effective power stays with the relevant parties, but the apparent power is distributed amongst the stakeholders. The communication process between parties is also heavily formalized, and informal or verbal communication is not considered binding. The DB contractual measures are therefore defined by their voluntary and flexible aspects, elements evaluated by the public client during the procurement phase, and a heavily formalized relationship, in terms of communications, to ensure consideration of the public client's requirements for the project and quality, cost, and schedule. Furthermore, the immutability measures provided for by DB contracts reinforce the importance of the providers' propriety of means, which need to be thoroughly evaluated during the procurement process using quality- and qualifications-based criteria. This occurs through collaboration scenarios or the use of psychometric tests, which allow the public client to evaluate the behavior of individuals and their skills, such as reasoning, communication, leadership, or emotional intelligence (Jobidon et al. 2018).

At the other end of the spectrum lies CMGC/IPD, which formalizes a bevy of workshops regarding value, risk and opportunity analysis, commissioning and quality management, visual coordination, and interference detection. The use of BIM as a coordination and communication platform, which provides for the repetition of interactions between key stakeholders and a true multidisciplinary approach, creates a climate that can foster the development of trust and align the parties' expectations. With a total of three different governance committees, a steering committee, and the use of BIM, which focuses on data sharing, models, coordination before delivery, analysis software, and ongoing quality management processes governed by the project team, CMGC/IPD contracts demonstrate a propensity for formalized relational mechanisms. Unlike DBB and DB contracts, CMGC/IPD contracts stipulate that the substitution of the key personnel, ranging from the project managers to the superintendent, will lead to a pre-established fixed monetary penalty deducted from the payment of the contract, whether as a fixed percentage or a lump sum. These penalties exist to compensate the public client for the troubles, inconveniences, extra efforts, and additional delays that may be caused by the replacement of key personnel. Even though all delivery methods present a form of immutability for key personnel,

only CMGC/IPD contracts associate the replacement of key personnel with a monetary penalty, thus highlighting the key role and importance of the personnel contributing to the project.

*4.3. Flexibility: Third Parties or the Internalization of Processes*

The flexibility norm, referring mainly to the ability of stakeholders to adapt to changes using different mechanisms, such as third-party involvement and mutual planning, is another norm highlighting major differences between the project delivery methods. We have also included sustainability measures in this subsection, because external standards (e.g., LEED certification), a flexibility mechanism, are usually provided for by contracts. It was found that DB and CMGC/IPD contracts are more relational than DBB contracts. In DB contracts, more third parties are involved in the realization of the work, whereas CMGC/IPD contracts provide for mutual planning and agreements to agree and internalize sustainability and energy-efficiency processes instead of referring to external standards.

4.3.1. Third-Party Involvement: Independent Certifier and External Professionals

Direct third-party determination of performance refers to the use of an expert to determine contract content or performance. The involvement of third parties can also help foster trust, because they are not subject to the opportunism of the parties to a contract. While not offering a guarantee of smooth performance in the context of construction projects (MacNeil 1978), flexibility is still a relational indicator for contracts.

In Quebec, vertical infrastructure contracts are subject to the Framework Policy on the Governance of Major Public Infrastructure Projects (FPGMPIP) if they have a value of over 50 million dollars. The DBB contracts analyzed were subject to an older version of the FPGMPIP, which was updated in 2016. According to the old version of the FPGMPIP, the final business case of the project is subject to a continuous quality evaluation by a committee of independent experts composed of experts from various fields, such as architecture, engineering, finance, environment, economics, and project management, before final approval by the cabinet prior to the launching of public tenders for the realization of the project. The current version of the FPGMPIP does not use an independent expert committee but rather an internalized process: the Treasury Board, a standing committee of the executive council, whose function is to support ministries and public agencies in the management of resources in the public service, which now acts as the third party in the assessment of quality of the project's final business case throughout its development.

One of the DB contracts analyzed for this paper incorporated a new actor, the independent certifier, who is identified by a selection committee composed of the public client and the contractor. By agreeing to a certifier, a contractor consents to enter into an independent certifier agreement with the public client. The public client and the contractor have the right to attend and participate in all meetings, tests, inspections, audits, or other events that the independent certifier attends. Neither party can terminate the independent certifier agreement without the prior written consent of the other party, which cannot be reasonably denied. The parties also agree to cooperate with each other with respect to all matters covered by the independent certifier agreement. All instructions and declarations made by one of the parties to the independent certifier shall therefore be transmitted simultaneously to the other party. The payment of the independent certifier and all fees and expenses are equally shared by the contractor and the public client. The independent certifier has a duty to act in a fully independent and impartial manner.

The functions exercised by the independent certifier have to do with information and review measures. The contractor must ensure that the design database can be remotely accessed by the independent certifier to review drawings and electronically store and print the design data. The contractor must also provide the independent certifier with a detailed schedule for the execution of the project in an electronic format, as well as a monthly progress report identifying reasons for delays in the execution of the work and a plan to demonstrate the measures the contractor intends to take to eliminate or reduce the delay. The contractor must also produce a copy of his commissioning plan for

the independent certifier and invite him to attend and comment on all aspects of the commissioning. The independent certifier has the right to attend the PMCC meetings but has no right to vote. The contractor must provide the independent certifier with all the minutes of recommendations and meetings of the PMCC for review.

The independent certifier has access to the site to review the work if prior notice is given or can visit, subject to the consent of the relevant supplier, a site or workshop where materials or equipment are manufactured, assembled, or stored for use in the project to conduct a general inspection or attend a test or survey conducted as part of the work. The independent certifier may, once prior notice is sent to the contractor, require the contractor to open his work site for the inspection of any parts of the task or corrective task, if the independent certifier reasonably believes that such parts have defects in design or construction or that the contractor has not complied with the requirements of the public client.

The contractor must write and submit to the independent certifier a final acceptance plan to obtain the final approval of the contractor's work. An achievement certificate of a milestone can only be issued by the independent certifier once all the requirements described in the final acceptance plan for the relevant milestone are met. The issuance of the achievement certificate begins with the contractor notifying the independent certifier that the contractor believes that the necessary requirements for the granting of the certificate have been met. The public client then gives the independent certifier and the contractor its opinion on the contractor's compliance with the pre-established requirements. The independent certifier then determines whether the conditions for the issuance of the certificate are met, and the independent certifier either delivers the certificate or produces a detailed report outlining the steps the contractor must take to meet the conditions for earning the certificate. Once the contractor satisfies the necessary corrections to the work, the independent certifier issues the certificate, which is a final and binding decision on both parties. Any dispute relating to the decision of the independent certifier as to the issuance of the certificate may be subject to the dispute resolution mechanism outlined in the contract. If minor irregularities exist at the time the of the second and third milestones, the contractor delivers a list of those minor irregularities and an estimate of the cost and time necessary for their correction to the independent certifier. The independent certifier then prepares a final list of minor irregularities according to the comments provided by the public client. The independent certifier shall not refuse to issue the second and third milestone certificates solely because of the existence of minor irregularities. However, the independent certifier cannot issue certificates if the cost estimate of the minor irregularities is equal to or greater than 0.5% of the total cost of the work. The same process applies to the subsequent milestones. The main difference between this contract and the other DB contracts analyzed is that in the latter, the independent certifier's role is assumed by the public client and its project manager. The review and acceptance processes are thus internalized to a greater degree, and a gradual review process based on four different levels—review; correction of irregularities; correction of irregularities and resubmission; or rejection—occurs.

CMGC/IPD contracts involve very few third parties. One example of third-party involvement is the case in which the parties cannot agree on the choice of a mediator in the dispute resolution process. A mediator will then be chosen by an independent body, association, or professional order jointly designated by the parties. In the commissioning process, a professional called the commissioning manager is mandated by the public client to provide commissioning coordination services and must be independent of the design and construction professionals. The value analysis workshops involve a committee composed of the public client's project team and external project professionals to optimize the project and increase value while reducing project costs. The construction manager also acts as a type of third party in some of the tasks required by the project, such as producing cost estimates for the components of the mandate, which are independent of those established by the design professionals during the preliminary and final plan control stages, to align and reconcile the results with those established by the design professionals and to refine the forecast data on construction expenditures.

### 4.3.2. Mutual Planning and Incompleteness

Incomplete contracts are considered to be the most cost-effective form of governance for long-term contracts (Campbell and Harris 1993) and are included in the flexibility norm. Macneil also considered the use of agreements to agree to be flexible mechanisms, because they represent the futurization of the relationship instead of its presentation (MacNeil 1978). Moreover, mutual planning reflects the relational aspects of the contract, as the parties collaborate to jointly create the performance plan (MacNeil 1974b, 1978). The three different delivery methods analyzed here all displayed, to a different degree, forms of mutual planning and incompleteness.

Although the parties taking part in DBB contracts are not often required to work together, some flexible mechanisms have been stipulated in the contracts. For example, the architect must, in collaboration with the other professionals and the commissioning agent, choose the design strategies in accordance with (1) the orientations and requirements of the construction program and (2) the project budget and schedule. This allows all the professionals to participate in the preliminary plans and specifications stage. The architect must also prepare, in collaboration with the same stakeholders, all the technical documents required to proceed to the tendering stage and allow bidders to submit a fixed price bid for the construction of the infrastructure. The architect must also respect the master schedule, established in collaboration with the public client and its project manager, in the production of the final plans and specifications. The contracts also provide an agreement to agree to the following: if the public client makes changes to the mandate given to the service providers that will affect the costs and fees, financial responsibility limits may be revised through a new agreement negotiated between the parties.

A prime example of mutual planning in DB contracts is the preparation of the design development plan, which includes the planning of the different workshops by the contractor and the public client's representatives. The parties must also agree to develop an acceptable schedule for the workshops and determine who may participate in them. The public client in a DB contract is also responsible for the planning of the transfer of activities as part of the commissioning stage. The public client must prepare a transfer plan and submit it to the contractor a year before the anticipated provisional acceptance date. The contractor and the public client must jointly agree on the transfer plan, define the precise path for the transfer of equipment, and establish the necessary verifications and approvals to declare that the transfer is successful.

The other main mutual planning aspect of DB contracts is the review process of the public client's requirements regarding architecture and urban integration, structure and civil engineering, mechanical aspects, and electricity. The public client's main representative is responsible for evaluating different documents produced by the contractor, namely progress reports, construction plans, technical specifications, shop drawings, samples, and other relevant information pre-established in the contract. The public client's representative reviews and comments on each of these documents using gradual comments: review, correction of irregularities, correction of irregularities and resubmission, or rejection. Review means that the contractor complies with the requirements of the agreement and can implement them. Correction of irregularities means that the documents are generally in conformity with the requirements but contain minor irregularities discovered as part of the review carried out by the public client's representative, who must justify his decision. The contractor must correct these irregularities. provide the public client's representative with corrected copies, and implement the documents and the corollary comments. If, at the discretion of the public client's representative, the contractor has not corrected the irregularities, the contractor must resubmit his documents for review purposes at the contractor's own expense with respect to compensation and delays. Correction of irregularities and resubmission is the same process as correction of irregularities but means that the documents are not generally in conformity with the requirements. Rejection applies to documents with even more major irregularities and general nonconformity to the requirements.

One example of an agreement to agree in CMGC/IPD contracts occurs when the public client wishes to modify the scope of the mandate or tasks given to the construction manager. In this case, the original agreement must be reviewed and is subject to renegotiation. Another major component

of flexibility in these contracts is the ability to mutually plan the content of the IPD workshops. The steering committee has great latitude in the planning of the agenda and subject matter of the different IPD workshops. The contract provides a non-exhaustive list of subjects that can be discussed. Potential subjects include the following: (1) creating performance targets related to the public client's requirements, (2) understanding the construction program, (3) exploring different possible options for preliminary design, (4) analyzing and evaluating the development of options for the detailed design, and (5) establishing decision support mechanisms. The contract provides guidance pertaining to the possible elements addressed in these workshops, such as the analysis of project complexity for the public client's requirements and the clarification of functional program objectives to better understand the construction program.

In CMGC/IPD contracts, after the awarding of the construction management agreement, the construction manager must collaborate with the public client and all the stakeholders involved in the BIM approach to enable the public client and its team of BIM experts to draft the rules and modalities that will be included in the BIM management plan and the BIM agreement, which represents an agreement to agree.

The public client's team of BIM experts, which includes a senior manager, an integrator, and a coordinator, is mandated to support the project as a whole. It must, in collaboration with the other stakeholders involved in the BIM approach, establish the rules and procedures of the BIM agreement. The team must first examine the authorized uses of the models and how to identify the authors of the modeling elements. The different levels of detail (LOD) for the modeling elements must also be established by the team of BIM experts according to each stage of the project to allow for functional and technical analysis. Because professionals will be responsible for modeling these elements, the team must create a modeling grid enabling the identification of the LODs required for the BIM management plan. This grid will include various aspects, such as the matching of needs for property management; health, safety, and commissioning; the identification of elements with a characterization code for modeling; and the identification of construction classification standards. The process by which stakeholders involved in the BIM approach can exchange and share models at different intervals, the coordination system including the accuracy levels of the models, the file formats and the units used, the hosting location of files and folders, the transfer process and access to the model files, and the nomenclature conventions are also included in this modeling grid. Finally, the modeling grid establishes the consolidation and interoperability processes of the models according to (1) different software platforms, (2) the access rights to the models, (3) the coordination procedure of design and interference detection, and (4) the security requirements of the models. Thus, the team of BIM experts will work closely with the various stakeholders to ensure an optimized and collaborative implementation of the BIM approach to add value to the project.

If, during the process of developing the BIM agreement, a stakeholder who is to become a signatory of the agreement believes that the rules and procedures discussed in connection with the conclusion of the BIM agreement will result in a substantial change to the scope of the work or services, the stakeholder must notify the public client in writing. Failure to provide this notice is equivalent to a waiver by this stakeholder as to the effects of the relevant BIM agreement provision. Should it be necessary, the stakeholders that are part of the BIM agreement must, in consultation with all the other stakeholders involved in the BIM approach, review and propose a revision of the modeling rules and procedures. The contract clearly stipulates that the BIM agreement and management plan are of an evolving nature and, thus, will be modified and/or improved throughout the duration of the project according to the elements raised by the stakeholders. Any revision suggestion must be submitted to the main BIM manager. In the event that the main BIM manager agrees with the suggestion, the BIM agreement will be modified by the senior BIM manager in collaboration with the BIM discipline managers. Furthermore, the construction manager is responsible, in collaboration with the team of BIM experts and in compliance with the BIM agreement and management plan, for planning, coordinating, and managing the deployment of the BIM approach, such as the development of the modeling strategy.

This is yet another example of mutual planning between the different stakeholders that are participating in CMGC/IPD contracts. Finally, the main BIM manager must, in collaboration with the BIM discipline managers, elaborate the quality control processes covering the implementation of the BIM.

### 4.3.3. Sustainability Measures: From External Standards to Relational Trust

The different project delivery methods analyzed all have different ways of achieving the sustainability objectives of a project, whether through a different design process or through the goal of obtaining a certification from an external institution, such as LEED.

The first noticeable finding regarding sustainability is the pluralistic sources of the requirements. In DBB contracts, the design professionals must not only ensure compliance with the requirements of LEED Canada but must also conform to the Act Respecting the Conservation of Energy in Buildings (Québec Government n.d.) and the Regulation Respecting Energy Conservation in New Buildings (Québec Government n.d.). Professionals must also consider different energy-efficiency programs made available by the public energy supplier of the province. They must also coordinate the design of architectural components and build electrical and mechanical systems that limit annual energy consumption to no more than 75% of the consumption prescribed in the National Energy Code of Canada for Buildings. To achieve these requirements, the professionals must, at the pre-conceptual stage, ensure that all information regarding sustainable development is understood and used in the preparation of the concept. Afterwards, they must present to the public client their understanding of the construction program by identifying the main issues related to sustainable development and register the project with the Canada Green Building Council for LEED certification. During the conceptual stage, the architect must, in collaboration with the other professionals and the commissioning agent, propose to the client the main strategy for obtaining LEED certification. The preliminary plan and specification stage ensures that the architect and other design professionals present a detailed list of LEED credits selected for the project and the description of their application, as well as the studies, analyses, and computer simulations required to demonstrate the achievement of the performance needed to obtain LEED certification. Finally, during the final plan and specifications stage, the architect and other design professionals must present the public client with all the technical documents developed, which consider comments made by the public client, including instructions to contractors for commissioning and LEED certification. The architect is also responsible, during the construction phase, for studies, analyses, and computer simulations that reiterate conformity with LEED requirements. For example, the architect is responsible for performing an energy-efficiency simulation to achieve the public client's energy performance targets, studying the possibility of developing the site to reduce heat islands, and facilitating alternative transportation, as well as conducting an analysis to ensure that 10% of the materials used are extracted and manufactured regionally. The architect is also responsible for providing the necessary documentation for obtaining the certification.

Other measures used to ensure the realization of a green building that responds to integrated strategies and cutting-edge standards in terms of air quality, energy efficiency, and environmental performance include the need for the architecture firm to have a LEED-certified professional architect on the project team. To achieve the prescribed performance in terms of energy efficiency, the design professionals must perform a profitability analysis of each investment by considering the rising cost of energy, the operational and interest costs, and the return on investment for the useful life of each system involved. Finally, despite the fact that DBB contracts do not provide for IPD workshops, the public client can benefit from the architectural firm's expertise during the planning stage to provide input for the construction program. In one of the contracts analyzed, for example, the firm opted for an integrated design process to identify innovative strategies, techniques, and material that could be incorporated into the project with respect to energy, water management, heat islands, and the selection of materials. The architects must favor the use of local products and materials for the realization of the project.

DB contracts diverge from traditional contracts in more than one way. These differences mainly concern the process by which the sustainability requirements are achieved. As in DBB contracts,

most of the sustainable requirements concern LEED certification, but instead of the architect, it is the contractor who is responsible for obtaining the certification. The contractor must register the project with the Canada Green Building Council for LEED certification and obtain the certification for the project. The contractor must perform all tasks and complete the building to obtain the certification as soon as possible but no later than 30 months after the date of provisional acceptance. The contractor must also prepare and submit the necessary documents required for the certification and regularly inform the public client of the progress concerning data compilation and achievement of LEED credits in the monthly report. The PMCC also has a role to play by receiving and reviewing issues related to the work, including the LEED progress report, thus ensuring that the main parties to the contract are aware of the developments regarding the attainment of sustainability objectives. A main difference with respect to DBB contracts in comparison with those of the other project delivery methods are the consequences associated with the failure to obtain LEED certification. If the contractor does not obtain LEED certification within 30 months of the provisional acceptance or if the Canada Green Building Council issues a final and non-appealable decision pertaining to the issuance of the certification, then the contractor must pay the public client a predetermined lump sum as liquidated damages to compensate the public client for damages specifically related to the non-achievement of LEED certification.

To achieve the public client's pre-established energy targets, DB contracts provide specifically for the use of an energy model and energy simulation software that are authorized by CanmetENERGY, Canada's leading agency for clean energy research and technology. There are two different energy targets: the absolute target, which is a fixed value expressed in gigajoules, and the relative target, which represents energy consumption 45% lower than the reference simulation. The contractor's energy model considers the simulation hypotheses provided by the public client, until actual data, such as the occupation schedules, the density of equipment for each department, or the diversity of operation of the various equipment, are available. Once available, the data replace the public client's simulation hypotheses, and the absolute and relative targets are adjusted. The contractor must, at different stages of advancement (preliminary designs at 30% and 65%, and designs at 100% with shop drawings for construction and again three months before the provisional acceptance date) submit an updated version of the energy model to the public client to demonstrate that the infrastructure complies with the energy target in accordance with the review process previously described. The contractor and the public client each have an energy representative who must interact on all points related to energy consumption and the energy target. The contractor is responsible for providing the equipment to record and measure the consumption and demand of each type of energy and to perform detailed monitoring of energy consumption, such as heating, ventilation, and air conditioning (HVAC). This energy monitoring equipment must also allow for the identification of energy consumption trends for the analysis of collected data. Furthermore, the equipment must provide relevant information on issues such as comparisons between the real energy consumption and the energy target, as well as alerts in the event of any major discrepancies between these two values.

The data collected by the contractor must be secured and submitted to the public client at the same time as the energy consumption certificate. This certificate, covering the period from the provisional acceptance date until the end of the energy calibration period, must be submitted by the contractor to the public client and must include the energy consumption of the building in gigajoules, as well as the temperature data for each month. Another major difference between DBB and DB contracts is that the contractor in DB contracts guarantees that the building is designed and constructed so that the energy consumption per building area is equal to or less than the absolute energy target and that the energy consumption reaches the relative target or offers a higher reduction. In the case in which the absolute target is not reached, the contractor has a discretionary choice as to whether to provide the public client with a plan of modifications or improvements to be made to attain the absolute energy target, subject to review and comments by the public client, or pay liquidated damages to compensate the public client for damages specifically related to the failure to achieve the absolute target. If the contractor chooses

the first option, he must undertake the work at his own expense and pay the liquidated damages if the corrective work does not permit the attainment of the target.

The commissioning agent is another stakeholder that is involved in achieving the sustainability objectives of the project by developing and implementing a building envelope sustainability plan that must cover all stages of the building's life cycle, including (1) the selection of the building envelope components; (2) the development of details at the design stage, which is a quality control process during construction; and (3) maintenance for the building's operation stage. To do so, the commissioning agent must collaborate closely with the project stakeholders through exchanges and reviews of plans during the design phase and during the construction stage by reviewing shop drawings, conducting site visits, and so on. The commissioning agent is also responsible for preparing and submitting the required documentation to obtain the corollary LEED credit. Finally, and as previously stated, one of the DB contracts analyzed promoted the integration of key stakeholders in an integrated design and construction process. This interdisciplinary team was tasked with continually refining the project definition by working on the construction aspects of the project. This integrated approach was complemented with life cycle cost analysis, but no other specific tasks related to sustainability issues were assigned to the interdisciplinary team.

Similar to DBB and DB contracts, CMGC/IPD contracts provide for a multiplicity of sources concerning the sustainability objectives of the project. The infrastructure project must consider the 16 principles of sustainable development legalized in the Sustainable Development Act, the eco-responsible criteria of the public client, the project's sustainable development orientation plan, and Quebec's action plan on climate change, which promotes, among other things, the improved energy performance of buildings, the reduction of greenhouse gases, and the use of renewable energy sources for heating. However, the projects analyzed did not target LEED environmental certification. One of the projects analyzed nonetheless stated that the professional services providers and the construction manager must follow the LEED procedure for the credits identified by the public client in the functional and technical program and incorporate LEED measures in the work to achieve the objectives of the project's sustainable development orientation plan. The professional services providers and the construction manager also needed to monitor the LEED measures to ensure compliance. Moreover, the construction manager was required to help integrate all the LEED requirements identified in the functional and technical program, as well as to consider the LEED scorecard, a table highlighting the 16 principles of sustainable development, the LEED criteria, and the public client's eco-responsibility criteria attached to the program. Furthermore, the construction manager was responsible for participating in their monitoring, updating, and improvement, if necessary. All the requirements identified by the program were also enforced by the subcontractors. The construction manager also participated in planned coordination meetings for sustainable development.

Workshops, which are planned by the steering committee and which bring together a bevy of stakeholders, can help define the economic, environmental, and social objectives of the project. During the workshops addressing performance targets related to the public client's requirements, the stakeholders can mutually define the sustainable development and energy-efficiency goals, as well as the strategies to achieve and measure each objective. During the real estate solution workshop, the parties can develop optimized and integrated design solutions, optimize the work with all the professionals, and confirm energy targets and energy-efficiency strategies. The preliminary design workshops, which explore various possible design options, can identify energy-efficiency measures, while the decision support workshops enable the stakeholders to perform energy simulations and life cycle cost analysis to validate the solutions in relation to the mutually established performance targets and objectives. The design and construction professionals are required to perform a simulation to validate the energy performance targets of the program, to document the measures necessary to achieve such a performance, and to conduct an energy-efficiency study indicating the technical measures proposed and their added values according to criteria such as cost, return on investment, and external financial incentives. One of the CMGC/IPD contracts analyzed also provided for the ex post evaluation of energetic performance.

Finally, BIM serves as a communication and visualization tool during the IPD workshops and other quality workshops, including value analysis, design audits, and sustainable development, to boost exchanges and optimize decision-making. BIM is also useful for supporting the transfer of data for energy-efficiency analysis exercises. The BIM process also assists the stakeholders responsible for monitoring the system performance one year after equipment delivery, carrying out a bioclimatic analysis and performing an energy analysis. If the operating and maintenance costs are reduced, the objective is achieved.

4.3.4. Discussion and Summary Regarding Flexibility

Flexibility concerns the ability to adapt to the changes certain to arise during the relationship. The transactional pole is characterized by using rigid agreements, which are difficult to adjust or adapt, such that revisions cannot happen or are hindered by the contract. The contracts closer to the relational pole are supple agreements that can be modified and for which revisions are expected. Because of the inherent incomplete nature of long-term arrangements, flexibility measures are considered to be the most cost-effective form of governance (Campbell and Harris 1993). The main types of flexibility mechanisms identified in the literature are the use of external standards (i.e., LEED), the direct determination of performance by third parties (i.e., the role of the architect in traditional construction contracts), one-party control of terms, the use of cost (i.e., cost-plus contracts), and agreements to agree (MacNeil 1978), as well as redetermination and renegotiation mechanisms (Crocker and Masten 1991; Crocker and Reynolds 1993; Nystén-Haarala et al. 2010). The use of flexibility mechanisms can lead to cost and time savings in future disputes (Nystén-Haarala et al. 2010). The notions of third-party involvement, mutual planning processes that also relate to solidarity, the incompleteness of contracts, and agreements to agree were included in the analysis of Quebec's public procurement contracts. Sustainability measures are also included in the discussion here, although such measures represent a great example of the difficulty of analyzing contracts through the relational contract lens because of the interdependent nature of norms: as will be explained in greater below, the transition from using external standards to relational trust processes positions sustainability measures in between the flexibility and solidarity norms, making it difficult to categorize them neatly. Table 6 summarizes the major relational differences regarding the flexibility norm between the project delivery methods analyzed here, which are discussed below.

**Table 6.** Key takeaways for the flexibility norm.

| | *Third-party involvement: independent certifier and external professionals* |
|---|---|
| DBB | • Historically, an independent experts' committee was responsible for the assessment of the project's final business case quality throughout its development. <br> • Recent internalization of the process, has now put this assessment under the authority of the Treasury Board. |
| DB | • One DB contract incorporated an independent certifier, who can review drawings, access the site to review the works, and require the contractor to open the work for inspection. The independent certifier issues achievement certificates of milestones when irregularities are corrected. <br> • The other DB contracts internalize this process, and the independent certifier's role is assumed by the public client's project manager. |
| CMGC/IPD | • A mediator is chosen by an independent body when parties cannot agree on the nomination. <br> • An independent commissioning manager provides commissioning coordination services. <br> • External project professionals are involved in value analysis workshops. <br> • Construction managers sometimes act as third parties, such as in the production of cost estimates independent of those established by the design professionals. |

**Table 6.** *Cont.*

| | |
|---|---|
| CMGC/IPD | • Visual coordination and interference detection serve a quality control purpose through comments and validation of the design. There are two types of workshops: (1) interdisciplinary coordination workshop and (2) coordination and monitoring of the concept. |

*Mutual planning and incompleteness*

| | |
|---|---|
| DBB | • The architect, design professionals, and commissioning agent collaborate to choose the design strategies to meet construction program requirements, schedule, and budget.<br>• The architect, design professionals, and commissioning agent prepare technical documents in collaboration.<br>• The architect and the public client collaborative to establish the master schedule.<br>• Agreement to agree: substantial changes to the service provider's mandate (costs and fees) by the public client may trigger negotiations to revise the agreement. |
| DB | • Mutual planning of the design development plan occurs.<br>• There is a mutual agreement to develop an acceptable schedule for workshops and their participants.<br>• There is mutual acceptance of the infrastructure transfer plan.<br>• Mutual planning occurs via a gradual review process (review, correction of irregularities, correction of irregularities and resubmission, rejection) of the public client's requirements regarding the following: (1) architecture; (2) urban integration; (3) structure and civil engineering; (4) mechanical aspects; and (5) electricity. |
| CMGC/IPD | • The steering committee jointly plans the IPD workshops.<br>• Agreement to agree: if public client modifies the scope/work given to the construction manager, the original agreement must be reviewed and is subject to renegotiation.<br>• Agreement to agree: mutual planning of a portion of the BIM agreement. The public client's team of BIM experts and all the stakeholders involved in the BIM approach must collaboratively establish the rules and procedures of the BIM agreement (use of models, levels of detail (LOD), modeling grid, etc.).<br>• The BIM agreement and management plan are of an evolving nature and will be modified/improved throughout the project.<br>• The construction manager and team of BIM experts are responsible for collaboratively developing the modeling strategy;<br>• The main BIM manager and BIM discipline managers must collaboratively elaborate the quality control process covering the implementation of BIM. |

*Sustainability measures: from external standards to relational trust*

| | |
|---|---|
| DBB | • There are many sources of requirements including the following: LEED, the Act Respecting the Conservation of Energy in Buildings, the Regulation Respecting Energy Conservation in New Buildings, energy efficiency programs, and the National Energy Code of Canada for Buildings.<br>• The architect, design professionals, and commissioning agent collaboratively propose a strategy to obtain LEED certification.<br>• The design professionals must perform profitability analysis of each investment by considering the cost of energy and operation, interest costs, and return on investment for the useful life of each system involved. |
| DB | • The contractor is responsible for obtaining LEED certification.<br>• The PMCC reviews issues relating to the LEED progress report.<br>• Failure to obtain LEED certification forces the contractor to pay a predetermined lump sum as liquidated damages.<br>• The contract specifically provides for the use of an energy model and energy simulation software to achieve the public client's pre-established energy targets.<br>• The absolute target represents a fixed value.<br>• The relative target represents energy consumption 45% lower than the reference simulation.<br>• The contractor submits updated versions of the energy model to the public client when the preliminary designs reach 30%, 65%, and 100% of completion. |

**Table 6.** *Cont.*

| | |
|---|---|
| DB | • The contractor and the public client each have an energy representative.<br>• An energy consumption certificate is necessary for the provisional acceptance of the work.<br>• The contractor guarantees to the public client that energy consumption will be equal or less than the absolute target and that energy consumption will reach or exceed the relative target for energy reduction. If the contractor does not comply, the contractor can make modifications or improvements or pay liquidated damages.<br>• The commissioning agent develops and implements a building envelope sustainability plan covering all stages of the building's life cycle, collaborating with the project stakeholders during the design phase. |
| CMGC/IPD | • There is an additional requirement of a project sustainable development orientation plan.<br>• The projects analyzed here did not target LEED environmental certification.<br>• However, the project used LEED procedure as a framework to achieve the objectives of the project's sustainable development orientation plan.<br>• The economic, environmental, and social objectives of the project are developed through IPD workshops mobilizing a bevy of stakeholders.<br>• All the stakeholders in the IPD process mutually define sustainable development and energy-efficiency goals, as well as strategies to achieve objectives and measure performance.<br>• One of the CMGC/IPD contracts analyzed here provided for the ex post evaluation of energy performance. |

In DBB contracts, the value analysis workshops held during the conceptual stage to meet the goals of the construction program, budget, and schedule are led by an expert coordinator independent of the project team and may involve other external professionals. These third parties are not evaluating the performance of the contract but represent added value in terms of having a different perspective and potentially bringing up solutions or expertise. DBB contracts also stipulate the mutual selection of design strategies, the mutual preparation of technical documents, and the mutual planning of the master schedule—mechanisms that allow for a minimal level of latitude in the mutual planning of the relationship. Additionally, DBB contracts provide an agreement to agree when significant changes are made by the public client. As to sustainability measures, pluralistic sources of requirements create a bevy of prescriptive and performance requirements that must be considered by the design professionals, who, in DBB contracts, do not have many formalized contractual processes to optimize the design between the various stakeholders. One of the architectural firms analyzed here implemented an integrated design process during the planning stage, which fell into the category of voluntary approaches evaluated by the public client during the ex-ante procurement process. DBB contracts are considered to provide few formalized flexibility mechanisms. DBB contracts provide little latitude for the mutual planning of the relationship, forcing the design professionals to cooperate without a formalized structure or mechanism to help with the collaboration. Furthermore, these design professionals are limited by the scope of their mandate. Thus, DBB contracts are closer to the transactional pole of the exchange spectrum.

One of the DB contracts analyzed presented a higher level of flexibility through the use of an independent certifier, whose duties mainly concerned information and review measures. In the other contracts analyzed, the independent certifier's role was assumed by the public client and its project manager, thus internalizing the review and acceptance process with the use of a gradual review system based on four different levels: review, correction of irregularities, correction of irregularities and resubmission, or rejection. The internalization of these processes leads to a more relational process by which the parties must achieve project goals without external aid. DB contracts also provided for a mutual planning process through the design development plan. However, it cannot be overlooked that the public client–contractor relationship is one of power and vertical control, based on comments and reviews, in contrast to the CMGC/IPD process, in which the public client has an active designer role in a horizontal decision-making process. This mutual planning process concerning major aspects of

conception and construction rather resembles a review and conformity process to the pre-established contractual requirements and is therefore more of a mutual performance review process. It should be noted that during the pre-contractual procurement stage, DB contracts are mutually planned and optimized via multiple interactions through a collaborative digital platform between the public client and the bidders, therefore reducing the need for mutual planning in the contract. With respect to sustainability measures, the main difference between DB and DBB contracts is the consequence in DB contracts associated with the failure to obtain LEED certification, which forces the contractor pay the public client a predetermined lump sum as liquidated damages to compensate for damages specifically related to the non-achievement of LEED certification. Another major difference between DB and DBB contracts is that the contractor in DB contracts guarantees to the attainment of the public client's absolute energy target. Failure to do so forces the contractor to pay liquidated damages to compensate the public client for damages specifically related to the inability to achieve the target. Obtaining certification and attaining energy targets are formalized in the contract but still rely on ex-ante evaluated voluntary approaches concerning how the DB firm intends to achieve sustainability. This reinforces the need for a rigorous evaluation process focused on quality and for a sustainable development implementation plan.

CMGC/IPD contracts involve very few third parties given the fact that the process is very internalized and mutualized. All the DBB contracts analyzed, as well as one DB contract, included third parties that determined the performance of the contract but in a rather limited fashion. Some third parties are involved in CMGC/IPD contracts to provide expertise and add value to the project, but one of the DB and all the CMGC/IPD contracts internalized the performance determination process—a trait specific to relational contracts as opposed to the neoclassical form of third-party involvement. One of the most flexible mechanisms provided by CMGC/IPD contracts is the ability for the steering committee to plan the IPD workshops: this offers a wide range of possibilities, such as assessing the public client's requirements, exploring different preliminary designs, determining the sustainable and energetic performance objectives, and so on. The non-exhaustive nature of the list of elements treated in the thematic workshops allows the stakeholders to mutually determine the most important aspects to be analyzed throughout the IPD process and offers them latitude in terms of planning. Furthermore, the option for stakeholders to take part in a CMGC/IPD contract that uses BIM to mutually plan the BIM agreement is probably one of the most flexible mechanisms found in all the contracts analyzed, as it represents a prime example of an agreement to agree. The herculean task of establishing the rules and procedures to be incorporated in the BIM agreement is of utmost importance for the efficient running of the BIM process, and the process uniquely involves a bevy of different stakeholders in a mutual planning exercise. The contract also clearly stipulates that the BIM agreement and management plan are of an evolving nature and thus will be modified and/or improved over the course of the project according to the issues raised by the stakeholders. As to the sustainability measures, while parties are not always contractually obligated to obtain LEED certification, the stakeholders still often rely on the spirit of the program as a guide to achieve the sustainability objectives. The main difference from the other delivery methods is the mutual planning of the sustainability aspects of the project. In this paper, we consistently highlight the pertinence of workshops for building trust amongst team members, facilitating the comprehension of the project, and finding innovative solutions. This is also the case for the sustainability objectives. Indeed, in the workshops, which are mutually planned by the steering committee and which bring together a bevy of stakeholders, the parties can address the performance targets related to the public client's requirements, mutually define the sustainable development and energy efficiency goals, and choose strategies to achieve and measure each objective. Interestingly, in one of the CMGC/IPD contracts analyzed, an obligation to record and measure the global energy consumption of the infrastructure approximately one year after the provisional acceptance date and the start of operations was included in the contract, demonstrating the need for ex post evaluation of performance. This process ensures that the systems function as required in terms of energy efficiency, because the infrastructure has been tested for four seasons. While DB

contracts offer more mutual planning processes, such as the preparation of the design development plan, the public client–contractor relationship is a form of vertical control based on comments and reviews. On the other hand, the public client in CMGC/IPD contracts has an active designer role in a horizontal decision-making process. Therefore, for the achievement of sustainability objectives, we can observe a shift from compliance with external standards in DBB contracts to a focus on relational trust, particularly using workshops. Vertical hierarchical integration of power relations leads to formal rationality, and under such circumstances, it is harder to consider sustainability objectives as anything other than a mechanical consideration—some type of check-list rather a value-creating mechanism that goes beyond norms and conventions. Horizontally integrated teams, such as those in CMGC/IPD projects, allow for a more organic and rationally substantive consideration of sustainability and energy-efficiency issues, especially coupled with the co-designer role played by the public client. The shift from compliance to external standards to relational trust mechanisms also reinforces the need for a propriety of means norm closer to the relational pole and thus procurement based on the intrinsic qualities of organizations and individuals assembled to build public infrastructure. The variety and multiplicity of stakeholders involved, the flexibility and latitude in the decision-making process, the mutually planned rules and procedures, and the reliance on relational trust to achieve sustainability objectives push CMGC/IPD contracts closer towards the relational pole of the exchange spectrum.

### *4.4. Harmonization of Conflict: Progressive Dialogue, Mediation, and Arbitration*

The final norm highlighting minor differences between the project delivery methods is that of harmonization of conflict. While all the project delivery methods studied provided for some sort of gradual conflict resolution process, DB and CMGC/IPD contracts emphasize starting the process at the lowest possible level of decision-making and are thus more relational.

### 4.4.1. Hybrid Forms of Dispute Resolution: From Amicable Negotiation to Reliance on Courts

The more relational contracts usually internalize dispute resolution instead of relying on litigation. DBB contracts do not provide measures or mechanisms for contractual dispute resolution, because these are already provided for in legislation. The Regulation Respecting Certain Service Contracts of Public Bodies (2008), which is applicable to design professionals such as architects and engineers, states that the public client and the service provider must attempt to amicably settle any difficulty that may arise from a contract by resorting to the dispute resolution provision in the contract. However, such provisions often do not exist, rendering this section of the regulation inapplicable. Because the matter cannot be settled this way, the dispute is referred to an arbitrator, a court of justice, or an adjudicative body. On the other hand, construction contracts are subject to the Regulation Respecting Construction Contracts of Public Bodies (2008). As is the case with service provider contracts, disputes between the public client and the contractor must be settled amicably. However, unlike their counterpart, the regulation of dispute settlement in construction contracts does not rely on dispute resolution clauses that are supposed to be present in the contract, but it relies instead on the mechanisms provided by the regulation. Once the contractor sends a notice of dispute, both the public client and the contractor must try to settle the dispute through their respective representatives. If negotiations fail, the public client or the contractor may require mediation by sending a notice to the other party within 10 days of the preceding step. This process must be carried out within 60 days of the receipt of the notice of mediation, otherwise the negotiation process is terminated. The mediator is mutually chosen by the public client and the contractor. The parties and the mediator then establish rules applicable to the mediation, such as the duration and the role of the mediator, which is usually to assist the parties in clarifying the dispute and defining their positions and interests and to discuss and explore mutually satisfying solutions. The fees of the mediator are equally split between the two parties unless a different arrangement is reached. Each party retains their rights and remedies if no arrangement is found between the parties following the mediation.

As is the case in the analyzed DBB contracts, each party of a DB contract must first try to settle disputes by amicable negotiation. However, DB contracts go a step further by asking that such disputes be settled at the lowest possible level of decision-making before initiating the dispute resolution mechanisms provided in the contract. If the parties cannot settle their quarrel at the lowest level of decision-making, either party can deliver to the representative of the other party a notice of dispute. Upon receipt of this notice, the representatives of the public client and the contractor must attempt to resolve the dispute in good faith. Each representative shall provide to the other, promptly and diligently, the facts, information, and documents required to facilitate the settlement. If this last step does not enable the parties to reach an agreement within 60 days, either party may request that the dispute be dealt with by a qualified, experienced, and independent expert, excluding the courts. Three months after the signing of the DB contract, the parties must agree on experts to be appointed for different types of disputes. These experts will be automatically assigned to such disputes as they arise. If the parties fail to agree on an expert at this time, either may appeal to the Superior Court of Quebec. The appointed expert must determine the procedure for an effective resolution of the dispute. He may solicit arguments and documents from both parties, require evidence, order parties to prepare and provide documents, test results, inspect project activities, and convene meetings between the parties. The expert essentially has the necessary latitude to request any information necessary to make a final decision within 10 days of his nomination. The expert has the discretionary power to split his fees equally between the parties or attribute them to a specific party. Subject to the parties' exceptional right to refer the dispute to arbitration or to the courts, the parties agree that the decision of the expert is final and binding and may not be appealed or referred to arbitration, courts, or other dispute resolution mechanisms. Both parties therefore waive all their rights to appeal the expert's decision.

Exceptional cases in which parties can rely on arbitration to settle disputes exist in DB contracts. The first case is if the amount awarded to a party by the expert is between $250,000 and $10,000,000. The second case is if the result of the expert's decision ensures that a party poses or refrains from doing an act estimated by both parties to cost between $250,000 and $10,000,000. The final case involves a non-monetary dispute considered to be important and meaningful by either party. In this case, the parties agree to settle the dispute through arbitration but not through the courts. The documents prepared for the expert's decision are considered to be confidential, and the parties are not obligated to transfer them to the arbitrator. As is the case with the expert, the parties must, no later than three months after the conclusion of the DB contract, agree on the arbitrator to be appointed. The disputes are settled by a single arbitrator, unless a party notifies the other of its desire to refer the issue to an arbitration board, in which case three arbitrators will be responsible for settling the dispute. Each party chooses an arbitrator, and those arbitrators will in turn mutually elect the third arbitrator. The arbitrators appointed by the parties must always remain neutral and act impartially. The arbitrators essentially have the same powers as the experts, except that they can also issue interim orders; grant interlocutory, interim, or permanent injunctions; order an in-kind repair; and award costs including legal fees and expert fees according to the criteria they deem appropriate. During the exercise of their discretionary power to allocate their fees to either party, the arbitrators must consider the wish of each party to see the costs incurred by each of them in proportion to the relative success that each party obtains in the arbitration process. The arbitrator's decision is final and binding, and both parties waive all their rights to appeal. Subject to an express agreement between the parties to refer the dispute to the arbitration process, parties may refer the dispute to the Superior Court of Quebec if the dispute has a potential value of $10,000,000 or more or if it involves public health or safety issues.

CMGC/IPD contracts have different ways of dealing with the settlement of disputes depending on whether the construction manager or the service professionals are involved. In cases involving construction managers, the public client and the construction manager must try to amicably settle the dispute through their respective representatives following the receipt of a notice. If these negotiations do not completely resolve the dispute, the public client or construction manager may, by sending a written notice within 10 days of the end of the previous step, require mediation, which must then be

completed within 60 days. The mediator is mutually chosen by the public client and the construction manager. If no agreement is reached, a mediator will be chosen by an independent body, which is an association or a professional order jointly appointed by the parties after the signature of the contract. The mediator is responsible for helping the parties to identify their differences and their positions and interests. The mediator must also favor dialogue and explore mutually satisfactory solutions to resolve the parties' differences. Unlike the expert in DB contracts, who independently decides the rules and processes of the dispute settlement, parties to a CMGC/IPD contract define, jointly with the mediator, the rules applicable to the mediation, as well as the role and duties of the mediator. The parties agree to share any information on which they intend to rely during the mediation process. Because the parties must sign a confidentiality agreement beforehand, this information is confidential. The mediator's agreement developed and signed by the parties also states that the mediator will not represent or testify on behalf of either party in subsequent, potential legal proceedings. Unless otherwise agreed upon, the fees and expenses must be shared equally between the parties. If the mediation does not enable the parties to reach an agreement, they retain all their rights and remedies and can decide to resolve the dispute through an arbitrator or a court of law.

In dispute settlements for professional services, the public client and the service provider commit to a process of gradual dialogue consisting of three steps involving stakeholders of different authorities. The first step is a dialogue between the project managers of the service provider and the public client. The second step is between the service provider's boss and the immediate supervisor of the public client's project manager. The third step is between a senior partner of the service provider and the hierarchical superior of the immediate supervisor of the public client's project manager. At any stage during this process, the public client may invite the final user of the infrastructure to participate in the process. Disputes settled according to this process must not affect the management of project parameters, such as content, quality, cost, and schedule. This dialogue process, heavily reliant on mutual understanding and the intuitu personae qualities of the stakeholders, is made without admission and prejudice to the rights of the parties, such as recourse to an arbitrator or the courts. The contract also provides for the use of a mediator or a third-party conciliator if parties fail to reach an agreement regarding the conclusion of a fixed-price agreement for the fees and expenses of the contract. In this case, the third party is chosen by mutual agreement and at the expense of the public client and the service provider jointly, and both parties will define the rules applicable to this process. The mediator's or conciliator's decision is not binding, as it takes the form of a recommendation.

4.4.2. Discussion and Summary Regarding Harmonization of Conflict

The possibility of criticizing, investigating, and having a free discussion is a trust-generating mechanism (Quéré 2005). In other words, giving contractors or professionals the opportunity to express themselves is a factor that contributes to the perception of procedural fairness (Folger and Greenberg 1985; Lind and Tyler 1988). Having a voice, meaning having the ability to express one's opinion with respect to allocation decisions, is a typical rule of procedural fairness (Thibault and Walker 1975) and involves allowing people affected by a decision to present relevant information (Folger 1977). Usually, participants who can express their opinion judge the outcome of a decision more fairly than those who are not allowed to do so (Van den Bos et al. 1998). Such individuals will have the impression that their interests are protected in the long term and will consider the process to be fair (Thibault and Walker 1975). Giving members a voice in the process allows them some degree of indirect control over the decision (Korsgaard et al. 1995). By creating conditions for the use of a meaningful fair procedure, such as giving a voice to contractors and professionals in the case of appealing decisions, public clients will show a greater willingness to cooperate and could be perceived as more trustworthy (De Cremer and Tyler 2007). It is desirable for public clients to be sensitized to both the need to promote trust and to create a fair procedural climate (De Cremer and Tyler 2007). Furthermore, Macneil, Williamson, and Fuller have each expressed that the use of third parties in conflict resolution and performance evaluation has advantages over litigation by creating flexibility and

filling the gaps in incomplete contracts (Fuller 1963; MacNeil 1978; Williamson 1979). As Williamson pointed out, the difference between litigation and the use of an arbitrator is that continuity—understood as the completion of the contract—is presumed under arbitration mechanisms, while the presumption is much weaker when the parties resort to litigation (Williamson 1979). Relationships are thus fractured if a dispute reaches litigation (Friedman 2011). Asken also pointed out that the use of adjustments through arbitration does not endanger the project, but it favors continuity (Asken 1972). Table 7 summarizes the major relational differences regarding the harmonization of conflict norm between the project delivery methods, which are discussed below.

**Table 7.** Key takeaways for the harmonization of conflict norm.

| | *Hybrid forms of dispute resolution: from amicable negotiation to reliance on courts* |
|---|---|
| DBB | • Service providers must refer disputes to an arbitrator, court of justice, or adjudicative body.<br>• The contractor and the public client must try to amicably settle a dispute. If a settlement cannot be reached, they can mutually choose a mediator. |
| DB | • Parties must seek amicable settlement of disputes. Such disputes must be settled at lowest level of decision-making.<br>• An expert decision process, in which the expert is mutually agreed upon, may be utilized.<br>• Parties can rely on arbitration if settlement amount awarded by the arbitrator is between $250,000 and $10,000,000.<br>• Non-monetary disputes are settled through arbitration. |
| CMGC/IPD | • The construction manager and the public client must pursue the amicable settlement of disputes. If no settlement is reached, mutual nomination of mediator is utilized. If no settlement is reached through mediation, parties can rely on arbitrators or the courts.<br>• Professional service providers and public clients engage in a three-step gradual dialogue process: (1) dialogue between the project managers; (2) dialogue between the service provider's boss and the supervisor of the public client's manager; (3) dialogue between the senior partner of the service provider and the superior of the public client's manager's supervisor. If no settlement is reached, the parties may mutually name a mediator or conciliator, who makes non-binding recommendations. Finally, the parties may have recourse to an arbitrator or the courts. |

Although the external regulatory framework of DBB contracts provides for the amicable settlement of disputes by resorting to the contract's dispute resolution clauses, the inexistence of these clauses forces the parties to refer the dispute to an arbitrator, a court of justice, or an adjudicative body. DBB service provider contracts have more of a transactional nature, because there is virtually no effort made to internalize the conflict. On the other hand, DBB construction contracts are subject to the internalization of disputes, because the parties can mutually choose a mediator if an amicable settlement cannot be reached—a somewhat peculiar measure considering the intuitu personae nature of service contracts, which are usually more relational.

As is the case with DBB contracts, each party to a DB contract must first try to settle disputes by amicable negotiation. However, DB contracts go a step further by insisting that such disputes be settled at the lowest possible level of decision-making before initiating the dispute resolution mechanisms provided by the contract. The conflict harmonization process of DB contracts is therefore very gradual, starting as an amicable settlement at the lowest possible decision-making level. Afterwards, parties can rely on an expert decision process, which can lead either to arbitration or to court depending on the monetary value of the dispute or on its nature. While there is no complete waiver of the right to bring the claim to court, the process is deeply internalized and impartial because of the mutual choice of experts and arbitrators, although a claim could be made that the arbitrator's powers are somewhat closer to that of a judge in a traditional court setting.

Unlike the expert in DB contracts, who individually decides the rules and processes of the dispute settlement, parties to a CMGC/IPD contract jointly define with the mediator the rules applicable to the mediation, as well as the role and duties of the mediator. CMGC/IPD contracts also do not formalize an arbitration agreement in their contractual documents. The dispute settlement mechanisms of professional services contracts are even more internalized than those in construction manager contracts. This dialogue process, heavily reliant on mutual understanding and the *intuitu personae* qualities of the stakeholders, is made without admission and prejudice to the rights of the parties, including recourse to an arbitrator or the courts. CMGC/IPD contracts thus rely more on the internalization of the dispute settlement process than do DBB and DB contracts, even though each contractual form has a gradual dispute settlement process. The service provider's settlement process is probably the most relational process of all, because it relies on dialogue between the stakeholders at different decision-making levels. CMGC/IPD contracts also rely on non-binding decisions made by mediators or conciliators—an important note adding to the relational aspects of those contracts. Therefore, every delivery method emphasizes the need to amicably resolve disputes before turning to other methods of conflict resolution. While DBB contracts provide for no other immediate solution than those provided in the regulatory framework, DB and CMGC/IPD contracts provide for the settlement of disputes by relying on third parties, such as experts, mediators, or arbitrators, which is a more internalized and relational mechanism.

## 5. From IPD-ish to IPD and Theoretical Implications: Further Discussion

As previously discussed, Quebec's use of CMGC/IPD contracts does not fall into the category of pure IPD, but rather resembles an integrated design process associated with the construction manager at risk delivery method. To truly profit from pure IPD advantages, such as meeting or exceeding the public client's expectations regarding the budget, schedule, change orders, design quality, sustainability, trust, and collaboration (Cohen 2010; El Asmar et al. 2013; Sive 2009) and to evolve from IPD-ish to IPD, Quebec should ensure that it enlarges the scope and role of stakeholders in the process. It should also ensure that the construction manager is involved not only at the planning phase, but also at the start-up phase of the process, and it should also ensure that parties are linked together through a multi-party agreement that provides for risk and reward sharing.

### 5.1. Broaden Stakeholders Involved

To achieve not only environmental sustainability, but also social and economic stability, researchers have focused on a system-based approach (Eccleston and Smythe 2002; Holmberg 1998; MacDonald 2005; Robèrt 2000). The British Standards Institution also reiterated the importance of identifying the roles and concerns of stakeholders (Lam et al. 2010). An extremely important aspect in the preparation of sustainability specifications is the holistic inclusion of different stakeholders and the involvement of the public in project planning (Berke 2002). As previously stated, CMGC/IPD contracts provide for the inclusion of multiple and diverse stakeholders in multidisciplinary workshops. The contracts also stipulate that any other collaborator whose presence is deemed necessary at some point in the process should be invited. The CMGC/IPD team thus has the discretionary power to include other key stakeholders in the process. This power should be exercised by the CMGC/IPD team so as to include stakeholders such as the community and neighborhood in which the project is realized, subcontractors, suppliers, financial firms, town planners, facilities managers, buyers, suppliers, technicians, and craftsmen. It would of course be unnecessary and impractical to include all of these stakeholders throughout the start-up, planning, design, and construction phases, but their inclusion in the relevant phases could help achieve the objectives of the project. Public clients and the steering committee should thus, during the first workshop, not only to identify a larger scope of relevant stakeholders and their positions, but also the ideal moment to include them.

## 5.2. Involvement of the Construction Manager during the Start-Up Phase Instead of the Planning Phase

Regarding the ideal moment of inclusion of key stakeholders, an important means of achieving a fully integrated team is the early involvement of the construction manager. The life cycle of the project is separated into four major phases: (1) the start-up phase, which includes the creation of the program and the analysis of real estate options; (2) the planning phase, which includes defining the vision of the project and the performance objectives, an exploratory phase, and the development of the real estate solution and of the design; (3) the construction phase during which plans are finalized and work is initiated and finalized; and (4) the operational phase. In CMGC/IPD contracts, the construction manager is brought in during the final moments of design development during the planning phase, whereas the contract stipulates that the IPD process begins at the start-up phase during which the public client and the program implementation team are involved. This means the construction manager does not take part in the crucial start-up phase of the project in which parties define the vision and objectives in relation to the client's requirements; examine the functional and technical needs; look for optimization measures; validate and clarify the needs and issues of the various stakeholders; analyze real estate options; and obtain a consensus on a preferred real estate option. Contract drafters should therefore consider including the construction manager on the steering committee, which is composed of a project manager, an IPD expert or facilitator, a public client's representative, an agreement manager from the expertise department of the public client, and a representative of the design team, to enhance the variety and quality of expertise by including constructability skills. Furthermore, including the construction manager in the early stages of the process could help foster trusting relationships, which was found in the governance literature to facilitate the development of the relationship and to help parties resolve contractual issues arising in later project phases (Roehrich and Lewis 2014).

## 5.3. Rewards Sharing: Fixed Remuneration Scales and Absence of Benefits Sharing Mechanisms

Another major aspect differentiating IPD-ish project delivery methods from true IPD methods is the lack of risk/reward sharing mechanisms, which relates to the norm of reciprocity exemplified by the non-use of multilateral contracts. The reciprocity norm was not specifically analyzed in the earlier parts of this paper, because there is a flagrant lack of these mechanisms in Quebec's public procurement contracts. With respect to reward sharing, Quebec's procurement regulations provide for a fixed-remuneration scale for professional service providers to standardize rates applicable to different services or tasks (Jobidon et al. 2018; Tarif Architectes n.d.; Tarif Ingénieurs n.d.). This remuneration mechanism thus represents a binding form of benefits sharing, enhances focus on individual goals rather than project goals, and negatively affects the value-creating process (Jobidon et al. 2018). Professional services contracts do not provide for ex post benefits sharing mechanisms or financial incentives related to energy performance, except the disincentive penalties for not obtaining LEED certification and not meeting energy performance goals in DB contracts. Professional service providers are not the only parties suffering from a lack of reciprocity. The roughly estimated ex-ante price of the construction manager's contract is subject to redetermination given the changes that will necessarily take place during the realization of the project. This redetermination mechanism is not a negotiated process between the public client and the construction manager, but rather takes the form of an amendment to the contract by the public client according to his estimation of actual construction, supply, and other contracts' costs. While the control of terms by one party and the notion of price are considered to be flexibility mechanisms under neoclassical contract law (MacNeil 1978), their use is also tinted with power and does not necessarily represent a mutually acceptable solution. Asken pointed out that when joint venturers tie themselves to fixed percentages of the contract price, gross inequities can result and instead suggested the use of tentative percentages and arbitral adjustments to reallocate the income according to the actual work performed (Asken 1972). Researchers also identified the need for public bodies to have soft budget constraints, in comparison with ex-ante fixed prices, to facilitate renegotiation in the event that the ex-ante price is underestimated, which could cause the contractor to delay the completion of the project, and necessitate that the public client pay a price to

enhance innovative practices, such as the CMGC/IPD delivery method (Bös and Lülfesmann 1996; Rogerson 1992). Public clients in CMGC/IPD contracts should thus implement negotiated benefits sharing mechanisms based on the performance of the parties and particularly regarding the energy and sustainability objectives to ensure reciprocity and adhesion to project goals rather than individual ones.

*5.4. Risk Sharing: External Regulations Precluding Use of Liability and Claim Waivers*

With respect to risk sharing, Quebec's civil code provides in regulation 2118, that unless the contractor, architect, or engineer who, as the case may be, directed or supervised the work, as well as the subcontractors who performed the work, can be relieved from liability, they are held solidarily liable for the loss of the work occurring within five years after its completion, whether the loss results from faulty design, construction or production of the work, or defects in the ground (Civil Code of Quebec, RLRQ c CCQ 1991). This regulation establishes a perfect solidary obligation and not an *in solidum* obligation, in which there is no relationship between the debtors (Levesque 2009). Regulation 2119 provides for the situation in which the parties may be relieved from such liability. The design professionals, architect, or engineer may be relieved from liability if they can prove that the defects are not a result of their opinion, plans, or supervision of the work. Contractors may be relieved by proving that the defects effectively result from an error of the design professionals selected by the client. The same goes for the subcontractor, who may additionally argue that the defects result from the contractor's decisions. Furthermore, each may be relieved by proving the defects result from the client's decisions regarding land, materials, subcontractors, experts, or selection of construction methods (Civil Code of Quebec, RLRQ c CCQ 1991). In Quebec's civil law system, contracts may include clauses limiting or excluding liability, but these rules are subject to various limitations, such as certain contracts in which such limits on liability are precluded by law. This is the case for professional service contracts, according to their respective code of ethics. For example, the architect's code of ethics stipulates in regulation 17 that architects must assume full civil liability and may not include in a contract for professional services a clause that directly or indirectly excludes all or part of such liability (Code of Ethics of Architects n.d.). Although some form of risk sharing exists in the ex post assessment of the work, it is somewhat limited by the impossibility of using liability and claim waivers, which were identified by the literature as an integral part of a variety of IPD arrangements (Ballobin 2008; Dal Gallo et al. 2009; Zhang and Chen 2010). This is especially of concern in the case of CMGC/IPD contracts using BIM. The intricate, multidisciplinary, iterative, and collaborative process would be hindered by the lack of a true risk sharing and risk-allocating mechanism, which favors project success over individual success. Quebec's civil code may provide an answer as to what would happen in the case of traceability failures in a BIM process. Regulation 1480 stipulates that, where several persons have jointly participated in a wrongful act or omission, which has resulted in injury, or have committed separate faults, each of which may have caused the injury, and where it is impossible to determine, in either case, which of them actually caused the injury, they are solidarily bound to make reparation (Civil Code of Quebec, RLRQ c CCQ 1991). This hypothesis has not yet been tested by courts and represents a fragile foundation for the risk sharing notion of the reciprocity norm, which would be strengthened using liability and claim waivers in public procurement contracts.

*5.5. Theoretical Implications*

The theoretical implication arising from this research modulates Williamson's ideal-type governance structures for different types of transactions, which is based on Macneil's three-way contract classification. Williamson states that classical contracting applies to standardized transactions, that neoclassical contracting is needed for occasional, non-standardized transactions, and that relational contracting should be used for recurrent and non-standardized transactions (MacNeil 1978; Williamson 1979). Williamson's ideal-type governance structures are based on the premises that uncertainty is always of an intermediate degree and that recurrence constitutes repetition of a type of transaction, rather than repetition of the same type of transaction with the same firm or the same

individuals, leading to the conclusion that the form of governance best suited for the construction of a building would be neoclassical governance. However, construction delivery methods have evolved towards more collaborative means and methods presenting a higher level of uncertainty, especially considering the transition from contractual formalism and standardization to a more mutual, evolving, and organic approach. Moreover, the major actors in public markets, such as Quebec, are frequently the same, especially regarding major projects, which emphasizes the factors of uncertainty and recurrence, in addition to being significantly more idiosyncratic, thus making plausible the need for a more relational governance framework for the completion of construction works. Furthermore, a formal distinction in three categories, shies away from Macneil's concept of a transaction spectrum framed by transactional and relational poles, which is more dialectical than dichotomic. It is possible to assert that some project delivery methods, such as CMGC/IPD and pure IPD, exist between the neoclassical and relational governance types, mainly because the public procurement context still represents a mix of formal and informal structures, exacerbated by external and internalized discrete norms, such as the pre-established and fixed remuneration system, the impossibility of using liability and claims waivers, the nonexistence of handshake agreements, and rigid budget constraints, to name a few. Public construction and professional service contracts should therefore serve as a foundation for a more relational development of the relationship. Research has identified that many alliances begin with the use of formal mechanisms and then, over time, employ more informal ones (Gulati 1995) and that contracts help in creating and maintaining long-term relationships (Frankel et al. 1996). When new contractual arrangements, such as CMGC/IPD using BIM, emerge, the industry has not yet adapted and cannot respond as automatically to such processes as it can in the case of the traditional DBB method. The binding formal content of a contract serves as a foundation for the relationship in the context of innovation and contractual novelty. Subsequently, customs and change will certainly develop before being crystallized in standard contracts, regulations, or legislation. The formalized obligations will, of course, be distorted and adapted to the specific context of the relations, but for the moment, they have the virtue of serving as a foundation for structuring exchanges, which furthers the idea that both hard, or transactional, and soft, or relational, factors are important for relational contracting-based approaches, such as CMGC/IPD (Motiar Rahman and Kumaraswamy 2005).

## 6. Conclusions and Summary

This paper builds on the literature regarding the transactional and relational features of different public construction project delivery methods, namely design–bid–build, design–build, and construction manager/general contractor coupled with an integrated design process. With respect to the DBB method, many of the problems already identified in the literature were confirmed by our analysis, such as the silo effect and the lack of project goals. However, it was found that additional relational mechanisms, such as the use of an integrated design process during the planning phase and, more specifically, during the creation of the construction program, can create value while taking into account sustainability and energy-efficiency goals. The DB method was difficult to differentiate from the CMGC/IPD method in relational terms, but it can be argued that the process is characterized more by the voluntary internal approach of the firm and on the public client's validation and control process rather than on a contractually binding and multi-firm approach that creates value, thus facilitating the active and direct involvement of the public client. This reinforces the need for a rigorous ex-ante evaluation of the DB firm's program and the competency of the key personnel assigned to the project. It also underlines the importance of a procurement process that allows for repeated interactions between the different actors to identify the needs of the public body. On the other hand, the mechanisms identified in CMGC/IPD contracts demonstrate the need for strong institutional capacity and a competent internal staff of the public client given their involvement in the project, encouraging the mobilization of a team of internal experts and bolstering the co-designer role of the client. The CMGC/IPD method is the most relational project delivery method found in the jurisdiction of Québec, mainly due to its propensity to use relational trust mechanisms based on interactions

between a multitude of stakeholders, as exemplified by the multitude and diversity of workshops, mutual governance structures and planning processes, the more active role of the public client, and the immutability of key personnel. The use of the BIM process also complicates roles and makes them more multidimensional and thus more relational, in addition to creating additional interactions and serving as a communication and coordination tool. However, an IPD-ish delivery method such as the CMGC/IPD has not reached the maturity level of a pure IPD method because of the lack of risk and reward sharing mechanisms and the absence of a multilateral contract. Nonetheless, the CMGC/IPD method represents an alternative for jurisdictions that do not have the option of using pure IPD. Regarding the achievement of sustainability and energy-efficiency objectives, we highlight how Quebec's contracts are shifting from compliance to formal external standards, such as LEED, to relational trust mechanisms, in addition to mutually defined performance objectives, showing a clear relational shift via increased internalization. Furthermore, CMGC/IPD involve multidisciplinary teams and processes, an increasing complexity of roles, the use of relational trust mechanisms and iterations, a horizontal rather than vertical distribution of power, and the consideration of life cycle impacts as an example of substantive rather than formal rationality. The analysis presented in Section 4 with respect to the precise mechanisms and the concepts and trends identified should help policymakers, contract drafters, and public clients interested in implementing relational contracting practices in public construction projects. The limitations of this paper include the fact that only the formalized part of the relational content was analyzed in order to exhibit contracts as a governance tool that can serve as a relational foundation for long-term arrangements. Therefore, the external and institutional contexts, as well as the actual development of the relationship, were not part of the scope of this research, even though they represent a notably important aspect of relational contracting. Future research should thus focus on the interaction between external, internalized, and internal norms in order to holistically evaluate the transactional and relational features of public procurement contracts for construction projects and professional services. The impact of the BIM process on the relational aspects of the relationship should also be isolated and thoroughly analyzed, considering the rapid growth in popularity of BIM in the industry.

**Author Contributions:** G.J. reviewed the literature, analyzed the contracts, wrote and conceptualized the paper. P.L. and R.B. contributed to the discussions on the content, to the review of the article and to the final edit.

**Funding:** The authors are grateful to the Natural Sciences and Engineering Research Council of Canada for the financial support through its IRC (IRCPJ 461745-12) and CRD (RDCPJ 445200-12 and RDCPJ 445200-12) programs, as well as the industrial partners of the NSERC industrial chair on eco-responsible wood construction (CIRCERB).

**Conflicts of Interest:** The authors declare no conflict of interest. The funders had no role in the design of the study; in the collection, analyses, or interpretation of data; in the writing of the manuscript, and in the decision to publish the results.

## Abbreviations

| | |
|---|---|
| BIM | Building information modeling |
| CMGC | Construction manager–general contractor |
| CMGC/IPD | Construction manager–general contractor coupled with integrated project delivery |
| DBB | Design–bid–build |
| DB | Design–build |
| FPGMPIP | Framework policy on the governance of major public infrastructure projects |
| IPD | Integrated project delivery |
| LEED | Leadership in Energy and Environmental Design |
| PMCC | Project management control committee |
| SQI | Société québécoise des infrastructures |

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
