# Peer review of "Comparison of Quebec’s Project Delivery Methods: Relational Contract Law and Differences in Contractual Language"

_laws_

Round 1
Reviewer 1 Report
The authors investigated the association between relational aspects of contracts and the achievement of sustainable and energy-efficient infrastructure by employing a grounded theory approach. The study is well conducted and the methods used are appropriate. The data is presented clearly.
Overall, this is an outstanding study with excellent data and prudent analysis and interpretation. This manuscript was an absolute pleasure to review, and I cannot remember the last time I reviewed a manuscript and had no substantive edits or comments.
Author Response
Hello,
Thank you for your review. Following the comments from the other reviewers, changes were made to the paper.
The paper was corrected by a professional language editor.
A table of key takeaways of the literature review was included in section 2.1.4
A figure and further explanations were added to the methodology section
Multiple tables were added to the discussion sections to help the presentation of the results and facilitate the reader's comprehension. The discussion sections were also pruned out of redundancies.
Sustainable construction (2.3) was merged to Integrated Project Delivery (2.2.4) to de-emphasize the correlation between relational contracting and sustainability.
We hope those significant changes will help the paper and it's comprehension and that you will be satisfied with those modifications.
Best regards,
Reviewer 2 Report
The authors are to be commended for undertaking such complex research. In particular, the Conceptual Background is very strong, and provides a meaningful, detailed legal basis from which to discuss the evolution of construction contracting. In particular, the discussion of translational versus relational contracting is particularly apt for construction.
Where the research struggles, however, is in the application of grounded theory to actual construction contracts/projects. In particular, the anecdotal nature of the discussion is overly simplistic and in some cases misleading. Example statements which need further qualification include,
l. 1246 The other role created by the use of BIM in the CMGC/IPD contract is that of a BIM discipline manager, which is a BIM specialist from each of the firms of construction professionals taking part in the project.
l. 1510 potential physical conflicts are therefore detected prior to the bidding period
l. 2186 [In DBB] “The architect is also responsible, during the construction phase, of studies, analyzes and computer simulations to reiterate the conformity to LEED requirements, as well as providing the necessary documentation for obtaining the certification.”
While these statements can be true under certain circumstances, they are not necessarily representative across industry (in this case, Quebec) or contract type. The research does not effectively demonstrate that contract type is, necessarily, more impactful (with regard to sustainability metrics) than variations in specific project characteristic, stakeholders, schedules, sustainability requirements etc.
The biggest weakness of the paper is its overall treatment of sustainability. Specifically, with regard to sustainability (as introduced (l. 862)) the authors initially express the conclusion that (l. 882) “the interplay between the parties to a construction contract, and especially the role of designers and their interaction with the other parties of the supply chain, is considered as a huge facilitator of green infrastructure construction.” Therefore, isn’t it a rather banal conclusion that (l. 911) there is a, “link between relational contracting and the capacity to better achieve environmental and sustainable objectives in conjunction with formal processes.” In short, the paper is unconvincing with regard to providing new discoveries or proof of correlations between relational contracting and sustainable infrastructure.
The authors might consider reframing the contribution of the research to be more focused on the general examination of the impact of differences in relational contracting on construction projects (in Quebec). If the authors still wish to claim (l. 2820) that “results, for the precise mechanisms, and discussions section, for the concepts and trends identified, should help policy-makers, contract drafters and public agencies interested in achieving sustainable energy-efficient infrastructure.” The authors should be more explicit and provide specific recommendations for contractual language and relationships that will help to achieve such goals. In addition, it is recommended that the authors for Table 2. explain how the “terms used” were generated. In particular, the terms for “Flexibility” seem somewhat arbitrary.
Again, the authors are to be commended for their important work, and are encouraged to include more feedback from construction professionals/experts before resubmitting.
Additional Minor edits include:
l. 266 the use of the word “unadapted” is unclear
Typos in references. Example:
(Cheung et al. 2006) Cheung, Sai On, Kenneth T. W. Yiu, and Pui Shan Chim. 2006. How Relational are Construction Contracts. Journal of Professionnal Issues in Enginerring Education and Practice 132: 48-56.
(infrastructures -)
Author Response
Hello,
Thank you for your review.
First of all, an additional figure and explanations pertaining to the terms used (with a specific example of flexibility) regarding grounded theory were added.
The contribution of the research was modified to focus more on the relational differences in contractual language between delivery methods rather than the sustainability aspect. It was already the central topic, but the title/abstract/introduction/conclusions were modified to further reflect it.
As for sustainability, section 2.3 (sustainable construction) was merged with 2.2.4 (Integrated Project Delivery). We agree that the term "correlation" used throughout the paper regarding the possible link between relational contracting and achievement of sustainability was too strong of a word, and it was changed. However, we do feel that the shift from conformity to external standards to relational trust mechanisms is pertinent for the analysis of differences in relational contractual language and provides a new pathway to achieve sustainability objectives, even though we agree that it does not effectively demonstrate that contract type is, necessarily, more impactful (with regard to sustainability metrics) than variations in specific project characteristic, stakeholders, schedules, sustainability requirements etc., which was never our objective.
Since the SQI (who shared the contractual documents with us) is one of the 10 biggest public clients in Canada and that we analyzed their standardized contractual documents, we believe the findings are representative of Quebec's situation.
We included tables to be more explicit regarding the relational differences in contractual language between delivery methods, but we cannot provide the exact contractual language because the documents are in French.
The discussion sections were pruned out of redundancies and some additional meta-text was added. The paper was corrected by a professional language editor.
l. 1246 The other role created by the use of BIM in the CMGC/IPD contract is that of a BIM discipline manager, which is a BIM specialist from each of the firms of construction professionals taking part in the project.
We modified the wording, but the paragraph explains the BIM discipline manager's role and needed expertise.
l. 1510 potential physical conflicts are therefore detected prior to the bidding period
Some additional details were provided, but we think this might have been a translation problem: conflict are actually clashes.
l. 2186 [In DBB] “The architect is also responsible, during the construction phase, of studies, analyzes and computer simulations to reiterate the conformity to LEED requirements, as well as providing the necessary documentation for obtaining the certification.”
Examples of studies, analyzes and computer simulations were provided.
As you can see, significant changes were made to the paper, and we hope you will be satisfied.
Best regards,
Reviewer 3 Report
This was a highly interesting paper that, research-wise, merits publication. However, I would recommend the authors at least clarify some points and develop the overall clarity of structure, language and message of the paper to help guarantee it the readers and impact it deserves.
One of my primary concerns is the relationship of methodology and results. In Section 3 the authors describe how they have used Macneil's work on contractual norms to construct a listing of contract terms that can be used to evaluate the comparative 'relationality' of contracts using NVIVO. I find this approach to be highly interesting. However, once the authors have listed the contract terms they use for occurrences of Macneil’s contractual norms, it remained unclear what the results are—while I understand the analysis was conducted in the French it would have been interesting to hear more of how the material was used. Instead, the authors move right on to Section 4 which focuses on comparing the different representations of Macneil's contractual norms found in different types of construction contracts. While this comparison is also interesting in itself, it resembles more a close reading of the contracts themselves and thus the exact relation between the methodology presented in Section 3 and the results presented in Section 4 eludes me. There seems to be some kind of quantification in process, as of Macneil's ten norms that the authors have tied to specific textual representations five are found to provide major differences between the contracts and one minor differences. How was this result reached, and what about the remaining four norms? How did Nvivo help in reaching these findings, by facilitating a closer reading of the contracts?
Another issue is the focus of discussion in Section 2. I find it extremely commendable that the authors try to open up the rather abstract distinctions between Macneil’s phases of contract law. At the same time this part could be more rigorously analytical instead of the current rather loose literature review. Why have exactly these works been chosen for discussion instead of others, such as existing analytical literature reviews (e.g. Schepker, Oh, Martynov & Poppo, The Many Futures of Contracts: Moving Beyond Structure and Safeguarding to Coordination and Adaptation, 40 Journal of Management (2014) 193)? What are the key takeaways for the reader? Now, for example, the takeaways of 2.1.4 are partly embedded in that extremely long section. Furthermore, some of the takeaways might merit further elaboration. For example, in lines 508–512 the authors claim that research has overlooked Williamson’s ‘trilateral’ contracting and focused on the opposition of classical formal governance and relational governance. However, recent approaches to governance such as Gereffi et al.’s global value chain theory and Richard Locke’s empirical work on contract would instead seem to imply that something resembling ‘trilateral contracting’ has a prominent place in recent theorizations of governance. Furthermore, Williamson’s focus on ’frequency/recurrence’ seems to be less important for some newer governance approaches, such as Gereffi et al.’s global value chain theory which, while indebted to Macneil and Williamson, has replaced ‘frequency/recurrence’ with ‘the ability of actors to codify complex transactions’ (for one genealogy from Macneil to Williamson, Gereffi et al. and Locke and other work see Jaakko Salminen, From National Product Liability to Transnational Production Liability: Conceptualizing the Relationship of Law and Global Supply Chains, 2017, Part 3). On the other hand, perhaps this finding is limited specifically to the public procurement/construction context in which case it would suffice to make that point linguistically clearer.
The overall structure and contents of the paper were good and enjoyable to read. On the other hand, all sections could do with further editing and a thorough language check. There are many typos and small errors. Taken individually, these are for the most part minor, such as errata like listing Macneil’s "Contracts: Adjustment of Long-Term…" as being published in Northwestern University Law Review in 1977 instead of the actual 1978 (see page header in that paper—the volume is 1977–78 but the publishing year of the issue is 1978), and similarly Macneil’s The Many Futures of Contract should be dated 1974, not 1973. Similarly, there are numerous linguistic errors ranging from typos (e.g. lines 12 and 118) to vocabulary (line 12: not relational but perhaps relative or comparative?; line 136: not energetic but perhaps energy efficient or efficient?; line 180: instead of ‘and even impossible’ perhaps ‘if any’?). While these too are minor individually, similar errors are so prevalent throughout the whole paper that fixing them would improve readability to a major degree. Thus I would warmly recommend the whole manuscript to be checked by a professional language editor.
As such, the paper is also quite extensive and potentially overwhelming for readers not versed in both contractual theory and the construction context. For example, towards the end in Section 4 the plethora of acronyms and abbreviations (most of which were opened up in Section 2, in some cases tens of pages earlier) would merit a table of abbreviations to improve readability. Additional meta-text highlighting core themes to the reader and a pruning out of less important 'obiter' passages would also help. So would brief 'summary sections' of key takeaways—now these are generally embedded in extremely long sections, such as the end of 2.1.4, the several lengthy 'discussion' sections and also the long concluding section 5. Now, for example, the reader easily loses track of the sustainability aspects of the paper due to massive amounts of other material provided. Additional meta-text and pruning might also help direct the authors towards a more analytical and focused approach to the extensive material and themes discussed. A more radical alternative, but one perhaps also worth consideration, would be to separate the literature review and the contractual study into two different papers. Ultimately, though, I suppose this depends on Sustainability—many established legal journals today seem to be moving towards an upper limit of less than 10000 English words.
I do hope you take the time to consider these structural, linguistic and other proposals as they would no doubt help guarantee your paper the readership and impact that your research would merit.
Author Response
Hello,
Thank you for your review.
First of all, an additional figure and explanations pertaining to the terms used and the general grounded theory methodology were added. While NVivo does facilitate a closer a reading of the contract, it helps create categories and the theory (which contracts are more or less relational). We hope the clarifications will help the understanding of the methodology employed.
The contribution of the research was modified to focus more on the relational differences in contractual language between delivery methods rather than the sustainability aspect. It was already the central topic, but the title/abstract/introduction/conclusions were modified to further reflect it. Section 2.3 (sustainable construction) was merged with 2.2.4 (Integrated Project Delivery) (as per another reviewer).
We included multiple tables (key takeaways) to be more explicit regarding the relational differences in contractual language between delivery methods, but we cannot provide the exact contractual language because the documents are in French. The discussion sections were pruned out of redundancies (see discussions in section 4) and some additional meta-text was added (see section 5). The paper was corrected by a professional language editor who made numerous changes to the paper.
Regarding section 2 (literature review), we created a table of key takeaways for the reader in section 2.1.4. The objective of this section was not to effectuate a systematic literature review on contract, like the paper you suggested (e.g. Schepker, Oh, Martynov & Poppo, The Many Futures of Contracts: Moving Beyond Structure and Safeguarding to Coordination and Adaptation, 40 Journal of Management (2014) 193), but rather to identify the general principles of relational contract law and the literature gaps that this paper can help fill. The trio of Gereffi/Locke/Salminen was quite interesting to read and was briefly mentioned in the review, but the global supply chain field is broader than public procurement for construction works/complex performance so we made this point linguistically clearer.
A table of abbreviation and a table of figures and tables were added before the introduction.
As you can see, significant changes were made to the paper, and we hope you will be satisfied.
Best regards,
Round 2
Reviewer 2 Report
The authors have done a commendable job addressing reviewer comments. In particular, the addition of the summary tables makes the (lengthy) paper more accessible to the reader. In addition, the treatment of construction contract types has been expanded. The treatment of sustainability has been improved. However, the authors may want to further acknowledge limitations of the research within this domain. Other than that, the manuscripts appears ready for publication.